

# Quantum minimal surfaces from quantum error correction

Chris Akers[1*] and Geoff Penington[2,3†]

**1** Center for Theoretical Physics, Massachusetts Institute of Technology,
Cambridge, MA 02139, USA
**2** Center for Theoretical Physics, University of California,
Berkeley, CA 94720 USA
**3** Institute for Advanced Study, 1 Einstein Dr, Princeton, NJ 08540 USA

\* cakers@mit.edu, † geoffp@berkeley.edu

## Abstract

We show that complementary state-specific reconstruction of logical (bulk) operators is equivalent to the existence of a quantum minimal surface prescription for physical (boundary) entropies. This significantly generalizes both sides of an equivalence previously shown by Harlow [1]; in particular, we do not require the entanglement wedge to be the same for all states in the code space. In developing this theorem, we construct an emergent bulk geometry for general quantum codes, defining "areas" associated to arbitrary logical subsystems, and argue that this definition is "functionally unique." We also formalize a definition of bulk reconstruction that we call "state-specific product unitary" reconstruction. This definition captures the quantum error correction (QEC) properties present in holographic codes and has potential independent interest as a very broad generalization of QEC; it includes most traditional versions of QEC as special cases. Our results extend to approximate codes, and even to the "non-isometric codes" that seem to describe the interior of a black hole at late times.

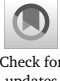

# 1 Introduction

## 1.1 Background and motivation

In the last decade, the ideas of quantum error correction have revolutionized our understanding of holography [1–11]. Central to this progress is the notion of the entanglement wedge of a boundary subregion $B$. Roughly speaking, this is a bulk subregion $\mathbf{b}$ that encodes the same information about the state as the boundary subregion $B$. An important signature of the relationship between these two regions is a relation between their entropies known as the QES prescription [12–14],

$$S(B)_{V|\psi\rangle} = \frac{A_B(\mathbf{b})}{4G} + S(\mathbf{b})_{|\psi\rangle}. \tag{1}$$

The entropy $S(\mathbf{b})_{|\psi\rangle} := -\mathrm{tr}[\psi_\mathbf{b} \log \psi_\mathbf{b}]$ is the von Neumann entropy of bulk state $\psi_\mathbf{b} = \mathrm{tr}_{\bar{\mathbf{b}}}[|\psi\rangle\langle\psi|]$ on subregion $\mathbf{b}$, and likewise $S(B)_{V|\psi\rangle}$ is the entropy of boundary region $B$ in the state dual to $|\psi\rangle$, obtained by acting with the bulk-to-boundary map $V$. $A_B(\mathbf{b})$ is the

area of the bulk surface bounding **b** and homologous to *B*, and *G* is Newton's constant. The entire right hand side of (1) is known as the *generalized entropy* of **b**.

For appropriate holographic states, (1) is true for any boundary region *B*, so long as one defines the entanglement wedge **b** to be bounded by the minimal quantum extremal surface (QES) homologous to *B* [14]. This means that the region **b** is (i) an extremum of the generalized entropy under local perturbations of its bounding surface and (ii) among all such extremal regions, **b** has minimal generalized entropy.[1]

For static states (or those at moments of time-reflection symmetry), the above prescription simplifies considerably; the entanglement wedge is simply the *smallest generalized entropy* bulk region contained in the static (or time-reflection symmetric) slice. While this simpler restricted prescription can't teach us about the detailed dynamics of information flow in quantum gravity,[2] it is sufficient to capture many other aspects of the information-theoretic structure of AdS/CFT. It will be the sole focus of the present paper. To emphasize this, from now on we will use the expression *quantum minimal surface* (QMS), rather than minimal QES.

The QMS prescription is a signature of a broader and deeper relationship between bulk and boundary information known as "entanglement wedge reconstruction." Entanglement wedge reconstruction is easiest to understand (and was first understood) when the entanglement wedge **b** of the boundary region *B* is the same for all states of interest. This will be (approximately) true so long as one restricts interest to small "code spaces" of bulk states with a fixed geometry in limits where the bulk entropy term can be ignored in (1) when finding the entanglement wedge. In such situations, (1) implies an approximate equality between the relative entropy [17,18] between any two boundary states on region *B* and the corresponding bulk relative entropy between the dual states on region **b**. In turn, this implies that every bulk operator $O_{\mathbf{b}}$ acting on **b** has a "boundary reconstruction" $O_B$ acting only on *B* that acts faithfully on the code space [4–6]. We say that such a reconstruction is *state independent* because, in contrast to the more general case discussed below, the same boundary representation of a bulk operator can be used for the entire code space of allowed bulk states.

In fact, (1) implies something even stronger in these settings: *complementary* state-independent recovery. The same equality between relative entropies that ensured *B* could reconstruct **b** also implies that a bulk operator acting on the complementary bulk region **b̄** can be state-independently reconstructed on the complementary boundary region *B̄*. In other words, so long as **b** is the same for all states of interest, the single condition (1) implies reconstruction of both **b** in *B* and **b̄** in *B̄*.

While this is already a very powerful result, in [1] Harlow proved something perhaps even more interesting: a converse statement is also true. Complementary state-independent recovery is sufficient to show that (1) holds – for *some* definition of "area" *A*. That is, for *any* complementary quantum error correcting (QEC) code – even one that a priori has nothing to do with holography – if the physical subsystem $\mathcal{H}_B$ can state-independently reconstruct a logical factor $\mathcal{H}_{\mathbf{b}}$ and likewise $\mathcal{H}_{\overline{B}}$ can reconstruct $\mathcal{H}_{\overline{\mathbf{b}}}$, then

$$S(B)_{V|\psi\rangle} = A + S(\mathbf{b})_{|\psi\rangle} \,, \tag{2}$$

for some constant "area" *A* that depends on the particular QEC code.[3] We can therefore *define* an entanglement wedge for *B* in any complementary QEC code as the reconstructible factor of the code space and obtain an associated holographic entropy formula. In this picture, rather

---

[1]Let us emphasize the proviso: this is only true for *appropriate* holographic states. As shown in [11] (see also [15]), there are some semiclassical holographic states – i.e. simple states of bulk quantum matter on fixed semiclassical geometric backgrounds – for which (1) does not apply for *any* bulk region **b**, even as a leading order semiclassical approximation. In such cases, we say that the entanglement wedge of *B* is not well-defined.

[2]For example, it doesn't know about the scrambling time delay before information escapes an evaporating black hole after the Page time [9,16].

[3]In the analogy with area, we are implicitly using units where $4G = 1$.

than the area term in (1) being an external input from quantum gravity, it simply quantifies a particular source of entanglement that is present in all (complementary) QEC codes.

In this sense, any QEC code has the beginnings of an emergent "bulk geometry" within it, albeit by "geometry" we so far mean the area of a single surface. In turn, this geometry gives us a clear justification of, and information-theoretic motivation for, the existence of a formula like (1), which is a result that has otherwise only been derived using the somewhat mysterious tool of gravitational path integrals.

That said, the framework of [1] is not yet the full story. It explains only the special case of a code space with a single, state-independent entanglement wedge.[4] In general, different states within the same large code space may have different entanglement wedges. See Figure 1 for an example. Hence the entanglement wedge cannot generally be defined as the region that is reconstructible in a state-independent way. That region, called the *reconstructible wedge* in [21], depends only on the code space, and not on the state itself, and is in fact equal to the intersection of the entanglement wedges over all states in the code space, pure and mixed [4, 6, 21]. State-independent complementary reconstruction is not possible when different states in the code space have different entanglement wedges; the assumptions of the theorems in [1] do not apply.

Can Harlow's arguments be generalized to these settings? Does bulk reconstruction still define the entanglement wedge? And does geometry still emerge from information? Answering such questions will be the primary goal of this paper. It turns out that, while state-independent complementary reconstruction is no longer always possible, a weaker version of complementary reconstruction *is* equivalent to (1). The new ingredient is that the reconstruction may need to be *state-specific*, with different reconstructions used for different states within the code space. Remarkably, not only does one still find an emergent bulk geometry – and a holographic entropy prescription – for any code with state-specific complementary reconstruction, but the emergent geometries can be much richer than those found before. Rather than a single surface with a single area, we define areas that bound *any* set of "bulk" subsystems.[5] And rather than (1) holding for a fixed subsystem $\mathcal{H}_\mathbf{b}$, the subsystem $\mathcal{H}_\mathbf{b}$ is determined by minimizing generalized entropy over all sets of subsystems – a true quantum minimal surface prescription.

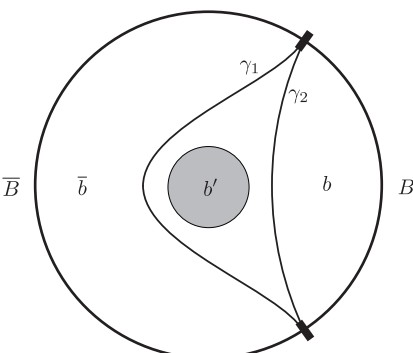

Figure 1: An AdS/CFT setup without complementary state-independent recovery. Between the two candidate extremal surfaces lies a black hole with horizon area much greater than the difference in area between the two surfaces, $A_{\mathrm{BH}} \gg A_{\gamma_2} - A_{\gamma_1}$. Neither $B$ nor $\overline{B}$ can reconstruct operators in $b'$ in a state-independent way.

---

[4]See [7, 8, 19, 20] for generalizations of [1] that nonetheless still use the same assumption of a single, state-independent entanglement wedge.

[5]Of course, even this is still a long way from having a smooth bulk metric!

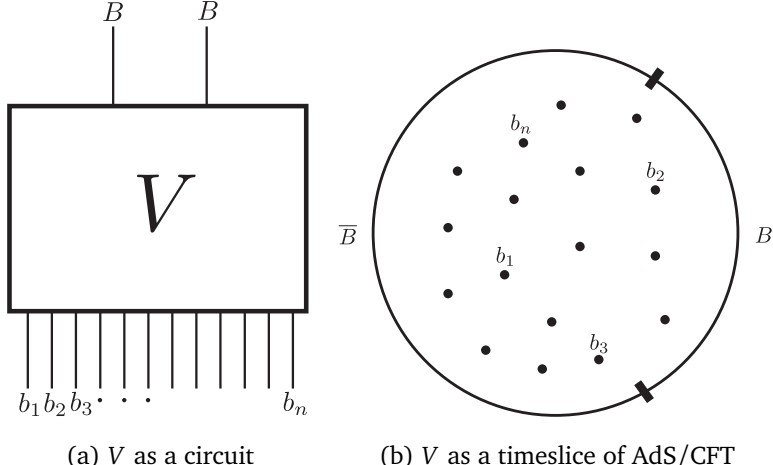

(a) $V$ as a circuit      (b) $V$ as a timeslice of AdS/CFT

Figure 2: Two illustrations of the quantum codes used throughout this paper. (a) A linear map $V : \otimes_i \mathcal{H}_{b_i} \to \mathcal{H}_B \otimes \mathcal{H}_{\overline{B}}$. (b) This linear map encompasses situations in holography. Each input leg corresponds to a bulk point or subregion, while the output legs correspond to boundary subregions. The map $V$ is implicit.

## 1.2 This paper

To explain our generalization of Harlow's results we must first define both "areas" and "entanglement wedges" for arbitrary quantum codes.

We define a quantum code as an isometry $V : \otimes_{i=1}^n \mathcal{H}_{b_i} \to \mathcal{H}_B \otimes \mathcal{H}_{\overline{B}}$, as in figure 2a. In a holographic context, the input subsystems $\mathcal{H}_{b_i}$ are each associated to different local bulk regions, as in figure 2b.

Given any subset $\mathbf{b} := \{b_{i_1}, b_{i_2}...\}$ of input legs, we define an "area" $A_B(\mathbf{b})$ as follows. (See Section 2 for more details.) For every isometry $V$ there is an associated special state, called the Choi-Jamiolkowski state, defined by acting with $V$ on half of a maximally-entangled state

$$|\text{CJ}\rangle_{B\overline{B}r_1...r_n} := V |\text{MAX}\rangle_{b_1...b_n r_1...r_n}, \tag{3}$$

where $\mathcal{H}_{r_i} \cong \mathcal{H}_{b_i}^*$, and $|\text{MAX}\rangle_{b_1...b_n r_1...r_n}$ is the canonical maximally entangled state on $\mathcal{H}_{b_1} \otimes ... \mathcal{H}_{b_n} \otimes \mathcal{H}_{r_1} \otimes ... \mathcal{H}_{r_n}$.[6] See Figure 3. This state consists of a product of maximally entangled states on each pair $\mathcal{H}_{b_i} \otimes \mathcal{H}_{r_i}$. In particular, $\mathcal{H}_{\mathbf{r}} = \mathcal{H}_{r_{i_1}} \otimes \mathcal{H}_{r_{i_2}}...$ purifies $\mathcal{H}_{\mathbf{b}} = \mathcal{H}_{b_{i_1}} \otimes \mathcal{H}_{b_{i_2}}...$. The area $A_B(\mathbf{b})$ is defined by

$$A_B(\mathbf{b}) := S(B\,\mathbf{r})_{|\text{CJ}\rangle}. \tag{4}$$

This definition works for any collection of input subsystems $\mathbf{b}$ for any QEC code $V$, and we will argue in Section 2 that it is "functionally unique" as a definition of area for codes that obey a QMS prescription.[7] The explicit dependence of (4) on $\mathcal{H}_B$ should be understood as enforcing the homology constraint on the surface bounding region $\mathbf{b}$.

Next we need to define the entanglement wedge $\text{EW}_B(|\psi\rangle)$ of a boundary region $B$ for bulk state $|\psi\rangle$. One approach would simply be to always define $\text{EW}_B(|\psi\rangle) = \mathbf{b}$ as the collection of input subsystems $\mathbf{b}$ that minimizes $A_B(\mathbf{b}) + S(\mathbf{b})_{|\psi\rangle}$. However this would be naive: for most

---

[6]Given an arbitrary orthonormal basis $\{|i\rangle\}$ for a Hilbert space $\mathcal{H}$, the canonical maximally entangled state $|\text{MAX}\rangle \in \mathcal{H} \otimes \mathcal{H}^*$ is defined to be $\sum_i |i\rangle |i\rangle^* / \sqrt{d}$. It is easy to verify that this is independent of the choice of basis $\{|i\rangle\}$.

[7]By this we mean that any other definition of area – for example the geometric area in holographic codes, and the log bond dimension in tensor network codes – will agree with (4) on any surface that is quantum minimal for some state in the code space. As we emphasize in Section 2, the different definitions may not agree for surfaces that are never quantum minimal; however in that case (4) still gives a lower bound on any other definition of area.

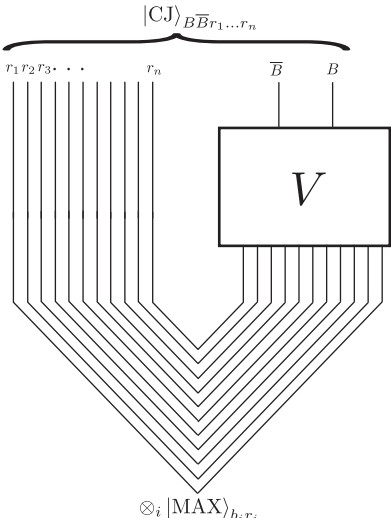

Figure 3: The Choi-Jamiolkowski state $|\mathrm{CJ}\rangle$. Angled lines represent maximally entangled states, with one half of the maximally entangled state input into $V$.

codes $V$ (and even for many states in holographic codes [11]) the quantum minimal surface defined in this way has no information-theoretic significance.[8] Instead we want to define the entanglement wedge (when it exists) to be the collection of bulk subsystems that are reconstructible from $\mathcal{H}_B$ in an appropriate sense. We will then show *as a consequence* that the entanglement wedge must always be quantum minimal.

So what is the sense in which the entanglement wedge needs to be reconstructible from $\mathcal{H}_B$? We describe a precise answer to this question in Section 3 where we also explain how that answer encompasses and generalizes previous definitions of quantum error correction and bulk reconstruction. The reconstruction needs to state-specific, or else the entanglement wedge would need to be the same for all states in the code space, as in the theorems in [1]. We say that a bulk unitary $U_{\mathbf{b}}$ can be reconstructed by the boundary unitary $U_B$ for the specific state $|\psi\rangle$ if

$$U_B V |\psi\rangle = V U_{\mathbf{b}} |\psi\rangle . \tag{5}$$

Note that it is important here that the boundary reconstruction $U_B$ is unitary – otherwise the reconstructed operator would be able to change the reduced state on $\mathcal{H}_{\overline{B}}$ despite nominally only acting on $\mathcal{H}_B$. Indeed, so long as the dimension $d_B$ of $\mathcal{H}_B$ is at least as large as the dimension $d_{\overline{B}}$ of $\mathcal{H}_{\overline{B}}$, then a generic state $|\psi\rangle \in \mathcal{H}_B \otimes \mathcal{H}_{\overline{B}}$ can be mapped to any other state by a non-unitary operator acting only on $\mathcal{H}_B$.[9]

However, it would be too strong to demand that $B$ can do state-specific reconstruction of *all* unitaries $U_{\mathbf{b}}$ that act on its entanglement wedge $\mathbf{b}$. By acting with arbitrary unitaries within $\mathbf{b}$ one can change the entanglement structure of the state $|\psi\rangle$, potentially changing the location of the quantum minimal surface and therefore the entanglement wedge; see Figure 4. This cannot ever be achieved by acting only with a unitary on $\mathcal{H}_B$, since it will change the entropy $S(\overline{B})_{|\psi\rangle}$. To avoid this issue, we should only insist that $B$ be able to reconstruct unitaries $U_{\mathbf{b}}$ that don't change the entanglement structure of the bulk state. Such operators are *product unitaries* – unitaries that can be written as a product of local unitaries $U_{\mathbf{b}} = U_{b_{i_1}} \otimes U_{b_{i_2}} \ldots$

---

[8]See Appendix B.1 and also [11] for examples.

[9]The celebrated Reeh-Schlieder Theorem [22] is a similar statement about continuum quantum field theory.

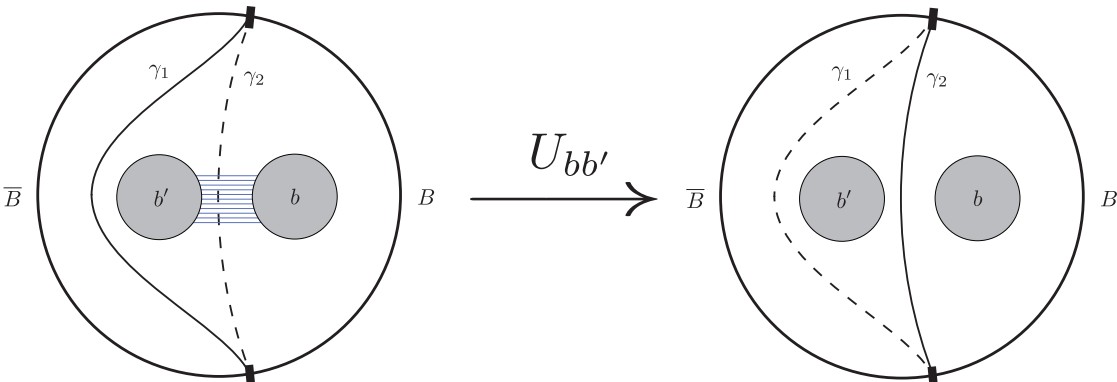

Figure 4: An AdS/CFT setup in which a unitary acting within an entanglement wedge changes the location of the quantum minimal surface. On the left, we have two black holes in an entangled pure state. Because of the large amount of entanglement, $\gamma_1$ is quantum minimal and therefore both black holes are inside the entanglement wedge of $B$. On the right, we have the situation after acting on both black holes with a unitary $U_{bb'}$ that maps the original state to a factorized pure state. Now $\gamma_2$ is quantum minimal. This unitary has changed the entropies $S(B)$ and $S(\overline{B})$ and therefore cannot be represented by a unitary on $B$.

We therefore define the entanglement wedge of $B$ for a pure state $|\psi\rangle_{b_1...b_n}$ as the bulk region **b** for which state-specific complementary reconstruction of product unitaries is possible. In other words, for any product unitaries $U_{\mathbf{b}}$ and $U'_{\overline{\mathbf{b}}}$, there need to exist boundary reconstructions $U_B$ and $U'_{\overline{B}}$ respectively such that

$$U_B U'_{\overline{B}} V |\psi\rangle = V U_{\mathbf{b}} U'_{\overline{\mathbf{b}}} |\psi\rangle \,. \tag{6}$$

What about the entanglement wedge of $B$ for mixed bulk states? A mixed state will generally not have complementary entanglement wedges for $B$ and $\overline{B}$. However, we can use the standard trick of purifying the mixed state with an auxiliary reference system $\mathcal{H}_{\overline{R}}$ to construct the pure state $|\Psi\rangle_{b_1...b_n\overline{R}}$. We have complementary state-specific reconstruction if, for any product unitaries $U_{\mathbf{b}}$ and $U'_{\overline{\mathbf{b}}}$, there exist boundary reconstructions $U_B$ and $U'_{\overline{BR}}$. In other words, the reconstruction of $U_{\overline{\mathbf{b}}}$ is allowed to act not only on the environment $\mathcal{H}_{\overline{B}}$, but also on the reference system $\mathcal{H}_{\overline{R}}$. In the interests of full generality, we can also add a reference system $\mathcal{H}_R$ to the $\mathcal{H}_B$ side:[10] in this case, we say that the entanglement wedge of the region $BR$ for the state $|\Psi\rangle_{b_1...b_n R\overline{R}}$ is **b**, if, for any product unitaries $U_{\mathbf{b}}$ and $U'_{\overline{\mathbf{b}}}$, there exist boundary reconstructions $U_{BR}$ and $U'_{\overline{BR}}$ such that

$$U_{BR} U'_{\overline{BR}} V |\Psi\rangle = V U_{\mathbf{b}} U_{\overline{\mathbf{b}}} |\Psi\rangle \,. \tag{7}$$

The central result of this paper will be to show that this definition of the entanglement wedge, based on its reconstruction properties, is equivalent to the QMS prescription being valid for the state $|\Psi\rangle$, together with all states related to $|\Psi\rangle$ by a product of local unitaries. Specifically, it is equivalent to the prescription

$$S(BR)_{VU_{\mathbf{b}}U'_{\overline{\mathbf{b}}}|\Psi\rangle} = A_B(\mathbf{b}) + S(\mathbf{b}R)_{U_{\mathbf{b}}U'_{\overline{\mathbf{b}}}|\Psi\rangle} \,, \tag{8}$$

---

[10]Mathematically, this generalization is essentially trivial. However, physically it is very important: studying the entanglement wedge of auxiliary nonholographic quantum systems is at the heart of recent progress on the black hole information problem [9, 10].

holding for all product unitaries $U_{\mathbf{b}} U'_{\bar{\mathbf{b}}}$.[11] This prescription is the obvious generalization of (1) to the entropy $S(BR)$; it was first introduced in gravity in [6].

The results can be formalized in the following theorem, which we prove in Section 4.

**Theorem 4.2.** *Let* $V : \mathcal{H}_{\mathrm{code}} \cong \otimes_i \mathcal{H}_{b_i} \to \mathcal{H}_B \otimes \mathcal{H}_{\overline{B}}$ *be an isometry, with* $\mathbf{b} = \{b_{i_1}, b_{i_2}...\}$ *a subset of input legs, and* $\bar{\mathbf{b}}$ *its complement. Finally, let* $|\Psi\rangle \in \mathcal{H}_{\mathrm{code}} \otimes \mathcal{H}_R \otimes \mathcal{H}_{\overline{R}}$ *be a fixed, arbitrary state with* $\mathcal{H}_R, \mathcal{H}_{\overline{R}}$ *reference systems of arbitrary dimension.*

*Then the following two conditions are equivalent:*

1. (Complementary Recovery) *For all product unitaries* $U_{\mathbf{b}}$ *and* $U'_{\bar{\mathbf{b}}}$*, there exist unitary operators* $U_{BR}$ *and* $U'_{\overline{B}\overline{R}}$ *respectively such that*

$$U_{BR} U'_{\overline{B}\overline{R}} V |\Psi\rangle = V U_{\mathbf{b}} U'_{\bar{\mathbf{b}}} |\Psi\rangle . \tag{9}$$

2. (Holographic Entropy Prescription) *For all product unitaries* $U_{\mathbf{b}}$ *and* $U'_{\bar{\mathbf{b}}}$*,*

$$S(BR)_{V U_{\mathbf{b}} U'_{\bar{\mathbf{b}}} |\Psi\rangle} = A_B(\mathbf{b}) + S(\mathbf{b}R)_{U_{\mathbf{b}} U'_{\bar{\mathbf{b}}} |\Psi\rangle} . \tag{10}$$

*Moreover, both statements imply:*

3. (One-shot Minimality) *For all* $\bar{\mathbf{b}}' \subseteq \bar{\mathbf{b}}$ *and* $\mathbf{b}' \subseteq \mathbf{b}$*,*

$$H_{\mathrm{min}}(\bar{\mathbf{b}}'|\mathbf{b}R)_{|\Psi\rangle} \geq A_B(\mathbf{b}) - A_B(\mathbf{b} \cup \bar{\mathbf{b}}') , \tag{11}$$

$$H_{\mathrm{min}}(\mathbf{b}'|\bar{\mathbf{b}}\overline{R})_{|\Psi\rangle} \geq A_{\overline{B}}(\bar{\mathbf{b}}) - A_{\overline{B}}(\mathbf{b}' \cup \bar{\mathbf{b}}) . \tag{12}$$

*This in turn implies:*

4. (Minimality) *For all* $\mathbf{b}' \subseteq \mathbf{b} \cup \bar{\mathbf{b}}$*,*

$$A_B(\mathbf{b}') + S(\mathbf{b}'R)_{|\Psi\rangle} \geq A_B(\mathbf{b}) + S(\mathbf{b}R)_{|\Psi\rangle} . \tag{13}$$

Since Condition 3 (One-shot minimality) here is probably less familiar or intuitive to readers than the other three conditions, let us take a moment to explain its significance. If the left hand sides of (11) and (12) were replaced by the conditional von Neumann entropies $S(\bar{\mathbf{b}}'|\mathbf{b}R)_{|\Psi\rangle}$ and $S(\mathbf{b}'|\bar{\mathbf{b}}\overline{R})_{|\Psi\rangle}$, then Conditions 3 and 4 would be equivalent. Instead, however, Condition 3 features an alternative entropy measure – the conditional min-entropy $H_{\mathrm{min}}(A|B)$. As a result, it is a strictly stronger condition. While, for any state $|\Psi\rangle$, there is always *some* set of subsystems $\mathbf{b}$ satisfying Condition 4, there is not always any $\mathbf{b}$ satisfying Condition 3.

In [11], we pointed out an important relationship between one-shot quantum information theory and holography. The upshot is that the generalized entropy of the minimal QES is *not* equal to the boundary entropy for all quantum states. In other words, Condition 4 does not imply Conditions 1 and 2 even in holographic codes. So when is the holographic entropy prescription valid for holographic states? In [11], we provided the answer: complementary state-specific reconstruction is possible in AdS/CFT, and the QES prescription is valid, whenever Condition 3 holds. The same is true for toy models of AdS/CFT such as random tensor networks.

---

[11]It turns out to be important to demand that the QMS prescription hold for all states related to $|\Psi\rangle$ by product unitaries, rather than just for the state $|\Psi\rangle$ itself. In general codes, regions that just satisfy (1) for a single state $|\Psi\rangle$ need not have any particular reconstruction relationship to $B$, see Appendix B.1. To really talk about a holographic entropy prescription as we know it from gravity, we need (1) to be satisfied not just by a single state, but by all states with a particular entanglement structure, i.e. related by product unitaries.

For general quantum codes, Condition 3 of course cannot be sufficient to derive Condition 1 and 2, since it does not depend on the isometry $V$ at all. However, it is always true for any quantum code $V$ and state $|\Psi\rangle$ satisfying the first two conditions, and as a consequence so is Condition 4.

In Section 5 we generalize Theorem 4.2 in two important ways. (Since the proof of this more general theorem is somewhat technical, we postpone it to Appendix C.) Firstly, we allow small errors in reconstruction and in the QMS prescription. This is crucial because the known nontrivial examples of state-specific product unitary reconstruction (including AdS/CFT itself) are inherently approximate. Indeed, in particularly simple versions of state-specific codes known as zero-bit codes, one can prove that exact state-specific reconstruction implies state-independent reconstruction, even though approximate state-specific reconstruction does not.

Secondly, we generalize Theorem 4.2 to allow linear maps $V$ that are not isometries. This allows us to understand code spaces in the interior of black holes with entropy larger than the Bekenstein-Hawking entropy. Such code spaces are completely irreconcilable with the usual framework of quantum error correction, but seem to arise naturally in semiclassical gravity. In particular, they play a vital role in recent progress on the black hole information problem [9,10]. We therefore view state-specific codes as crucial for a QEC-based understanding of the interior of black holes.

In Section 6 we summarize our results and discuss important consequences and potential future directions.

Finally, in Appendix A, we provide self-contained proofs of various technical results previously stated in the main body of the paper. These include a number of results that are well known in the quantum information community, but are included for the convenience of the reader because they may be less familiar to people with a background in high-energy physics. In Appendix B, we consider various seemingly plausible variations on Theorem 4.2, and construct counterexamples to each of them. These include replacing product unitaries by local unitaries, and extending the minimality statement in Condition 4 to more general sets of subalgebras.

## 1.3 Notation

We use lower-case letters to label bulk Hilbert spaces ($\mathcal{H}_{b_i}, \mathcal{H}_{r_i}$ etc.) and capital letters to label boundary Hilbert spaces. Bold letters denote sets, for example $\mathbf{b} = \{b_{i_1}, b_{i_2}, \dots\}$, and $\mathbf{U}(d)$ is the group of $d \times d$ unitary matrices, while $U \in \mathbf{U}(d)$ is a single unitary. For any set $\mathbf{b}$ of Hilbert space labels, we define $\mathcal{H}_{\mathbf{b}} = \otimes_{b_i \in \mathbf{b}} \mathcal{H}_{b_i}$. Finally, we use bars to denote the complement of a subsystem or set of subsystems, e.g. $\bar{\mathbf{b}} = \{b_i : 1 \le i \le n, \ b_i \notin \mathbf{b}\}$.

## 2 Areas in general quantum codes

Before we can prove a QMS prescription for arbitrary quantum codes, we need to define what we mean by "area" in such codes. After all, unlike gravity, quantum codes do not in general come pre-equipped with a notion of geometry! The task of this section will be to motivate such a definition. Our definition will be valid for any quantum code, by which we mean a quantum channel in the Stinespring picture. Said precisely, a quantum code is an isometry

$$V : \mathcal{H}_{\text{code}} \cong \otimes_i \mathcal{H}_{b_i} \to \mathcal{H}_B \otimes \mathcal{H}_{\bar{B}} \tag{14}$$

mapping a code space $\mathcal{H}_{\text{code}}$, made up of a collection of subsystems $\mathcal{H}_{b_i}$, to an output Hilbert space $\mathcal{H}_B$ together with an environment $\mathcal{H}_{\bar{B}}$.[12] The area $A_B(\mathbf{b})$ will depend only on the code

---

[12]In conventional applications of quantum error correction, it is helpful to distinguish the encoding map, which is chosen by the experimenter for its nice properties, from the noisy channel, in which the "errors" occur. Mathe-

$V$, the output Hilbert space $\mathcal{H}_B$, and the choice of input subsystems $\mathbf{b}$. It does not depend on any choice of input state $|\Psi\rangle$. Unlike previous definitions of area in QEC codes, the code will not need to have any particular error correcting properties.

However, the definition of "area" we introduce will not be completely unique. Other distinct definitions can also lead to a QMS prescription, such as the natural 'areas' in tensor networks and gravity. And, for some surfaces, our definition may not necessarily agree with those other definitions.

That said, the differences are not important; they correspond to essentially trivial redefinitions of the same QMS prescription. Consider, for example, a surface with very large area, such that it can never be quantum minimal for *any* input state. We can clearly redefine the area of this surface, increasing its size, without affecting the validity of the QMS prescription. Of course, by doing so, we have not *meaningfully* changed the QMS prescription that we are using, since the surface was never relevant to that prescription anyway!

Indeed, in Section 2.2, we argue that our definition of area is functionally unique in the sense hinted at above. We prove that any alternative definition of area consistent with a QMS prescription must agree with our definition on all surfaces that are ever quantum minimal according to that alternative prescription. Moreover, on *any* surface – including surfaces that are never quantum minimal – our definition of area is a lower bound on any alternative definition.

## 2.1 An area definition for arbitrary codes

We begin by stating our definition of the area of a set of subsystems, and then provide some comments on and motivation for this definition.

**Definition 2.1.** Let $V : \mathcal{H}_{\text{code}} \cong \otimes_i \mathcal{H}_{b_i} \to \mathcal{H}_B \otimes \mathcal{H}_{\overline{B}}$ be an isometry and let

$$|\text{CJ}\rangle_{B\overline{B}r_1 \ldots r_n} = V |\text{MAX}\rangle_{b_1 \ldots b_n r_1 \ldots r_n}$$

be the associated Choi-Jamiolkowski state, with $|\text{MAX}\rangle_{b_1 \ldots b_n r_1 \ldots r_n}$ the canonical maximally entangled state for $\mathcal{H}_{r_i} \cong \mathcal{H}_{b_i}^*$. Let $\mathbf{b} = \{b_{i_1}, b_{i_2} \ldots\}$ be a subset of input legs with $\mathcal{H}_{\mathbf{b}} \cong \mathcal{H}_{b_{i_1}} \otimes \mathcal{H}_{b_{i_2}} \ldots$ maximally entangled with $\mathcal{H}_{\mathbf{r}} \cong \mathcal{H}_{r_{i_1}} \otimes \mathcal{H}_{r_{i_2}} \ldots$ in $|\text{MAX}\rangle_{b_1 \ldots b_n r_1 \ldots r_n}$. We then define the area $A_B(\mathbf{b}) \in \mathbb{R}$ as

$$A_B(\mathbf{b}) = S(B\mathbf{r})_{|\text{CJ}\rangle}, \tag{15}$$

where $S(B\,\mathbf{r})_{|\text{CJ}\rangle}$ is the entanglement entropy of $\mathcal{H}_B \otimes \mathcal{H}_{\mathbf{r}}$ for the state $|\text{CJ}\rangle_{B\overline{B}r_1 \ldots r_n}$.

In other words the function $A_B : 2^{\{b_1 \ldots b_n\}} \to \mathbb{R}$ maps a subset $\mathbf{b}$ of input legs (i.e. an element of the power set $2^{\{b_1 \ldots b_n\}}$) to a real-valued "area" of the "surface" bounding the subset $\mathbf{b}$. Note that the function $A_B$ depends explicitly not only on the isometry $V$ but also on the output subsystem $\mathcal{H}_B$. The dependence on $\mathcal{H}_B$ represents the homology constraint present in the QMS prescription: two surfaces bounding the same bulk region $\mathbf{b}$ will have different areas if they are homologous to two different boundary regions $B$.[13]

**Remark 2.2.** Given any subset $\mathbf{b} \subseteq \{b_1 \ldots b_n\}$, the area $A_B(\mathbf{b}) = S(B\,\mathbf{r})_{|\text{CJ}\rangle}$ and the complementary area $A_{\overline{B}}(\overline{\mathbf{b}}) = S(\overline{B}\,\overline{\mathbf{r}})_{|\text{CJ}\rangle}$ are equal.

---

matically the properties of the code depend only on the composition of these two maps, and in holography there is no distinction between them. (Holographic codes are often interpreted as erasure codes where $V$ is the encoding map and the noisy channel consists only of the partial trace, but this separation is essentially arbitrary.) We shall therefore refer only to a single isometry $V$ mapping the logical state to the final physical state of system and environment.

[13] One way to see this is to consider the area of the surface bounding the empty set. $A_B(\varnothing)$ is not zero, and so the surface is not trivial; it is better interpreted as the surface homologous to $B$ but excluding the entire bulk.

**Remark 2.3.** Definition 2.1 is a generalization of the area defined for codes in [1]. Suppose that we have an isometry $V : \mathcal{H}_{b_1} \otimes \mathcal{H}_{b_2} \to \mathcal{H}_B \otimes \mathcal{H}_{\overline{B}}$ such that

$$V \left|\psi\right\rangle_{b_1 b_2} = V_1^{b_1 c_1 \to B} V_2^{b_2 c_2 \to \overline{B}} \left|\psi\right\rangle_{b_1 b_2} \left|\chi\right\rangle_{c_1 c_2} , \tag{16}$$

for any state $\left|\psi\right\rangle$ where $\left|\chi\right\rangle \in \mathcal{H}_{c_1} \otimes \mathcal{H}_{c_2}$ is a fixed state and $V_1^{b_1 c_1 \to B} : \mathcal{H}_{b_1} \otimes \mathcal{H}_{c_1} \to \mathcal{H}_B$ and $V_2^{b_2 c_2 \to \overline{B}} : \mathcal{H}_{b_2} \otimes \mathcal{H}_{c_2} \to \mathcal{H}_{\overline{B}}$ are isometries. In other words, suppose $V$ obeys the structure theorem of [1] for exact QEC codes with complementary recovery. Then it can be seen immediately that

$$A_B(b_1) = S(Br_1)_{\left|\text{CJ}\right\rangle} = S(c_1)_{\left|\chi\right\rangle} . \tag{17}$$

The right hand side of (17) is exactly the definition of area used in [1]. Definition 2.1 therefore agrees with the definition of area from [1] whenever both are defined. As emphasized above, however, Definition 2.1 is much more general, applicable to arbitrary input subsystems for arbitrary quantum codes. In contrast, the definition of area used in [1] relies crucially on the structure theorem (16) for complementary QEC codes, and so cannot be applied outside that context.

## 2.2 Comparisons with other definitions of area

While we do not know of any natural alternative definition of area for arbitrary subsystems in arbitrary quantum channels, there do exist standard notions of area for particular, special channels, such as tensor networks and of course gravitational systems themselves. If we are to relate our QMS prescription to the already known prescriptions for these special classes of channels, it is important to be able to compare the different definitions of area.

To give a sharp example, tensor networks codes are a particular class of quantum code where the isometry $V$ can be replaced by a network of smaller tensors contracted together. See Figure 5. For tensor network codes, the "area" of a set of input subsystems is commonly defined as the logarithm of the dimension of cut in-plane legs for a surface surrounding those input legs. Generically, random tensor networks (RTNs) [23] satisfy a QMS prescription with respect to this area. So what is the relationship between that area and the one from Definition 2.1? Can the two QMS prescriptions ever lead to different entanglement wedges for the same state?

We will give precise answers to these questions below in the form of two theorems and a corollary. The basic summary is that any alternative prescription will agree with Definition 2.1 on the location of the entanglement wedge and on the value of the area for any surfaces that are potentially quantum minimal according to the alternative prescription. For other surfaces, our definition of area lower bounds all possible alternative definitions. The area in Definition 2.1 is therefore "functionally unique."

In order to make more precise statements, we will need to first introduce some preliminary definitions.

**Definition 2.4.** Consider an isometry $V : \mathcal{H}_{\text{code}} \cong \otimes_i \mathcal{H}_{b_i} \to \mathcal{H}_B \otimes \mathcal{H}_{\overline{B}}$. An "area" function $A'_B : 2^{\{b_1 \dots b_n\}} \to \mathbb{R}$ is said to *lead to a quantum minimal surface (QMS) prescription for a state* $\left|\Psi\right\rangle \in \mathcal{H}_{\text{code}} \otimes \mathcal{H}_R \otimes \mathcal{H}_{\overline{R}}$ *and region BR* if there exists a set of input subsystems $\mathbf{b} \subseteq \{b_1 \dots b_n\}$ such that for all product unitaries $U = U_{b_1} \otimes U_{b_2} \dots$,

$$S(BR)_{VU\left|\Psi\right\rangle} = A'_B(\mathbf{b}) + S(\mathbf{b}R)_{U\left|\Psi\right\rangle} , \tag{18}$$

and moreover that

$$A'_B(\mathbf{b}') + S(\mathbf{b}'R)_{\left|\Psi\right\rangle} \geq A'_B(\mathbf{b}) + S(\mathbf{b}R)_{\left|\Psi\right\rangle} , \tag{19}$$

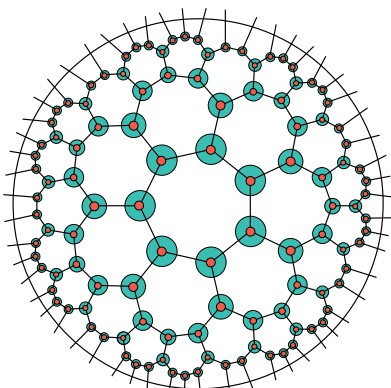

Figure 5: An example tensor network. The blue circles each represent tensors, four-partite states on the union of the small red circle and three black legs touching that blue circle. Each tensor can be viewed as a map from a state on the red circle to a state on the black legs. A solid black line between tensors represents postselection of the state on one leg from each tensor onto the state $|\text{MAX}\rangle$. The entire tensor network forms a map from states on the union of the small red circles to the union of the boundary legs, which are those legs crossing the outer circle.

for all other sets of subsystems $\mathbf{b}' \subseteq \{b_1 \dots b_n\}$. We then say that $\mathbf{b}$ is an entanglement wedge $\text{EW}'_{BR}(|\Psi\rangle)$ of $BR$ for the state $|\Psi\rangle$.

**Remark 2.5.** As a notational convention, we reserve $A_B$ to denote the use of area as defined in Definition 2.1. Alternative area functions are therefore denoted by $A'_B$. Similarly, we reserve EW for the entanglement wedge defined by (7), which Theorem 4.2 shows is equivalent to the entanglement wedge given in Definition 2.4 for $A_B$.

In essence, Definition 2.4 says that an area function $A'_B$ leads to a QMS prescription for the state $|\Psi\rangle$ if $|\Psi\rangle$ satisfies Conditions 2 and 4 of Theorem 4.2, with the area $A_B$ from Definition 2.1 replaced by the alternative area function $A'_B$.

**Remark 2.6.** For a given area function $A'_B$, quantum code $V$, and state $|\Psi\rangle$, there can exist more than one entanglement wedge $EW'_{BR}(|\Psi\rangle)$ if there exists more than one subset $\mathbf{b} \in \{b_1 \dots b_n\}$ with the same generalized entropy $A'_B(\mathbf{b}) + S(\mathbf{b}R)_{|\Psi\rangle}$.

For example, consider the area function $A_B$ from Definition 2.1, with a quantum code with $V$ the identity map after identifying $\mathcal{H}_{b_1} \cong \mathcal{H}_B$ and $\mathcal{H}_{b_2} \cong \mathcal{H}_{\overline{B}}$, with $d_{b_1} > d_{b_2}$, and with a state $|\psi\rangle \in \mathcal{H}_{b_1} \otimes \mathcal{H}_{b_2}$ that is maximally entangled between the two subsystems. For any product unitary $U = U_{b_1} \otimes U_{b_2}$, we have

$$S(B)_{V|\psi\rangle} = \log d_{b_2} = A_B(b_1 b_2) = A_B(b_1 b_2) + S(b_1 b_2)_{|\psi\rangle} . \tag{20}$$

However we also have

$$S(B)_{V|\psi\rangle} = S(b_1)_{|\psi\rangle} = A_B(b_1) + S(b_1)_{|\psi\rangle} . \tag{21}$$

It is easy to verify that $b_2$ and $\varnothing$ have larger generalized entropy. Hence both $b_1$ and $b_1 b_2$ are entanglement wedges $EW_B(|\psi\rangle)$ of the region $B$ for the state $|\psi\rangle$.

We also want to define a notion of an area function $A'_B$ leading to a QMS prescription for an isometry $V$ *in general*, rather than just doing so for one specific state $|\Psi\rangle$. A naive approach to doing so would be to demand that all states $|\Psi\rangle$ obey a QMS prescription. However, this would be too strong. As shown in [11], even the actual, geometric area of surfaces in AdS/CFT does

not lead to a QMS prescription that works for all states.[14] Instead, we use the area function $A_B$ from Definition 2.1 to define a reasonable set of states for which the QMS prescription ought to apply.

**Definition 2.7.** *We say that an area function $A'_B$ leads to a quantum minimal surface (QMS) prescription for the isometry $V$ if $A'_B$ leads to a QMS prescription for all states $|\Psi\rangle$ and regions $BR$ for which the area function $A_B$ from Definition 2.1 leads to a QMS prescription.*

With all the needed definitions in hand, we are finally ready to state the main technical results of this section. We first show that the area $A_B$ introduced in Definition 2.1 is a lower bound on any other definition of area that satisfies a QMS prescription.

**Theorem 2.8.** *If an area function $A'_B$ satisfies a QMS prescription for the isometry $V$, then*

$$A'_B(\mathbf{b}) \geq A_B(\mathbf{b}), \tag{22}$$

*for all collections of inputs $\mathbf{b} \subseteq \{b_1 \dots b_n\}$.*

*Proof.* See Appendix A.2. ∎

Finally, we show that any alternative area function $A'_B$ that leads to a QMS prescription for the isometry $V$ must agree with $A_B$ for all subsets $\mathbf{b}$ that can ever be an entanglement wedge.

**Theorem 2.9.** *Let $A'_B$ lead to a QMS prescription for the isometry $V$. Then, for any subset $\mathbf{b} \subseteq \{b_1 \dots b_n\}$, if there exists a state $|\Psi\rangle \in \mathcal{H}_{\text{code}} \otimes \mathcal{H}_R \otimes \mathcal{H}_{\overline{R}}$ such that $\mathbf{b} = \text{EW}'_{BR}(|\Psi\rangle)$, then*

$$A'_B(\mathbf{b}) = A_B(\mathbf{b}). \tag{23}$$

*Proof.* See Appendix A.2. ∎

**Corollary 2.10.** *For any area function $A'_B$ that satisfies a QMS prescription for an isometry $V$,*

$$\mathbf{b} = \text{EW}'_{BR}(|\Psi\rangle) \implies \mathbf{b} = \text{EW}_{BR}(|\Psi\rangle). \tag{24}$$

*Proof.* By assumption, for all product unitaries $U$, $S(BR)_{VU|\Psi\rangle} = A'_B(\mathbf{b}) + S(\mathbf{b}R)_{U|\Psi\rangle}$. By Theorem 2.9 this implies that also $S(BR)_{VU|\Psi\rangle} = A_B(\mathbf{b}) + S(\mathbf{b}R)_{U|\Psi\rangle}$, and therefore $\mathbf{b} = \text{EW}_{BR}(|\Psi\rangle)$. ∎

# 3 State-specific reconstruction

The same basic idea – that certain logical quantum information is encoded in a particular physical system – underlies the many versions of quantum error correction: exact, approximate, subsystem, algebraic, etc. In this section, we introduce a new kind of quantum error correction, which we call "state-specific product unitary reconstruction." It is important to holography because it naturally defines the entanglement wedge, a fact that we prove in Theorem 4.2. It is also of independent interest as a type of generalized quantum error correction, which includes most more conventional definitions as special cases. In order to motivate it, we first review the usual, "state-independent" version of quantum error correction. We then gradually generalize the definition until we reach state-specific reconstruction.

---

[14]In fact, it fails for highly incompressible states at leading order in $G$.

### 3.1 State-independent quantum error correction

State-independent quantum error correction can be formalized in a number of equivalent ways. The following theorem reviews some of these ways; see e.g. [1] for others.[15]

**Theorem 3.1** (Formulations of exact state-independent QEC)**.** *Let* $V : \mathcal{H}_{\text{code}} \rightarrow \mathcal{H}_B \otimes \mathcal{H}_{\overline{B}}$ *be an isometry between finite-dimensional Hilbert spaces. Then the following four statements are equivalent:*

1. (Schrödinger picture) *There exists a "recovery isometry"* $W_B : \mathcal{H}_B \rightarrow \mathcal{H}_{\text{code}} \otimes \mathcal{H}_E$ *such that for any state* $\rho_{\text{code}}$

$$\text{tr}_{\overline{B}E}[W_B V \rho_{\text{code}} V^\dagger W_B^\dagger] = \rho_{\text{code}} \,. \tag{25}$$

2. (Heisenberg picture) *There exists a "recovery isometry"* $W_B : \mathcal{H}_B \rightarrow \mathcal{H}_{\text{code}} \otimes \mathcal{H}_E$ *such that for any Hermitian operator* $O_{\text{code}}$, *the Hermitian operator* $O_B := W_B^\dagger O_{\text{code}} W_B$ *satisfies*

$$\langle \psi | V^\dagger O_B V | \psi \rangle = \langle \psi | O_{\text{code}} | \psi \rangle \,, \tag{26}$$

*for all* $|\psi\rangle \in \mathcal{H}_{\text{code}}$.

3. (Reconstruction of Hermitian operators) *For any Hermitian operator* $O_{\text{code}}$, *there exists a Hermitian operator* $O_B$ *such that for all* $|\psi\rangle \in \mathcal{H}_{\text{code}}$

$$O_B V | \psi \rangle = V O_{\text{code}} | \psi \rangle \,. \tag{27}$$

4. (Reconstruction of unitary operators) *For any unitary operator* $U_{\text{code}}$, *there exists a unitary operator* $U_B$ *such that for all* $|\psi\rangle \in \mathcal{H}_{\text{code}}$

$$U_B V | \psi \rangle = V U_{\text{code}} | \psi \rangle \,. \tag{28}$$

*Proof.* See Appendix A.3. □

All four conditions in Theorem 3.1 have natural physical interpretations. Condition 1 says that there exists a unitary evolution $W_B$ that recovers the code space state $|\psi\rangle$ from the reduced state of $V|\psi\rangle$ on $\mathcal{H}_B$. Condition 2 is the standard dual Heisenberg picture: for any measurement $O_{\text{code}}$ on the code space, we can find a measurement $O_B := W_B^\dagger O_{\text{code}} W_B$ with the same expectation values. Condition 3 has a similar physical interpretation to Condition 2 in terms of simulating measurements, but is naively slightly stronger (although equivalent in reality) since it requires the measurement operator $O_B$ to have the correct action on all states $V|\psi\rangle$ rather than merely the correct expectation values.

Finally, Condition 4 says that any unitary evolution of the code space can be simulated by a unitary evolution of the state on $\mathcal{H}_B$. In other words, we can perfectly manipulate the state $|\psi\rangle$ while only having control over $\mathcal{H}_B$.

It turns out that Condition 4 of Theorem 3.1 will most naturally generalize to the definition of state-specific reconstruction that we introduce in this paper. Although less common than some other definitions of quantum error correction – in particular the Schrödinger picture (Condition 1) – in the quantum computing literature, it is also a very natural perspective from the point of view of holography, where we are often interested in finding boundary protocols for preparing particular bulk states.[16] We will return to it when we talk about state-specific reconstruction.

---

[15]To see the equivalence of the conditions in Theorem 3.1 below and those in Theorem 3.1 of [1], note that Condition 3 below is manifestly equivalent to the Hermitian and anti-Hermitian parts of Condition 2 of Theorem 3.1 of [1].

[16]See e.g. the discussion of reconstruction *complexity* in [24–26]. The circuit complexity of reconstruction is only well defined when reconstructing unitary bulk operators using unitary boundary operators.

There is a natural generalization of QEC, called subsystem QEC, where only a subsystem $\mathcal{H}_b$ of $\mathcal{H}_{\text{code}}$ needs to be reconstructible. The four conditions for QEC from Theorem 3.1 all generalize to subsystem codes, as we now review; again, see e.g. [1] for other equivalent conditions.[17]

**Theorem 3.2** (Subsystem QEC). *Let* $V : \mathcal{H}_{\text{code}} \cong \mathcal{H}_b \otimes \mathcal{H}_{\bar{b}} \to \mathcal{H}_B \otimes \mathcal{H}_{\overline{B}}$ *be an isometry between finite-dimensional Hilbert spaces. Then the following four statements are equivalent:*

1. (Schrödinger picture) *There exists a "recovery isometry"* $W_B : \mathcal{H}_B \to \mathcal{H}_b \otimes \mathcal{H}_E$ *such that for any state* $\rho_{\text{code}}$

$$\text{tr}_{\overline{B}E}[W_B V \rho_{\text{code}} V^\dagger W_B^\dagger] = \rho_b \,. \tag{29}$$

2. (Heisenberg picture) *There exists a "recovery isometry"* $W_B : \mathcal{H}_B \to \mathcal{H}_b \otimes \mathcal{H}_E$ *such that for any Hermitian operator* $O_b$*, the Hermitian operator* $O_B := W_B^\dagger O_{\text{code}} W_B$ *satisfies*

$$\langle \psi | V^\dagger O_B V | \psi \rangle = \langle \psi | O_b | \psi \rangle \,, \tag{30}$$

*for all* $|\psi\rangle \in \mathcal{H}_{\text{code}}$.

3. (Reconstruction of Hermitian operators) *For any Hermitian operator* $O_b$*, there exists a Hermitian operator* $O_B$ *such that for all* $|\psi\rangle \in \mathcal{H}_{\text{code}}$

$$O_B V |\psi\rangle = V O_b |\psi\rangle \,. \tag{31}$$

4. (Reconstruction of unitary operators) *For any unitary operator* $U_b$*, there exists a unitary operator* $U_B$ *such that for all* $|\psi\rangle \in \mathcal{H}_{\text{code}}$

$$U_B V |\psi\rangle = V U_b |\psi\rangle \,. \tag{32}$$

*Proof.* See Appendix A.3. $\qquad\qquad\square$

**Remark 3.3.** Subsystem QEC can be further generalized to operator algebra QEC, where the set of reconstructible operators can be a von Neumann subalgebra acting on $\mathcal{H}_{\text{code}}$, but that generalization will only play a small part in this paper.

Note that subsystem QEC is a weaker condition than ordinary QEC – only certain unitaries need to be reconstructible – but it is also a generalization of ordinary QEC (ordinary QEC is the special case of a subsystem code where the reconstructible subsystem is the entire input Hilbert space).

**Definition 3.4** (Complementary Recovery). We say that state-independent *complementary recovery* is possible for an isometry $V : \mathcal{H}_b \otimes \mathcal{H}_{\bar{b}} \to \mathcal{H}_B \otimes \mathcal{H}_{\overline{B}}$ if system $\mathcal{H}_B$ forms a subsystem QEC code for $\mathcal{H}_b$ and simultaneously system $\mathcal{H}_{\overline{B}}$ forms a subsystem QEC code for the complementary subsystem $\mathcal{H}_{\bar{b}}$.

## 3.2 Approximate codes and state specificity

Theorem 3.1 and Theorem 3.2 are about 'exact' QEC codes, where information can always be perfectly recovered without any error. This assumption greatly simplifies the analysis, but in practice it is an assumption that is almost never truly valid, whether in real world experiments with quantum computers or in theoretical applications such as quantum gravity.

---

[17]Theorem 3.2 should be compared to Theorem 4.1 of [1]. Again, the correspondence between Condition 3 of Theorem 3.2 and Condition 2 of Theorem 4.1 of [1] is immediate.

To deal with errors, we need to use a more general framework called *approximate QEC* [27–30]. Often, doing so merely adds considerable effort, while leading to the same qualitative conclusions: so, for example, there exist analogous versions of Theorem 3.1 and Theorem 3.2 for approximate QEC codes, with all the various conditions replaced by approximate versions of the same condition, and with different errors all bounded in terms of one another. For example, (25) can be replaced by the statement that there exists a $W_B$ such that for any state $\rho_{\text{code}}$,

$$\|\text{tr}_{\overline{B}E}[W_B V \rho_{\text{code}} V^\dagger W_B^\dagger] - \rho_{\text{code}}\|_1 \leq \varepsilon, \tag{33}$$

for some small $\varepsilon > 0$, where $\|X\|_1 = \text{tr}\sqrt{X^\dagger X}$ is the Schatten 1-norm.[18]

However, the relationship between exact and approximate QEC is not always so simple. It turns out that for large quantum systems approximate QEC can be possible with very small $\varepsilon$, even when exact QEC is completely impossible. In essence, two limits do not commute: a) the number of qubits $n$ goes to infinity and b) the error $\varepsilon$ goes to zero. A classic example of this effect is that exact QEC is always impossible in the presence of arbitrary errors of greater than a quarter of the qubits, whereas approximate QEC is still possible so long as the errors are on fewer than half of the qubits [32].

To see these qualitative differences in practice, consider yet another definition of quantum error correction, known as the *information-disturbance trade-off*. In this definition, rather than studying the information accessible from $\mathcal{H}_B$, we focus on the information that is *inaccessible* from the thrown away degrees of freedom in $\mathcal{H}_{\overline{B}}$. These are equivalent because in quantum mechanics information can neither be copied (the no-cloning theorem) nor destroyed (the no-erasure theorem). Hence quantum information can be recovered from system $B$ if and only if the complementary subsystem $\overline{B}$ is completely ignorant of it.

In other words, equivalent to the four statements in Theorem 3.1 is the statement that for all states $|\psi\rangle \in \mathcal{H}_{\text{code}}$,

$$\psi_{\overline{B}} = \text{tr}_B[V|\psi\rangle\langle\psi|V^\dagger] = \omega_{\overline{B}}, \tag{34}$$

where $\omega_{\overline{B}}$ is a fixed state that is independent of $|\psi\rangle$. That said, this statement is only equivalent in the context of *exact* QEC. In approximate QEC we need to be more careful. Suppose for all $|\psi\rangle \in \mathcal{H}_{\text{code}}$ we have

$$\|\psi_{\overline{B}} - \omega_{\overline{B}}\|_1 \leq \varepsilon. \tag{35}$$

It seems like an observer with access only to $\mathcal{H}_{\overline{B}}$ learns nothing about $|\psi\rangle$, and so by the information-disturbance tradeoff, $\mathcal{H}_B$ should learn everything about $|\psi\rangle$. But that conclusion would be too quick.

Let us consider a new state $|\Psi\rangle \in \mathcal{H}_{\text{code}} \otimes \mathcal{H}_R$ where $\mathcal{H}_R$ is an auxiliary reference system isomorphic to $\mathcal{H}_{\text{code}}$. Crucially, suppose that the observer of $\mathcal{H}_{\overline{B}}$ also has access to $\mathcal{H}_R$. For the observer to truly learn nothing about $|\Psi\rangle$ from $\mathcal{H}_{\overline{B}}$ (beyond the reduced state $\Psi_R$ that they already know), then we need

$$\|\Psi_{\overline{B}R} - \omega_{\overline{B}} \otimes \Psi_R\|_1 \leq \varepsilon', \tag{36}$$

for some small $\varepsilon'$. But (35) being true for all $|\psi\rangle \in \mathcal{H}_{\text{code}}$ and small $\varepsilon$ does not mean that (36) is true for all $|\Psi\rangle \in \mathcal{H}_{\text{code}} \otimes \mathcal{H}_R$ and small $\varepsilon'$. Instead, the tightest possible bound on (36) from (35) is that $\varepsilon' \leq O(d_{\text{code}}\varepsilon)$ [33]. Since the code space dimension $d_{\text{code}}$ is exponential in the number of qubits, this means that we have essentially no control over $\varepsilon'$ at all, given reasonable

---

[18]This is only one possible definition of the reconstruction error $\varepsilon$. Other natural possibilities include replacing the Schatten 1-norm in (33) by the quantum fidelity, or replacing the states $\rho_{\text{code}}$ by states in $\mathcal{H}_{\text{code}} \otimes \mathcal{H}_R$ where the reference system $\mathcal{H}_R$ has dimension $d_R \geq d_{\text{code}}$. However all these definitions are equivalent in the sense that the different errors uniformly bound one another in the limit $\varepsilon \to 0$. See e.g. Proposition 4.3 of [31]. This is the same sense in which Conditions 1-4 are equivalent in Theorem 3.5 below.

values of $\varepsilon$. The tiny corrections to $\psi_{\overline{B}}$ allowed by (35) can build up in superposition and give large corrections to $\Psi_{\overline{B}R}$.

As the intuition from the information disturbance tradeoff suggests, it is (36) that is equivalent to state-independent approximate QEC as in (33) [34]. However, while (35) is too weak to imply *those* conditions, it does nonetheless imply something meaningful about the information accessible from $\mathcal{H}_B$. Specifically, in the language of [6, 35], it says that $\mathcal{H}_B$ *encodes the zero-bits of* $\mathcal{H}_{\text{code}}$. Other names for this are that $B$ can do universal subspace QEC, or that $B$ can do quantum identification [33, 35].

As with state-independent quantum error correction, there exist various equivalent operational definitions of a zero-bit code. While we review a number of these below in Theorem 3.5, in many ways the nicest definition is Condition 4, which is analogous to Condition 4 (Reconstruction of unitary operators) from Theorem 3.1. As in that condition, for any unitary operator $U_{\text{code}}$ and state $|\psi\rangle \in \mathcal{H}_{\text{code}}$ there needs to exist an (approximate) unitary reconstruction $U_B$ such that

$$U_B V |\psi\rangle \approx V U_{\text{code}} |\psi\rangle . \tag{37}$$

However, the reconstruction $U_B$ is now allowed to depend not only on the unitary $U_{\text{code}}$ that it is reconstructing, but also on the state $|\psi\rangle$ itself. We can implement an arbitrary evolution of the state $|\psi\rangle$ while acting only on $\mathcal{H}_B$, but we have to know the initial state $|\psi\rangle$ in order to do so. The reconstruction is therefore "state specific."

**Theorem 3.5** (Zero-bit codes)**.** *Let* $V : \mathcal{H}_{\text{code}} \rightarrow \mathcal{H}_B \otimes \mathcal{H}_{\overline{B}}$ *be an isometry between finite-dimensional Hilbert spaces. Then the following four statements are equivalent:*

1. (Forgetfulness of the environment) *For all states* $|\psi\rangle$, *the reduced density matrix* $\psi_{\overline{B}} = \text{tr}_B[V |\psi\rangle \langle\psi| V^{\dagger}]$ *satisfies*

$$\|\psi_{\overline{B}} - \omega_{\overline{B}}\|_1 \le \varepsilon_1 , \tag{38}$$

*for some fixed density matrix* $\omega_{\overline{B}}$.

2. (Universal subspace quantum error correction) *For any two-dimensional subspace* $\widetilde{\mathcal{H}}_{\text{code}} \subseteq \mathcal{H}_{\text{code}}$, *there exists an isometry* $\widetilde{W}_B : \mathcal{H}_B \rightarrow \widetilde{\mathcal{H}}_{\text{code}} \otimes \mathcal{H}_E$ *such that for all density matrices* $\tilde{\rho}_{\text{code}}$ *with support only on* $\widetilde{\mathcal{H}}_{\text{code}}$

$$\left\|\text{tr}_{\overline{B}E}[\widetilde{W}_B V \tilde{\rho}_{\text{code}} V^{\dagger} \widetilde{W}_B^{\dagger}] - \tilde{\rho}_{\text{code}}\right\|_1 \le \varepsilon_2 . \tag{39}$$

3. (Distinguishing quantum states) *For any pair of orthogonal states* $|\psi\rangle, |\phi\rangle \in \mathcal{H}_{\text{code}}$, *there exists a projector* $\Pi_B$ *such that*

$$\langle\psi|V^{\dagger}\Pi_B V|\psi\rangle \ge 1 - \varepsilon_3 , \quad \text{and} \quad \langle\phi|V^{\dagger}\Pi_B V|\phi\rangle \le \varepsilon_3 . \tag{40}$$

4. (State-specific reconstruction of unitary operators) *For some fixed state* $|\psi_0\rangle \in \mathcal{H}_{\text{code}}$ *and for any unitary operator* $U_{\text{code}}$, *there exists a unitary* $U_B$ *such that the inner product*

$$\left\|V U_{\text{code}} |\psi_0\rangle - U_B V |\psi_0\rangle\right\| \le \varepsilon_4 . \tag{41}$$

*Specifically, Condition 1 implies Condition 2 with* $\varepsilon_2 \le 4\sqrt{3\varepsilon_1}$; *Condition 2 implies Condition 3 with* $\varepsilon_3 \le 2\varepsilon_2$; *Condition 3 implies Condition 4 with* $\varepsilon_4 \le 2\sqrt{\varepsilon_3}$; *and Condition 4 implies Condition 1 with* $\varepsilon_1 \le 2\varepsilon_4$.

*Proof.* See Appendix A.3. $\qquad\square$

It is worth emphasizing that Condition 4 of Theorem 3.5 does not depend on the choice of fixed state $|\psi_0\rangle$. Given any alternative choice $|\psi_0'\rangle$, there exists a unitary $U_{\text{code}}'$ such that $|\psi_0'\rangle = U_{\text{code}}' |\psi_0\rangle$. Now let $|\psi\rangle = U_{\text{code}} |\psi_0'\rangle = U_{\text{code}}'' |\psi_0\rangle$. We have

$$U_B'' U_B'^\dagger V |\psi_0'\rangle \approx U_B'' V |\psi_0\rangle \approx V |\psi\rangle \,. \tag{42}$$

So $U_B'' U_B'^\dagger$ is a state-specific reconstruction of $U_{\text{code}}$ for the state $|\psi_0'\rangle$.

It is illuminating to note how this story changes if we instead consider states $|\Psi_0\rangle \in \mathcal{H}_{\text{code}} \otimes \mathcal{H}_R$ for some large reference system $\mathcal{H}_R \cong \mathcal{H}_{\text{code}}$. Suppose we know that, analogously to Condition 4, for any $U_{\text{code}}$, there exists $U_B$ such that

$$U_B V |\Psi_0\rangle \approx V U_{\text{code}} |\Psi_0\rangle \,. \tag{43}$$

If $|\Psi_0\rangle$ factorizes between $\mathcal{H}_{\text{code}}$ and $\mathcal{H}_R$, then this is exactly Condition 4 and we only have a zero-bit code. However, if $|\Psi_0\rangle$ is highly entangled, then it becomes a much stronger condition: roughly speaking, rather than only needing to act correctly on a single state, $U_B$ needs to act (approximately) correctly on all states in the Schmidt decomposition of $|\Psi_0\rangle$. When $|\Psi_0\rangle$ is maximally entangled, this is (average case) state-independent approximate quantum error correction.[19]

Here we can see the first hints at how phase transitions in the size of an entanglement wedge work; as the entanglement entropy of $|\Psi_0\rangle$ grows, it becomes harder to reconstruct unitaries $U_{\text{code}}$ using $\mathcal{H}_B$.

## 3.3 State-specific product unitary reconstruction

Entanglement wedge reconstruction in holographic codes involves all of the generalizations of quantum error correction introduced in Sections 3.1 and 3.2:

- A boundary region $B$ can only reconstruct operators acting on certain subsystems of the bulk Hilbert space, namely those within its entanglement wedge.

- The reconstructions are state-specific, in a way that generalizes zero-bit codes [6, 35].

- The reconstructions are approximate, with errors of at least $e^{-\mathcal{O}(1/G)}$ [4, 18].

To include all these features within a single formalism, we need to introduce a new type of reconstruction that we call "state-specific product unitary reconstruction," or just "state-specific reconstruction" for short. This is of course the type of reconstruction that appears as Condition 1 in Theorem 4.2 (except for the non-zero error $\varepsilon$, which is incorporated in Condition 1 in Theorem 5.5); we will now introduce it carefully. While the aim here is to formalize the specific features of entanglement wedge reconstruction in holographic codes, state-specific product unitary reconstruction is very general and includes as special cases all the types of quantum error correction discussed above.

Let $\mathcal{H}_{\text{code}}$ be an input Hilbert space factorizing as a product of $n$ "local" Hilbert spaces $\mathcal{H}_{b_i}$, with the isometry

$$V : \mathcal{H}_{\text{code}} \cong \mathcal{H}_{b_1} \otimes \mathcal{H}_{b_2} \dots \mathcal{H}_{b_n} \to \mathcal{H}_B \otimes \mathcal{H}_{\overline{B}} \,,$$

as in Figure 2. Consider some subset of the input subsystems,

$$\mathbf{b} = \{b_{i_1}, b_{i_2}, \dots\} \subseteq \{b_1, \dots b_n\} \,. \tag{44}$$

---

[19]Technically, (43) is slightly weaker than e.g. (33), because there can exist a few states $|\psi\rangle \in \mathcal{H}_{\text{code}}$ for which the reconstruction $U_B$ does not work, so long as such states only make up a small fraction of the full Hilbert space $\mathcal{H}_{\text{code}}$. In contrast, we demanded that (33) was true for all states without exception. However, in practice, the two conditions are comparably "difficult" to achieve; for example the quantum capacity of a channel is the same by either definition.

Finally let $|\Psi\rangle \in \mathcal{H}_{\text{code}} \otimes \mathcal{H}_R \otimes \mathcal{H}_{\overline{R}}$ be an arbitrary state. We want to introduce an appropriate definition of state-specific reconstruction of $\mathcal{H}_{\mathbf{b}}$ using $\mathcal{H}_B \otimes \mathcal{H}_R$

It turns out to be most natural to define state-specific reconstruction in terms of the reconstruction of unitary operators as in Condition 4 of Theorems 3.1, 3.2, and 3.5. This is, after all, the only condition that appeared in closely related form in all three theorems! A more precise reason is that Conditions 2 and 3 of Theorem 3.5 really involve a choice of two states (either two orthogonal states or two states whose span defines $\tilde{H}_{\text{code}}$) rather than just one,[20] and this is hard to generalize to entangled states in subsystem codes.

A naive first guess then would be to define state-specific reconstruction for the state $|\Psi\rangle$ by the requirement that for any unitary $U_{\mathbf{b}}$ acting on $\mathcal{H}_{\mathbf{b}} \cong \mathcal{H}_{b_{i_1}} \otimes \mathcal{H}_{b_{i_2}} \ldots$, there exists a reconstruction $U_{BR}$ such that

$$U_{BR} V |\Psi\rangle \approx V U_{\mathbf{b}} |\Psi\rangle . \tag{45}$$

However, it turns out that this is not always possible in holography, even if the reconstruction $U_{BR}$ can be specific to the state $|\Psi\rangle$ with $\mathbf{b}$ the entanglement wedge of $BR$ for the state $|\Psi\rangle$. The reason is that the entanglement wedge $\mathbf{b}$ depends on the entanglement structure of the bulk state $|\Psi\rangle$. If the operator $U_{\mathbf{b}}$ can change the entanglement structure of $|\Psi\rangle$, it can change the entanglement wedge, and thereby, for example, change the entropy $S(BR)$. Clearly this change cannot be achieved by any unitary $U_{BR}$.

It is therefore natural to restrict $U_{\mathbf{b}}$ to operators that don't change the entanglement structure of $|\Psi\rangle$ – namely product unitaries.

**Definition 3.6** (Product unitary). A unitary $U_{\mathbf{b}}$ acting on $\mathcal{H}_{\mathbf{b}} = \mathcal{H}_{b_{i_1}} \otimes \mathcal{H}_{b_{i_2}} \otimes \ldots$ is said to be a product unitary if it can be written as a product of local unitaries $U_{b_{i_1}} \otimes U_{b_{i_2}} \otimes \ldots$.

We saw a similar effect in the discussion at the end of Section 3.2, where the entanglement structure of $|\Psi_0\rangle \in \mathcal{H}_{\text{code}} \otimes \mathcal{H}_R$ – which, crucially, couldn't be changed by unitaries $U_{\text{code}}$ – controlled how easy reconstruction was. We therefore define state-specific product unitary reconstruction by the following set of two equivalent conditions:

**Theorem 3.7** (State-specific product unitary reconstruction).
*Let $V : \mathcal{H}_{\text{code}} \cong \mathcal{H}_{b_1} \otimes \mathcal{H}_{b_2} \ldots \mathcal{H}_{b_n} \to \mathcal{H}_B \otimes \mathcal{H}_{\overline{B}}$ be an isometry, with $\mathbf{b} = \{b_{i_1}, b_{i_2} \ldots\}$ a subset of input legs. Finally, let $|\Psi\rangle \in \mathcal{H}_{\text{code}} \otimes \mathcal{H}_R \otimes \mathcal{H}_{\overline{R}}$ be a fixed, arbitrary state with $\mathcal{H}_R, \mathcal{H}_{\overline{R}}$ reference systems of arbitrary dimension. Then the following two statements are equivalent:*

1. *(State-specific reconstruction of product unitaries) For any product unitary $U_{\mathbf{b}}$, there exists a unitary reconstruction $U_{BR}$ such that*

$$\left\| U_{BR} V |\Psi\rangle - V U_{\mathbf{b}} |\Psi\rangle \right\| \leq \varepsilon_1 . \tag{46}$$

2. *(Forgetfulness of product unitaries by the environment) For any product unitary $U_{\mathbf{b}}$, we have*

$$\left\| \text{tr}_{BR}[V U_{\mathbf{b}} |\Psi\rangle \langle\Psi| U_{\mathbf{b}}^\dagger V^\dagger] - \text{tr}_{BR}[V |\Psi\rangle \langle\Psi| V^\dagger] \right\|_1 \leq \varepsilon_2 . \tag{47}$$

*Specifically, Condition 1 implies Condition 2 with $\varepsilon_2 \leq 2\varepsilon_1$, while Condition 2 implies Condition 1 with $\varepsilon_1 \leq \sqrt{\varepsilon_2}$.*

*Proof.* See Appendix A.3. □

---

[20]Physically, this is because state evolution is nontrivial to implement even when the initial state is already known, whereas measurements are only nontrivial if the system can be in at least two possible states.

**Remark 3.8.** In the special case where $n = 1$, $\mathbf{b} = \{b_1\}$ and $d_R = d_{\overline{R}} = 1$, Theorem 3.7 reduces to Conditions 1 and 4 of Theorem 3.5.

**Remark 3.9.** The "no cloning theorem" does not prevent state-specific product unitary reconstruction of the same subsystem from both $\mathcal{H}_B$ and $\mathcal{H}_{\overline{B}}$.[21] Consider the example from Remark 2.6 where the state $|\psi\rangle$ has $\mathcal{H}_{b_1} \cong \mathcal{H}_B$ maximally entangled with $\mathcal{H}_{b_2} \cong \mathcal{H}_{\overline{B}}$ (with $V = \mathbb{1}$, $d_{b_1} > d_{b_2}$ and $d_R = d_{\overline{R}} = 1$). Since all maximally entangled states are related by a unitary operator acting on the larger Hilbert space, $\{b_1, b_2\}$ can be reconstructed from $\mathcal{H}_B$. However $b_2$ can also clearly be reconstructed from $\mathcal{H}_{\overline{B}}$.

This type of reconstruction is almost sufficient to define the entanglement wedge. However it still misses one important aspect: complementary recovery. Given any holographic bulk state $|\Psi\rangle$ with a well defined entanglement wedge, the entanglement wedge of $\overline{B}\,\overline{R}$ is always the complement of the entanglement wedge of $BR$. Hence, any bulk subsystem $\mathcal{H}_{b_i}$ that cannot be reconstructed from $\mathcal{H}_B \otimes \mathcal{H}_R$ can always be reconstructed from $\mathcal{H}_{\overline{B}} \otimes \mathcal{H}_{\overline{R}}$.

Note it is crucial here that we are considering the purified state $|\Psi\rangle$. In general, the entanglement wedges of $B$ and $\overline{B}$ alone will not be complementary for a mixed state $V\rho_{\text{code}}V^\dagger$, unless we are working within a code space where the entanglement wedges of $B$ and $\overline{B}$ are fixed and hence state-independent complementary reconstruction is possible, as in [1].

With this, we have finally reached the full version of reconstruction that defines the entanglement wedge – complementary state-specific product unitary reconstruction.

**Definition 3.10** (Complementary state-specific product unitary reconstruction)**.**
Let $V : \mathcal{H}_{\text{code}} \cong \otimes_i \mathcal{H}_{b_i} \to \mathcal{H}_B \otimes \mathcal{H}_{\overline{B}}$ be an isometry, with $\mathbf{b} = \{b_{i_1}, b_{i_2}...\}$ a subset of input legs. Finally, let $|\Psi\rangle \in \mathcal{H}_{\text{code}} \otimes \mathcal{H}_R \otimes \mathcal{H}_{\overline{R}}$ be a fixed, arbitrary state with $\mathcal{H}_R, \mathcal{H}_{\overline{R}}$ reference systems of arbitrary dimension. We say that complementary state-specific product unitary reconstruction is possible if for any product unitary $U_{\mathbf{b}}$ there exists a unitary reconstruction $U_{BR}$, and for any product unitary $U'_{\mathbf{b}}$ there exists a unitary reconstruction $U'_{\overline{B}\,\overline{R}}$, such that for all pairs of product unitaries $U_{\mathbf{b}}, U'_{\mathbf{b}}$

$$U_{BR} U'_{\overline{B}\,\overline{R}} V |\Psi\rangle \approx V U_{\mathbf{b}} U'_{\mathbf{b}} |\Psi\rangle \,. \tag{48}$$

Since complementary state-specific reconstruction seems a priori to be fairly distinct from traditional definitions of quantum error correction, one might wonder what the relationship is between Theorems 4.2 and 5.5 and Theorem 4.1 of [1]. The answer is that complementary state-independent reconstruction is possible whenever complementary *state-specific* reconstruction is possible for all states $|\Psi\rangle$, with the entanglement wedge $\mathbf{b}$ independent of the state $|\Psi\rangle$.

**Theorem 3.11** (From state-specific to state-independent reconstruction)**.**
*Let $V : \mathcal{H}_{\text{code}} \cong \otimes_i \mathcal{H}_{b_i} \to \mathcal{H}_B \otimes \mathcal{H}_{\overline{B}}$ be an isometry, with $\mathbf{b} = \{b_{i_1}, b_{i_2}...\}$ a subset of input legs. Finally, let $|\Psi\rangle \in \mathcal{H}_{\text{code}} \otimes \mathcal{H}_{\overline{R}}$ be an arbitrary state with the reference system $\mathcal{H}_{\overline{R}}$ having dimension $d_{\overline{R}} = d_{code}$. Then the following three statements are equivalent:*

1. *(State-specific reconstruction of product unitaries for all states) For every state $|\Psi\rangle$ and product unitary $U_{\mathbf{b}}$, there exists a unitary reconstruction $U_B$ such that*

$$\|U_B V |\Psi\rangle - V U_{\mathbf{b}} |\Psi\rangle\| \le \varepsilon_1 \,. \tag{49}$$

2. *(State-independent reconstruction of product unitaries) For every product unitary $U_{\mathbf{b}}$, there exists a unitary reconstruction $U_B$ such that for all states $|\Psi\rangle$*

$$\|U_B V |\Psi\rangle - V U_{\mathbf{b}} |\Psi\rangle\| \le \varepsilon_2 \,. \tag{50}$$

---

[21]Instead, the no cloning theorem only rules out reconstruction of a subsystem by both $\mathcal{H}_B$ and $\mathcal{H}_{\overline{B}}$ if in the state $|\Psi\rangle$ that subsystem is maximally entangled with a reference system.

3. (Schrödinger-picture subsystem code) *There exists an isometry* $W_B : \mathcal{H}_B \to \mathcal{H}_{\mathbf{b}} \otimes \mathcal{H}_E$, *such that for all states* $\rho_{\text{code}}$

$$\left\| \text{tr}_{\overline{B}E} [W_B V \rho_{\text{code}} V^\dagger W_B^\dagger] - \text{tr}_{\overline{\mathbf{b}}} [\rho_{\text{code}}] \right\|_1 \leq \varepsilon_3 . \tag{51}$$

*Specifically, Condition 1 implies Condition 2 with $\varepsilon_2 \leq \sqrt{3}\varepsilon_1$, Condition 2 implies Condition 3 with $\varepsilon_3 \leq 2\varepsilon_2$ and Condition 3 implies Condition 1 with $\varepsilon_1 \leq 2\sqrt{\varepsilon_3}$.*

*Proof.* See Appendix A.3. □

# 4 Exact codes

## 4.1 Theorem Statement

In this section we prove that the existence of an exact QMS prescription is equivalent to the existence of exact complementary state-specific product unitary reconstruction. Before stating the theorem, we first formally define the min-entropy $H_{\min}(A|B)_{|\psi\rangle}$, which will play an important role.

**Definition 4.1** (Min-entropy)**.** The conditional min-entropy $H_{\min}(A|B)_{|\psi\rangle}$ of a state $|\psi\rangle \in \mathcal{H}_A \otimes \mathcal{H}_B \otimes \mathcal{H}_C$ is defined as follows. Let $\psi_{AB} = \text{tr}_C[|\psi\rangle \langle\psi|]$. Then

$$H_{\min}(A|B)_{|\psi\rangle} = -\min_{\sigma_B} D_{\max}(\rho_{AB}|\mathbb{1}_A \otimes \sigma_B) = -\min_{\sigma_B} \inf\{\lambda : \psi_{AB} \leq e^\lambda \mathbb{1}_A \otimes \sigma_B\}, \tag{52}$$

where the minimization is over all density matrices $\sigma_B$.

**Theorem 4.2.** *Let $V : \mathcal{H}_{\text{code}} \cong \otimes_i \mathcal{H}_{b_i} \to \mathcal{H}_B \otimes \mathcal{H}_{\overline{B}}$ be an isometry, with $\mathbf{b} = \{b_{i_1}, b_{i_2}...\}$ a subset of input legs, and $\overline{\mathbf{b}}$ its complement. Finally, let $|\Psi\rangle \in \mathcal{H}_{\text{code}} \otimes \mathcal{H}_R \otimes \mathcal{H}_{\overline{R}}$ be a fixed, arbitrary state with $\mathcal{H}_R, \mathcal{H}_{\overline{R}}$ reference systems of arbitrary dimension.*
   *Then the following two conditions are equivalent:*

1. (Complementary Recovery) *For all product unitaries $U_{\mathbf{b}}$ and $U'_{\overline{\mathbf{b}}}$, there exist unitary operators $U_{BR}$ and $U'_{\overline{B}\overline{R}}$ respectively such that*

$$U_{BR} U'_{\overline{B}\overline{R}} V |\Psi\rangle = V U_{\mathbf{b}} U'_{\overline{\mathbf{b}}} |\Psi\rangle . \tag{53}$$

2. (Holographic Entropy Prescription) *For all product unitaries $U_{\mathbf{b}}$ and $U'_{\overline{\mathbf{b}}}$,*

$$S(BR)_{V U_{\mathbf{b}} U'_{\overline{\mathbf{b}}} |\Psi\rangle} = A_B(\mathbf{b}) + S(\mathbf{b}R)_{U_{\mathbf{b}} U'_{\overline{\mathbf{b}}} |\Psi\rangle} . \tag{54}$$

*Moreover, both statements imply:*

3. (One-shot Minimality) *For all $\overline{\mathbf{b}}' \subseteq \overline{\mathbf{b}}$ and $\mathbf{b}' \subseteq \mathbf{b}$,*

$$H_{\min}(\overline{\mathbf{b}}'|\mathbf{b}R)_{|\Psi\rangle} \geq A_B(\mathbf{b}) - A_B(\mathbf{b} \cup \overline{\mathbf{b}}') , \tag{55}$$

$$H_{\min}(\mathbf{b}'|\overline{\mathbf{b}}\overline{R})_{|\Psi\rangle} \geq A_{\overline{B}}(\overline{\mathbf{b}}) - A_{\overline{B}}(\mathbf{b}' \cup \overline{\mathbf{b}}) . \tag{56}$$

*This in turn implies:*

4. (Minimality) *For all $\mathbf{b}' \subseteq \mathbf{b} \cup \overline{\mathbf{b}}$,*

$$A_B(\mathbf{b}') + S(\mathbf{b}'R)_{|\Psi\rangle} \geq A_B(\mathbf{b}) + S(\mathbf{b}R)_{|\Psi\rangle} . \tag{57}$$

## 4.2 Proof

Before proceeding to the main parts of the proof of Theorem 4.2, we first need to introduce some preliminary lemmas. These introduce one of the most important technical tools used in the main proofs, which involves using an isometry $W$ to extract information out of the holographic code and into an auxiliary Hilbert space isomorphic to the space of square-integrable functions $L^2(\mathbf{U})$ on the unitary group $\mathbf{U}$.

**Lemma 4.3.** *Consider the group $\mathbf{U}(d)$ of unitary operators in d-dimensions. Let $\mathcal{H}_{\mathbf{U}(d)}$ be the Hilbert space $L^2(\mathbf{U}(d))$, with "position basis" $|U\rangle_{\mathbf{U}(d)}$, i.e. $\langle U|U'\rangle = \delta(U - U')$, with $\int dU \langle U|U'\rangle = 1$ where $dU$ is the Haar measure on $\mathbf{U}(d)$ normalized so that $\int dU = 1$. Finally, let $\pi : \mathbf{U}(d) \to \mathbf{U}_A$ be a (measurable) map from $U \in \mathbf{U}(d)$ to unitary operators $\pi(U)_A$ on the Hilbert space $\mathcal{H}_A$. Then the following defines an isometry $W_\pi : \mathcal{H}_A \to \mathcal{H}_A \otimes \mathcal{H}_{\mathbf{U}(d)}$ for any map $\pi$:*

$$W_\pi := \int dU\, |U\rangle_{\mathbf{U}(d)} \otimes \pi(U)_A. \tag{58}$$

*Proof.* It suffices to show that $W_\pi$ preserves the norm for all $|\psi\rangle_A$, i.e. $\langle W_\pi \psi | W_\pi \psi \rangle = \langle \psi | \psi \rangle$.[22] This follows directly from

$$\langle \psi | W_\pi^\dagger W_\pi | \psi \rangle = \int dU \langle \psi | \pi(U)^\dagger \pi(U) | \psi \rangle = \langle \psi | \psi \rangle. \tag{59}$$

$\square$

It will be helpful to introduce the following powerful change of basis for $L^2(\mathbf{U}(d))$. Let $\mu$ label an irreducible representation of $\mathbf{U}(d)$ and let $D_{ij}^\mu(U)$ label the matrix elements of the unitary $U$ for that representation. A famous theorem in representation theory guarantees these matrix elements are orthogonal when interpreted as wavefunctions on $L^2(\mathbf{U}(d))$. Specifically, we have

$$d_\mu \int dU D_{ij}^\mu(U)^* D_{i'j'}^{\mu'}(U) = \delta_{\mu\mu'}\delta_{ii'}\delta_{jj'}, \tag{60}$$

see e.g. Appendix A of [36]. Another famous theorem, the Peter-Weyl theorem, guarantees these wavefunctions form a complete basis (see Appendix A of [36]). All wavefunctions $\psi(U)$ can therefore be written

$$\psi(U) = \sum_\mu \sum_{ij} \sqrt{d_\mu} D_{ij}^\mu(U) \psi_{ij}^\mu, \tag{61}$$

where the sum over $\mu$ is over all irreducible representations of $\mathbf{U}(d)$ and

$$\psi_{ij}^\mu = \int dU D_{ij}^\mu(U)^* \psi(U). \tag{62}$$

In Dirac notation, we have

$$|U\rangle = \sum_\mu \sqrt{d_\mu} \sum_{ij} D_{ij}^\mu(U)^* |\mu; ij\rangle. \tag{63}$$

This means that the Hilbert space $L^2(\mathbf{U}(d))$ naturally decomposes into a direct sum over finite-dimensional blocks $\mathcal{H}_\mu \otimes \mathcal{H}_\mu^*$:

$$\mathcal{H}_{\mathbf{U}(d)} = \bigoplus_\mu \mathcal{H}_\mu \otimes \mathcal{H}_\mu^*. \tag{64}$$

With this rewriting in hand, we can now prove the following lemma:

---

[22]This also ensures that $\langle W_\pi \psi | W_\pi \psi \rangle$ is finite for any $|\psi\rangle$, and hence that $W$ is a bounded operator.

**Lemma 4.4.** *Consider the setup from Lemma 4.3, specialized to the case $d = d_A$, and let $\pi = \mu_0$ be the fundamental representation of $\mathbf{U}_A$. Then*

$$W |\psi\rangle_A = \int dU |U\rangle_{\mathbf{U}_A} \otimes U |\psi\rangle_A = |\psi\rangle_{\mu_0} \otimes |\text{MAX}\rangle_{A\mu_0^*} , \tag{65}$$

*where $\mathcal{H}_{\mu_0} \otimes \mathcal{H}_{\mu_0}^*$ is the block associated to the fundamental representation $\mu_0$ in the decomposition (64) and $|\text{MAX}\rangle_{A\mu_0^*} = \frac{1}{\sqrt{d}} \sum_{i=1}^{d} |i\rangle_A |i\rangle_{\mu_0^*}^*$ is the canonical maximally entangled state on $\mathcal{H}_A \otimes \mathcal{H}_{\mu_0}^* \cong \mathcal{H}_A \otimes \mathcal{H}_A^*$.*

*Proof.* Rewriting $W$ using the irrep basis for $L^2(\mathbf{U}(d))$ introduced above, we find

$$
\begin{aligned}
W |\psi\rangle_A &= \int dU |U\rangle_{\mathbf{U}_A} \otimes U |\psi\rangle_A \\
&= \int dU \sum_\mu \sqrt{d_\mu} \sum_{ij} D_{ij}^\mu(U)^* |\mu; ij\rangle_{\mathbf{U}_A} \otimes \sum_{i'} |i'\rangle_A \langle i'| U \sum_{j'} |j'\rangle_A \langle j'|\psi\rangle \\
&= \sum_\mu \sqrt{d_\mu} \sum_{\substack{ij \\ i'j'}} \left( \int dU D_{ij}^\mu(U)^* D_{i'j'}^{\mu_0}(U) \right) |\mu; ij\rangle_{\mathbf{U}_A} \otimes |i'\rangle_A \langle j'|\psi\rangle \\
&= \frac{1}{\sqrt{d}} \sum_{ij} |\mu_0; ij\rangle_{\mathbf{U}_A} \otimes |i\rangle_A \langle j|\psi\rangle \\
&= \left( \sum_j \langle j|\psi\rangle |j\rangle_{\mu_0} \right) \otimes \left( \frac{1}{\sqrt{d}} \sum_i |i\rangle_{\mu_0^*} |i\rangle_A \right) \\
&= |\psi\rangle_{\mu_0} \otimes |\text{MAX}\rangle_{\mu_0^* A} .
\end{aligned}
\tag{66}
$$

$\square$

Using Lemma 4.4, we can construct an isometry $W_{b_i} : \mathcal{H}_{b_i} \to \mathcal{H}_{b_i} \otimes \mathcal{H}_{\mathbf{U}_{b_i}}$ that extracts all the quantum information out of $\mathcal{H}_{b_i}$ and into an isomorphic Hilbert space $\mathcal{H}_{a_i} \cong \mathcal{H}_{b_i}$, with the original subsystem $\mathcal{H}_{b_i}$ left maximally entangled with a reference system $\mathcal{H}_{r_i} \cong \mathcal{H}_{b_i}^*$. Specifically, we have

$$W_{b_i} |\Psi\rangle_{b_1 \dots b_i \dots b_n R \overline{R}} = \int dU_{b_i} |U_{b_i}\rangle_{\mathbf{U}_{b_i}} \otimes U_{b_i} |\Psi\rangle_{b_1 \dots b_i \dots b_n R \overline{R}} \tag{67}$$

$$= |\Psi\rangle_{b_1 \dots a_i \dots b_n R \overline{R}} |\text{MAX}\rangle_{b_i r_i} , \tag{68}$$

where we have identified $\mathcal{H}_{a_i} \cong \mathcal{H}_{\mu_0}$ and $\mathcal{H}_{r_i} \cong \mathcal{H}_{\mu_0}^*$ with the fundamental representation block $\mathcal{H}_{\mu_0} \otimes \mathcal{H}_{\mu_0}^* \subset \mathcal{H}_{\mathbf{U}_{b_i}}$ from Lemma 4.4.

It is then natural to define $W_\mathbf{b} : \mathcal{H}_\mathbf{b} \to \mathcal{H}_\mathbf{b} \otimes \mathcal{H}_{\mathbf{U}_\mathbf{b}}$, where $\mathcal{H}_{\mathbf{U}_\mathbf{b}} \cong \mathcal{H}_{\mathbf{U}_{b_{i_1}}} \otimes \mathcal{H}_{\mathbf{U}_{b_{i_2}}} \dots$ is the Hilbert space of square-integrable functions $L^2(\mathbf{U}_\mathbf{b})$ on the group of *product unitaries* $\mathbf{U}_\mathbf{b} \cong \mathbf{U}_{b_{i_1}} \times \mathbf{U}_{b_{i_2}} \dots$, by

$$W_\mathbf{b} |\Psi\rangle_{\mathbf{b}\overline{\mathbf{b}} R \overline{R}} = [W_{b_{i_1}} W_{b_{i_2}} \dots] |\Psi\rangle_{\mathbf{b}\overline{\mathbf{b}} R \overline{R}} \tag{69}$$

$$= \int dU_\mathbf{b} |U_\mathbf{b}\rangle_{\mathbf{U}_\mathbf{b}} U_\mathbf{b} |\Psi\rangle_{\mathbf{b}\overline{\mathbf{b}} R \overline{R}} \tag{70}$$

$$= |\Psi\rangle_{\mathbf{a}\overline{\mathbf{b}} R \overline{R}} |\text{MAX}\rangle_{\mathbf{br}} . \tag{71}$$

In other words, $W_{\mathbf{b}}$ is an isometry, built out of product unitaries acting on $\mathcal{H}_{\mathbf{b}}$, that extracts the quantum state out of $\mathcal{H}_{\mathbf{b}}$ and into a copy $\mathcal{H}_{\mathbf{a}} \cong \mathcal{H}_{a_{i_1}} \otimes \mathcal{H}_{a_{i_2}} \dots$, leaving the original $\mathcal{H}_{\mathbf{b}}$ maximally entangled with $\mathcal{H}_{\mathbf{r}} \cong \mathcal{H}_{r_{i_1}} \otimes \mathcal{H}_{r_{i_2}} \dots$[23]

Now suppose that $BR$ can do state-specific product operator reconstruction of $\mathbf{b}$ for the state $|\Psi\rangle$, i.e. for every product unitary $U_{\mathbf{b}}$ acting on $\mathcal{H}_{\mathbf{b}}$ there exists a unitary $U_{BR}$ acting on $\mathcal{H}_B \otimes \mathcal{H}_R$ such that

$$U_{BR} V |\Psi\rangle = V U_{\mathbf{b}} |\Psi\rangle . \tag{72}$$

Then we can define an isometry $W_{BR} : \mathcal{H}_B \otimes \mathcal{H}_R \to \mathcal{H}_B \otimes \mathcal{H}_R \otimes \mathcal{H}_{\mathbf{U_b}}$ by

$$W_{BR} = \int dU_{\mathbf{b}} \, |U_{\mathbf{b}}\rangle \otimes U_{BR} , \tag{73}$$

such that

$$W_{BR} V |\Psi\rangle_{\mathbf{b}\bar{\mathbf{b}}R\bar{R}} = V W_{\mathbf{b}} |\Psi\rangle_{\mathbf{b}\bar{\mathbf{b}}R\bar{R}} = V |\Psi\rangle_{\mathbf{a}\bar{\mathbf{b}}R\bar{R}} |\text{MAX}\rangle_{\mathbf{br}} . \tag{74}$$

Importantly, on the right hand side of (74), $V : \mathcal{H}_{\mathbf{b}} \otimes \mathcal{H}_{\bar{\mathbf{b}}} \to \mathcal{H}_B \otimes \mathcal{H}_{\bar{B}}$ and does *not* act on $\mathcal{H}_{\mathbf{a}}$.

**Remark 4.5** (Measurability). Since the map $U_{\mathbf{b}} \to V U_{\mathbf{b}} |\Psi\rangle$ is continuous, we can always choose $U_{BR}$ such that $U_{\mathbf{b}} \to U_{BR}$ is piecewise continuous and hence measurable, as required for $W_{BR}$ to be well defined.

By acting only on $\mathcal{H}_B \otimes \mathcal{H}_R$, we have successfully extracted the quantum state out of $\mathcal{H}_{\mathbf{b}}$ and into the copy $\mathcal{H}_{\mathbf{a}}$, leaving the original $\mathcal{H}_{\mathbf{b}}$ maximally entangled with $\mathcal{H}_{\mathbf{r}}$. Schematically, this can be represented by the following figure

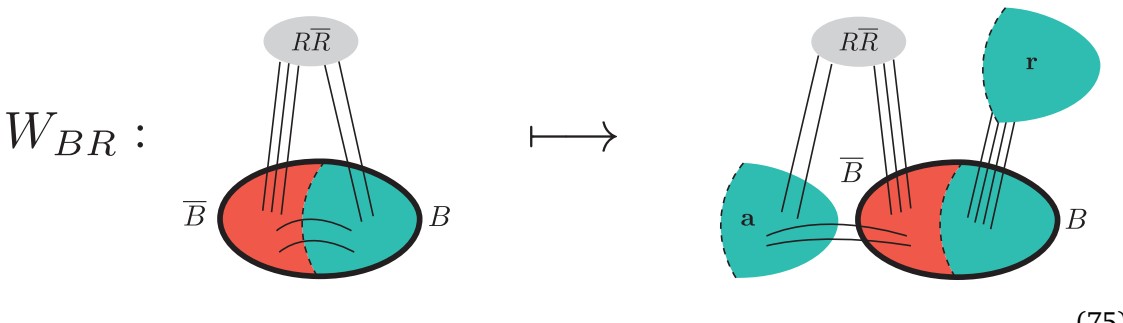

$$\tag{75}$$

Similarly, we can define the isometry $W_{\bar{\mathbf{b}}} : \mathcal{H}_{\bar{\mathbf{b}}} \to \mathcal{H}_{\bar{\mathbf{b}}} \otimes \mathcal{H}_{\mathbf{U}_{\bar{\mathbf{b}}}}$ by

$$W_{\bar{\mathbf{b}}} = \int dU_{\bar{\mathbf{b}}} \, |U_{\bar{\mathbf{b}}}\rangle \otimes U_{\bar{\mathbf{b}}} , \tag{76}$$

that extracts the quantum state out of $\mathcal{H}_{\bar{\mathbf{b}}}$ and into $\mathcal{H}_{\bar{\mathbf{a}}}$. If $\bar{B}\bar{R}$ can do state-specific product operator reconstruction of $\bar{\mathbf{b}}$, there exists an isometry

$$W_{\bar{B}\bar{R}} = \int dU_{\bar{\mathbf{b}}} \, |U_{\bar{\mathbf{b}}}\rangle \otimes U_{\bar{B}\bar{R}} , \tag{77}$$

---

[23]Note that this is not quite the same as directly applying the construction from Lemma 4.4 to the Hilbert space $\mathcal{H}_{\mathbf{b}}$; that would involve integrating over all unitaries acting on $\mathcal{H}_{\mathbf{b}} \cong \mathcal{H}_{b_{i_1}} \otimes \mathcal{H}_{b_{i_1}} \dots$ rather than just product unitaries.

such that $W_{\overline{B}\overline{R}}V|\Psi\rangle = VW_{\overline{\mathbf{b}}}|\Psi\rangle$. Finally, if state-specific product operator reconstruction of *both* $\mathbf{b}$ from $BR$ and $\overline{\mathbf{b}}$ from $\overline{B}\overline{R}$ are possible for the same state $|\Psi\rangle$, as in Condition 1, then we have

$$W_{BR}W_{\overline{B}\overline{R}}V|\Psi\rangle_{\mathbf{b}\overline{\mathbf{b}}R\overline{R}} = |\Psi\rangle_{\mathbf{a}\overline{\mathbf{a}}R\overline{R}} \, V\,|\mathrm{MAX}\rangle_{\mathbf{b}\mathbf{r}}\,|\mathrm{MAX}\rangle_{\overline{\mathbf{b}}\overline{\mathbf{r}}} = |\Psi\rangle_{\mathbf{a}\overline{\mathbf{a}}R\overline{R}}\,|\mathrm{CJ}\rangle_{B\overline{B}\mathbf{r}\overline{\mathbf{r}}}\,. \tag{78}$$

Graphically, we have

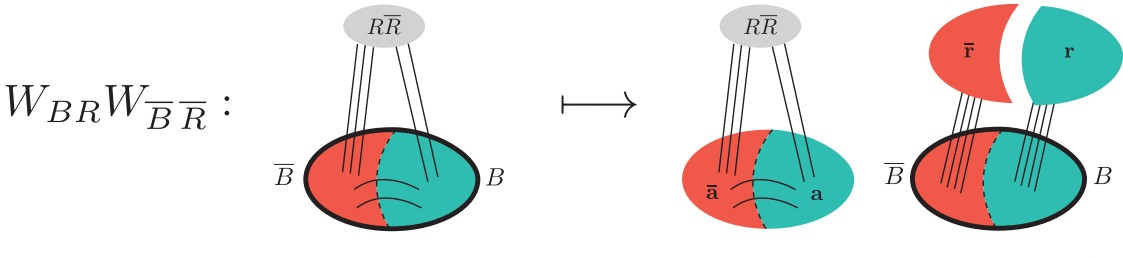

$$\tag{79}$$

To reiterate, if we assume Condition 1 is true, then by acting with the product of an isometry $W_{BR}$ acting only on $\mathcal{H}_B \otimes \mathcal{H}_R$ with an isometry $W_{\overline{B}\overline{R}}$ acting only on $\mathcal{H}_{\overline{B}} \otimes \mathcal{H}_{\overline{R}}$, we can map the boundary state $V|\Psi\rangle$ to the product of the Choi-Jamiolkwoski state $|\mathrm{CJ}\rangle$ and the bulk state $|\Psi\rangle$ stored in the auxilary copy $\mathcal{H}_{\mathbf{a}} \otimes \mathcal{H}_{\overline{\mathbf{a}}}$ of the bulk Hilbert space. This trick will be at the heart of the proof of Theorem 4.2, to which we now turn. It will allow us to relate the boundary entropy $S(BR)_{V|\Psi\rangle}$ to the sum of the bulk entropy $S(\mathbf{b}R)_{|\Psi\rangle}$ and the area term $A_B(\mathbf{b}) = S(B\mathbf{r})_{|\mathrm{CJ}\rangle}$.

**Proof** $(1) \implies (2)$**:**

With (78) in hand, the proof that Condition 1 implies Condition 2 is almost immediate. Since $W_{BR}$ and $W_{\overline{B}\overline{R}}$ are isometries, we have

$$S(BR)_{V|\Psi\rangle} = S(BR\mathbf{U}_{\mathbf{b}})_{W_{BR}W_{\overline{B}\overline{R}}V|\Psi\rangle} \tag{80}$$

$$= S(B\mathbf{r})_{W_{BR}W_{\overline{B}\overline{R}}V|\Psi\rangle} + S(\mathbf{a}R)_{W_{BR}W_{\overline{B}\overline{R}}V|\Psi\rangle} \tag{81}$$

$$= S(B\mathbf{r})_{|\mathrm{CJ}\rangle} + S(\mathbf{b}R)_{|\Psi\rangle} \tag{82}$$

$$= A_B(\mathbf{b}) + S(\mathbf{b}R)_{|\Psi\rangle}\,. \tag{83}$$

So the entropy $S(BR)_{V|\Psi\rangle}$ is given by a holographic entropy prescription. All that remains to complete the proof is to show that, for any $U_{\mathbf{b}}, U'_{\overline{\mathbf{b}}}$, the entropy $S(BR)_{VU_{\mathbf{b}}U'_{\overline{\mathbf{b}}}|\Psi\rangle}$ is also given by a holographic entropy prescription. To see this, first note that

$$S(\mathbf{b}R)_{U_{\mathbf{b}}U'_{\overline{\mathbf{b}}}|\Psi\rangle} = S(\mathbf{b}R)_{|\Psi\rangle}\,, \tag{84}$$

so all we need to do is show that

$$S(BR)_{VU_{\mathbf{b}}U'_{\overline{\mathbf{b}}}|\Psi\rangle} = S(BR)_{V|\Psi\rangle}\,. \tag{85}$$

But Condition 1 tells us that there exist $U_{BR}$ and $U'_{\overline{B}\overline{R}}$ such that

$$U_{BR}U'_{\overline{B}\overline{R}}V|\Psi\rangle = VU_{\mathbf{b}}U'_{\overline{\mathbf{b}}}|\Psi\rangle\,. \tag{86}$$

Hence (85), and thus Condition 2, follows immediately.

**Proof** (2) $\implies$ (1)**:**

The idea here is to show Condition 2 can only hold if a product unitary $U_{\mathfrak{b}}$ (resp. $U'_{\bar{\mathfrak{b}}}$) always leaves the reduced state on $\overline{B}\,\overline{R}$ (resp. $BR$) unchanged. Suppose for example we know that for any $U_{\mathfrak{b}}$

$$\mathrm{tr}_{BR}\left[V U_{\mathfrak{b}} |\Psi\rangle\langle\Psi| U_{\mathfrak{b}}^{\dagger} V^{\dagger}\right] = \mathrm{tr}_{BR}\left[V |\Psi\rangle\langle\Psi| V^{\dagger}\right]. \tag{87}$$

Then $V U_{\mathfrak{b}} |\Psi\rangle$ and $V |\Psi\rangle$ would both be purifications of the same reduced density matrix on $\mathcal{H}_{\overline{B}} \otimes \mathcal{H}_{\overline{R}}$. Since all purifications of a given state are related by a unitary operator acting on the purifying system, we would know that there exists $U_{BR}$ such that

$$U_{BR} V |\Psi\rangle = V U_{\mathfrak{b}} |\Psi\rangle, \tag{88}$$

as desired for Condition 1 to hold.

To show this, we again use the isometries $W_{\mathfrak{b}}$ and $W_{\bar{\mathfrak{b}}}$. Without any assumptions about $V$, we know

$$V W_{\mathfrak{b}} W_{\bar{\mathfrak{b}}} |\Psi\rangle_{\mathfrak{b}\bar{\mathfrak{b}}R\overline{R}} = |\Psi\rangle_{\mathbf{a}\bar{\mathbf{a}}R\overline{R}} |CJ\rangle_{B\overline{B}\mathbf{r}\bar{\mathbf{r}}}, \tag{89}$$

and hence that

$$S(BR\mathbf{ar})_{V W_{\mathfrak{b}} W_{\bar{\mathfrak{b}}} |\Psi\rangle} = S(\overline{B}\,\overline{R}\,\bar{\mathbf{a}}\bar{\mathbf{r}})_{V W_{\mathfrak{b}} W_{\bar{\mathfrak{b}}} |\Psi\rangle} = A_B(\mathbf{b}) + S(\mathbf{b}R)_{|\Psi\rangle}. \tag{90}$$

Tracing out $\mathcal{H}_B \otimes \mathcal{H}_R \otimes \mathcal{H}_{U_{\mathfrak{b}}}$ from (89) leaves

$$\begin{aligned}
\Psi_{\bar{\mathbf{a}}\overline{R}} \otimes |CJ\rangle\langle CJ|_{\overline{B}\bar{\mathbf{r}}} &= \mathrm{tr}_{BRU_{\mathfrak{b}}}\left[V W_{\mathfrak{b}} W_{\bar{\mathfrak{b}}} |\Psi\rangle\langle\Psi| W_{\mathfrak{b}}^{\dagger} W_{\bar{\mathfrak{b}}}^{\dagger} V^{\dagger}\right] \\
&= \mathrm{tr}_{BR}\left[\left[\int dU_{\mathfrak{b}} dU'_{\mathfrak{b}} \langle U'_{\mathfrak{b}}|U_{\mathfrak{b}}\rangle \otimes V U_{\mathfrak{b}} W_{\bar{\mathfrak{b}}} |\Psi\rangle\langle\Psi| W_{\bar{\mathfrak{b}}}^{\dagger} U_{\mathfrak{b}}^{\prime\dagger} V^{\dagger}\right]\right] \\
&= \int dU_{\mathfrak{b}} \,\mathrm{tr}_{BR}\left[V U_{\mathfrak{b}} W_{\bar{\mathfrak{b}}} |\Psi\rangle\langle\Psi| W_{\bar{\mathfrak{b}}}^{\dagger} U_{\mathfrak{b}}^{\dagger} V^{\dagger}\right].
\end{aligned} \tag{91}$$

Therefore,

$$S(\overline{B}\,\overline{R}\,\bar{\mathbf{a}}\bar{\mathbf{r}})_{V W_{\mathfrak{b}} W_{\bar{\mathfrak{b}}} |\Psi\rangle} \geq \int dU_{\mathfrak{b}} \, S(\overline{B}\,\overline{R}\,\bar{\mathbf{a}}\bar{\mathbf{r}})_{V U_{\mathfrak{b}} W_{\bar{\mathfrak{b}}} |\Psi\rangle} = \int dU_{\mathfrak{b}} \, S(BR)_{V U_{\mathfrak{b}} W_{\bar{\mathfrak{b}}} |\Psi\rangle}, \tag{92}$$

where the inequality follows from $S\left(\int dU \rho(U)\right) \geq \int dU S(\rho(U))$ and the last equality follows from the purity of $V U_{\mathfrak{b}} W_{\bar{\mathfrak{b}}} |\Psi\rangle$. Because of the strict concavity of von Neumann entropy, this inequality is saturated if and only if the state

$$\rho_{\overline{B}\,\overline{R}\,\bar{\mathbf{a}}\bar{\mathbf{r}}}(U_{\mathfrak{b}}) := \mathrm{tr}_{BR}\left[V U_{\mathfrak{b}} W_{\bar{\mathfrak{b}}} |\Psi\rangle\langle\Psi| W_{\bar{\mathfrak{b}}}^{\dagger} U_{\mathfrak{b}}^{\dagger} V^{\dagger}\right] \tag{93}$$

is independent of $U_{\mathfrak{b}}$. Now note that

$$\begin{aligned}
\rho_{BR}(U_{\mathfrak{b}}) &= \mathrm{tr}_{\overline{B}\,\overline{R}U_{\bar{\mathfrak{b}}}}\left[V U_{\mathfrak{b}} W_{\bar{\mathfrak{b}}} |\Psi\rangle\langle\Psi| W_{\bar{\mathfrak{b}}}^{\dagger} U_{\mathfrak{b}}^{\dagger} V^{\dagger}\right] \\
&= \int dU'_{\bar{\mathfrak{b}}} dU''_{\bar{\mathfrak{b}}} \,\mathrm{tr}_{\overline{B}\,\overline{R}}\left[V U_{\mathfrak{b}} U'_{\bar{\mathfrak{b}}} |\Psi\rangle\langle\Psi| U''^{\dagger}_{\bar{\mathfrak{b}}} U_{\mathfrak{b}}^{\dagger} V^{\dagger}\right] \langle U''_{\bar{\mathfrak{b}}}|U'_{\bar{\mathfrak{b}}}\rangle \\
&= \int dU'_{\bar{\mathfrak{b}}} \,\mathrm{tr}_{\overline{B}\,\overline{R}}\left[V U_{\mathfrak{b}} U'_{\bar{\mathfrak{b}}} |\Psi\rangle\langle\Psi| U'^{\dagger}_{\bar{\mathfrak{b}}} U_{\mathfrak{b}}^{\dagger} V^{\dagger}\right].
\end{aligned} \tag{94}$$

Therefore,

$$S(BR)_{V U_{\mathfrak{b}} W_{\bar{\mathfrak{b}}} |\Psi\rangle} \geq \int dU'_{\bar{\mathfrak{b}}} S(BR)_{V U_{\mathfrak{b}} U'_{\bar{\mathfrak{b}}} |\Psi\rangle}. \tag{95}$$

Finally, we know from our assumption of Condition (2) that

$$S(BR)_{VU_{\mathbf{b}}U'_{\bar{\mathbf{b}}}|\Psi\rangle} = A_B(\mathbf{b}) + S(\mathbf{b}R)_{|\Psi\rangle}, \tag{96}$$

for all $U_{\mathbf{b}}, U'_{\bar{\mathbf{b}}}$. Combining everything together, we find

$$
\begin{aligned}
A_B(\mathbf{b}) + S(\mathbf{b}R)_{|\Psi\rangle} = S(\overline{B}\,\overline{R}\,\overline{\mathbf{a}}\overline{\mathbf{r}})_{VW_b W_{\bar{b}}|\Psi\rangle} &\geq \int dU_{\mathbf{b}} S(BR)_{VU_{\mathbf{b}}W_{\bar{b}}|\Psi\rangle} \\
&\geq \int dU_{\mathbf{b}} dU'_{\bar{\mathbf{b}}} S(BR)_{VU_{\mathbf{b}}U'_{\bar{\mathbf{b}}}|\Psi\rangle} \\
&= A_B(\mathbf{b}) + S(\mathbf{b}R)_{|\Psi\rangle}.
\end{aligned}
\tag{97}
$$

Both inequalities must be saturated and so we see that a) $\rho_{\overline{B}\overline{R}\overline{\mathbf{a}}\overline{\mathbf{r}}}(U_{\mathbf{b}})$ from (93) is independent of $U_{\mathbf{b}}$ and b) for any fixed $U_{\mathbf{b}}$,

$$\sigma_{BR}(U_{\mathbf{b}}, U'_{\bar{\mathbf{b}}}) := \text{tr}_{\overline{B}\overline{R}}\left[VU_{\mathbf{b}}U'_{\bar{\mathbf{b}}}|\Psi\rangle\langle\Psi|U'^{\dagger}_{\bar{\mathbf{b}}}U^{\dagger}_{\mathbf{b}}V^{\dagger}\right] \tag{98}$$

is independent of $U'_b$.

Since all purifications of a fixed density matrix are related by a unitary on the purifying system, we know from (a) that for all $U_{\mathbf{b}}$, there exists a unitary $U_{BR}$ such that

$$U_{BR}VW_{\bar{\mathbf{b}}}|\Psi\rangle = VU_{\mathbf{b}}W_{\bar{\mathbf{b}}}|\Psi\rangle, \tag{99}$$

and thus for any $U'_{\bar{\mathbf{b}}}$

$$\langle U'_{\bar{\mathbf{b}}}|_{U_{\bar{\mathbf{b}}}} U_{BR}VW_{\bar{\mathbf{b}}}|\Psi\rangle = U_{BR}VU'_{\bar{\mathbf{b}}}|\Psi\rangle = VU_{\mathbf{b}}U'_{\bar{\mathbf{b}}}|\Psi\rangle. \tag{100}$$

Similarly, condition (b) implies that for all $U'_{\bar{\mathbf{b}}}$, there exists a $U'_{\overline{B}\overline{R}}$ such that

$$VU'_{\bar{\mathbf{b}}}|\Psi\rangle = U'_{\overline{B}\overline{R}}V|\Psi\rangle. \tag{101}$$

Together, (100) and (101) tell us that for all $U_b, U'_{\bar{b}}$,

$$U_{BR}U'_{\overline{B}\overline{R}}V|\Psi\rangle = VU_{\mathbf{b}}U'_{\bar{\mathbf{b}}}|\Psi\rangle, \tag{102}$$

which is exactly Condition 1.

**Proof** (1) $\implies$ (3):

We will need the following lemmas:

**Lemma 4.6.** *(Corollary 5.9 of [37]) For any product state $\rho_{AB} \otimes \sigma_{A'B'}$,*

$$H_{\min}(AA'|BB')_{\rho\otimes\sigma} = H_{\min}(A|B)_{\rho} + H_{\min}(A'|B')_{\sigma}. \tag{103}$$

*Proof.* For any state $\rho_{AB}$, the min-entropy and max-entropy are defined respectively by

$$H_{\min}(A|B) = -\min_{\sigma_B} D_{\max}(\rho_{AB}|\mathbb{1}_A \otimes \sigma_B), \tag{104}$$

$$H_{\max}(A|B) = -\min_{\sigma_B} D_{\min}(\rho_{AB}|\mathbb{1}_A \otimes \sigma_B), \tag{105}$$

where

$$D_{\max}(\rho|\sigma) := \inf\{\lambda : \rho \leq e^{\lambda}\sigma\}, \tag{106}$$

$$D_{\min}(\rho|\sigma) := -\log\left(F(\rho,\sigma)^2\right), \tag{107}$$

where $F(\rho, \sigma) = \|\sqrt{\rho}\sqrt{\sigma}\|_1$ is the fidelity. It follows that

$$
\begin{aligned}
H_{\min/\max}(AA'|BB')_{\rho\otimes\sigma} &= -\min_{\omega_{BB'}} D_{\max/\min}(\rho_{AB}\otimes\sigma_{A'B'}||\mathbb{1}_{AA'}\otimes\omega_{BB'}) \\
&\geq -\min_{\omega_B,\omega'_{B'}} D_{\max/\min}(\rho_{AB}\otimes\sigma_{A'B'}||\mathbb{1}_A\otimes\omega_B\otimes\mathbb{1}_{A'}\otimes\omega'_{B'}) \\
&= H_{\min/\max}(A|B) + H_{\min/\max}(A'|B').
\end{aligned}
\tag{108}
$$

For the other inequality, introduce $\rho_{ABC}$ (resp. $\sigma_{A'B'C'}$) as the purification of $\rho_{AB}$ (resp. $\sigma_{A'B'}$). From the above inequality, we have

$$
H_{\max}(AA'|CC')_{\rho\otimes\sigma} \geq H_{\max}(A|C)_\rho + H_{\max}(A'|C')_\sigma.
\tag{109}
$$

For any tripartite pure state on $XYZ$, it holds that $H_{\min}(X|Y) + H_{\max}(X|Z) = 0$. It follows that

$$
H_{\min}(AA'|BB')_{\rho\otimes\sigma} \leq H_{\min}(A|B) + H_{\min}(A'|B').
\tag{110}
$$

Combining the two inequalities concludes the proof. $\qquad\square$

**Lemma 4.7.** *Consider the state* $|CJ\rangle_{B\bar{B}\mathbf{r}\bar{\mathbf{r}}} \otimes |\Psi\rangle_{\mathbf{a}\bar{\mathbf{a}}R\bar{R}}$. *Let* $\bar{\mathbf{b}}' \subseteq \bar{\mathbf{b}}$ *(and let* $\bar{\mathbf{a}}' = \{a_i : b_i \in \bar{\mathbf{b}}'\}$ *and* $\bar{\mathbf{r}}' = \{r_i : b_i \in \bar{\mathbf{b}}'\}$). *Then*

$$
H_{\min}(\bar{\mathbf{a}}'\bar{\mathbf{r}}'|BR\,\mathbf{ar})_{|CJ\rangle|\Psi\rangle} \geq 0 \implies H_{\min}(\bar{\mathbf{b}}'|\mathbf{b}R)_{|\Psi\rangle} \geq A_B(\mathbf{b}) - A_B(\bar{\mathbf{b}}' \cup \mathbf{b}).
\tag{111}
$$

*Proof.* From Lemma 4.6, we know that

$$
H_{\min}(\bar{\mathbf{a}}'\bar{\mathbf{r}}'|BR\,\mathbf{ar})_{|CJ\rangle|\Psi\rangle} = H_{\min}(\bar{\mathbf{r}}'|B\,\mathbf{r})_{|CJ\rangle} + H_{\min}(\bar{\mathbf{b}}'|\mathbf{b}R)_{|\Psi\rangle}.
\tag{112}
$$

Moreover,

$$
H_{\min}(\bar{\mathbf{r}}'|B\,\mathbf{r})_{|CJ\rangle} \leq S(\bar{\mathbf{r}}'|B\,\mathbf{r})_{|CJ\rangle} = A_B(\bar{\mathbf{b}}' \cup \mathbf{b}) - A_B(\mathbf{b}).
\tag{113}
$$

Combining these completes the proof. $\qquad\square$

**Lemma 4.8.** *The following inequality holds for any state* $|\psi\rangle \in \mathcal{H}_A \otimes \mathcal{H}_B$ *and Haar random unitaries* $U_A$ *acting on* $\mathcal{H}_A$

$$
\int dU_A dU'_A \, |U_A\rangle\langle U'_A|_{\mathbf{U}_A} \otimes U_A|\psi\rangle\langle\psi|_{AB} U_A'^\dagger \leq \int dU_A |U_A\rangle\langle U_A|_{\mathbf{U}_A} \otimes U_A|\psi\rangle\langle\psi|_{AB} U_A^\dagger.
\tag{114}
$$

*Proof.* The left-hand side is a rank one projector onto the state

$$
W_A|\psi\rangle = \int dU_A |U_A\rangle \otimes U_A|\psi\rangle_{AB}.
\tag{115}
$$

Hence it suffices to show that the right-hand side is also a projector whose support includes $W_A|\psi\rangle$. To see that it is a projector, note that

$$
\left(\int dU_A |U_A\rangle\langle U_A| \otimes U_A|\psi\rangle\langle\psi|U_A^\dagger\right)^2 = \int dU_A |U_A\rangle\langle U_A| \otimes U_A|\psi\rangle\langle\psi|U_A^\dagger.
\tag{116}
$$

Its support includes $W_A|\psi\rangle$ because

$$
\left(\int dU'_A |U'_A\rangle\langle U'_A| \otimes U'_A|\psi\rangle\langle\psi|U_A'^\dagger\right) W_A|\psi\rangle = W_A|\psi\rangle.
\tag{117}
$$

This concludes the proof. $\qquad\square$

We are now ready to prove that Condition 1 implies Condition 3. We want to show that, assuming Condition 1, then for all $\bar{\mathbf{b}}' \subseteq \bar{\mathbf{b}}$,

$$H_{\min}(\bar{\mathbf{a}}'\bar{\mathbf{r}}'|B R\,\mathbf{ar})_{V W_{\mathbf{b}} W_{\bar{\mathbf{b}}}|\Psi\rangle} \geq 0\,. \tag{118}$$

From this (55) follows immediately via Lemma 4.7 and (89). Similarly, if we can show that, for all $\mathbf{b}' \subseteq \mathbf{b}$

$$H_{\min}(\mathbf{a}'\mathbf{r}'|\overline{B}\,\overline{R}\,\bar{\mathbf{a}}\bar{\mathbf{r}})_{V W_{\mathbf{b}} W_{\bar{\mathbf{b}}}|\Psi\rangle} \geq 0\,, \tag{119}$$

then (56) follows immediately via Lemma 4.7 with all complementary Hilbert spaces exchanged (so for example $\mathcal{H}_{\mathbf{b}} \leftrightarrow \mathcal{H}_{\bar{\mathbf{b}}}$, $\mathcal{H}_B \leftrightarrow \mathcal{H}_{\overline{B}}$ and so on). We shall focus on (118), since the proof of (119) is completely identical up to the relabelling of Hilbert spaces.

By definition, for any state $\rho_{AB}$

$$\begin{aligned} H_{\min}(A|B) &= -\min_{\sigma_B} D_{\max}(\rho_{AB}|\mathbb{1}_A \otimes \sigma_B) \\ &\geq -D_{\max}(\rho_{AB}|\mathbb{1}_A \otimes \rho_B)\,, \end{aligned} \tag{120}$$

where

$$D_{\max}(\rho|\sigma) = \inf\{\lambda : \rho \leq e^{\lambda}\sigma\}\,. \tag{121}$$

Therefore, in order to show that $H_{\min}(A|B)_\rho \geq 0$, it is sufficient to show $\rho_{AB} \leq \mathbb{1}_A \otimes \rho_B$.

Using Condition 1, we can write

$$\begin{aligned} |\phi\rangle &:= |\mathrm{CJ}\rangle_{B\overline{B}\mathbf{r}\bar{\mathbf{r}}} \otimes |\Psi\rangle_{\mathbf{a}\bar{\mathbf{a}}R\overline{R}} \\ &= V W_{\mathbf{b}} W_{\bar{\mathbf{b}}} |\Psi\rangle_{\mathbf{b}\bar{\mathbf{b}}R\overline{R}} \\ &= W_{BR} W_{\overline{B}\overline{R}} V |\Psi\rangle_{\mathbf{b}\bar{\mathbf{b}}R\overline{R}}\,. \end{aligned} \tag{122}$$

Let $\bar{\mathbf{b}}'' = \bar{\mathbf{b}} \setminus \bar{\mathbf{b}}'$ be the complement of $\bar{\mathbf{b}}'$ in $\bar{\mathbf{b}}$, and similarly $\bar{\mathbf{a}}'' = \bar{\mathbf{a}} \setminus \bar{\mathbf{a}}'$ and $\bar{\mathbf{r}}'' = \bar{\mathbf{r}} \setminus \bar{\mathbf{r}}'$. Taking a partial trace over $\mathcal{H}_{\overline{B}} \otimes \mathcal{H}_{\overline{R}} \otimes \mathcal{H}_{\bar{\mathbf{a}}''} \otimes \mathcal{H}_{\bar{\mathbf{r}}''}$, we then find

$$\begin{aligned} \phi_{BR\mathbf{ar}\bar{\mathbf{a}}'\bar{\mathbf{r}}'} &:= \operatorname{tr}_{\overline{B}\overline{R}\bar{\mathbf{a}}''\bar{\mathbf{r}}''} |\phi\rangle \langle\phi| \\ &= \operatorname{tr}_{\overline{B}\overline{R}\mathbf{U}_{\bar{\mathbf{b}}''}} \left[ W_{BR} W_{\overline{B}\overline{R}} V |\Psi\rangle \langle\Psi| V^{\dagger} W_{\overline{B}\overline{R}}^{\dagger} W_{BR}^{\dagger} \right] \\ &= \int dU_{\bar{\mathbf{b}}} dU_{\bar{\mathbf{b}}}' \, \langle U_{\bar{\mathbf{b}}''}'|U_{\bar{\mathbf{b}}''}\rangle W_{BR} \operatorname{tr}_{\overline{B}\overline{R}} \left[ U_{\overline{B}\overline{R}} V |\Psi\rangle \langle\Psi| V^{\dagger} U_{\overline{B}\overline{R}}'^{\dagger} \right] W_{BR}^{\dagger} \otimes \Pi_{\mu_0} |U_{\bar{\mathbf{b}}'}\rangle \langle U_{\bar{\mathbf{b}}'}'| \Pi_{\mu_0} \\ &\leq W_{BR} \operatorname{tr}_{\overline{B}\overline{R}} \left[ V |\Psi\rangle \langle\Psi| V^{\dagger} \right] W_{BR}^{\dagger} \otimes \int dU_{\bar{\mathbf{b}}'} \Pi_{\mu_0} |U_{\bar{\mathbf{b}}'}\rangle \langle U_{\bar{\mathbf{b}}'}| \Pi_{\mu_0} \\ &\leq \phi_{BR\mathbf{ar}} \otimes \mathbb{1}_{\bar{\mathbf{a}}'\bar{\mathbf{r}}'}\,. \end{aligned} \tag{123}$$

In the third line, we expanded $W_{\overline{B}\overline{R}}$, including explicitly expanding the product unitaries $U_{\bar{\mathbf{b}}} = U_{\bar{\mathbf{b}}'} \otimes U_{\bar{\mathbf{b}}''}$. We also used Lemma 4.4 to introduce projectors $\Pi_{\mu_0}$ onto the fundamental representation block $\mathcal{H}_{\bar{\mathbf{a}}'} \otimes \mathcal{H}_{\bar{\mathbf{r}}'}$ of $\mathcal{H}_{\mathbf{U}_{\bar{\mathbf{b}}'}}$ without affecting the state. The inequality in the fourth line then follows from Lemma 4.8. This concludes the proof, since by the discussion above, (123) implies (118) and hence (55).

**Remark 4.9.** This proof easily extends to a slightly stronger statement than (55). In (113), we replaced a min-entropy in the $|\mathrm{CJ}\rangle$ state with a conditional entropy, so that we could write it as a difference in two areas. Had we left it in terms of the min-entropy, we would have ended up with a stronger a statement, a bound on the min-entropy of the bulk state by the min-entropy in the $|\mathrm{CJ}\rangle$ state. One possible physical interpretation of this $|\mathrm{CJ}\rangle$ min-entropy is a bound on the *fluctuations* in the area difference. The stronger version of (55) would then lower bound $H_{\min}(\bar{\mathbf{b}}'|\mathbf{b}R)_{|\Psi\rangle}$ by an upper bound on the area-difference, rather than the average area difference.

**Proof** (3) $\implies$ (4):

This was previously proven in [11]. Recall that the min-entropy and max-entropy satisfy strong sub-additivity [38],

$$H_{\min/\max}(A|B) \geq H_{\min/\max}(A|BC). \tag{124}$$

Consider an arbitrary subset $\mathbf{b}' \subseteq \{b_1 \dots b_n\}$. We want to show that

$$A_B(\mathbf{b}') + S(\mathbf{b}'R)_{|\Psi\rangle} \geq A_B(\mathbf{b}) + S(\mathbf{b}R)_{|\Psi\rangle}. \tag{125}$$

By the assumption of Condition 3,

$$A_B(\mathbf{b} \cap \mathbf{b}') - A_B(\mathbf{b}) \geq H_{\max}(\mathbf{b} \setminus \mathbf{b}'|[\mathbf{b} \cap \mathbf{b}']R)_{|\Psi\rangle}. \tag{126}$$

Here we combined (56) with the duality identity $H_{\max}(A|B) = -H_{\min}(A|C)$ for any tripartite pure state, and the equality between complementary areas described in Remark 2.2. By strong subadditivity,

$$\begin{aligned}
A_B(\mathbf{b}') - A_B(\mathbf{b} \cup \mathbf{b}') &= -S(\mathbf{r} \setminus \mathbf{r}'|B\,\mathbf{r}')_{|\mathrm{CJ}\rangle} \\
&\geq -S(\mathbf{r} \setminus \mathbf{r}'|B\,\mathbf{r} \cap \mathbf{r}')_{|\mathrm{CJ}\rangle} \\
&= A_B(\mathbf{b} \cap \mathbf{b}') - A_B(\mathbf{b}),
\end{aligned} \tag{127}$$

and

$$H_{\max}(\mathbf{b} \setminus \mathbf{b}'|[\mathbf{b} \cap \mathbf{b}']R)_{|\Psi\rangle} \geq H_{\max}(\mathbf{b} \setminus \mathbf{b}'|\mathbf{b}'R)_{|\Psi\rangle} \geq S([\mathbf{b} \cup \mathbf{b}']R)_{|\Psi\rangle} - S(\mathbf{b}'R)_{|\Psi\rangle}, \tag{128}$$

where we also used $H_{\max}(A|B) \geq S(A|B)$. Hence

$$A_B(\mathbf{b}') - A_B(\mathbf{b} \cup \mathbf{b}') \geq S([\mathbf{b} \cup \mathbf{b}']R)_{|\Psi\rangle} - S(\mathbf{b}'R)_{|\Psi\rangle}. \tag{129}$$

Meanwhile, (55) tells us

$$A_B(\mathbf{b}) - A_B(\mathbf{b} \cup \mathbf{b}') \leq H_{\min}(\mathbf{b}' \setminus \mathbf{b}|\mathbf{b}) \leq S([\mathbf{b}' \cup \mathbf{b}]R) - S(\mathbf{b}R), \tag{130}$$

where in the second inequality we used $H_{\min}(A|B) \leq S(A|B)$. Combining (129) and (130) gives (125), which is what we set out to show.

**Remark 4.10.** One can also more directly prove Condition 4 from Condition 1, without any reference to min- and max-entropies. By an analogue of Lemma 4.7, together with the strong sub-additivity arguments above, it is sufficient to show that

$$S(\bar{\mathbf{a}}'\bar{\mathbf{r}}'|\overline{B}\,\overline{R}\,\bar{\mathbf{a}}''\bar{\mathbf{r}}'')_{|\mathrm{CJ}\rangle|\Psi\rangle} = -S(\bar{\mathbf{a}}'\bar{\mathbf{r}}'|B\,R\,\mathbf{a}\mathbf{r})_{|\mathrm{CJ}\rangle|\Psi\rangle} \leq 0. \tag{131}$$

But, if $|\phi\rangle = |\mathrm{CJ}\rangle\,|\Psi\rangle$ as before,

$$\begin{aligned}
\phi_{\overline{B}\,\overline{R}\,\bar{\mathbf{a}}''\bar{\mathbf{r}}''} &= \int dU_{\bar{\mathbf{b}}}\, dU'_{\bar{\mathbf{b}}}\, \langle U'_{\bar{\mathbf{b}}'}|U_{\bar{\mathbf{b}}'}\rangle\, U_{\overline{B}\,\overline{R}}\, \mathrm{tr}_{BR}\left[V\,|\Psi\rangle\,\langle\Psi|\,V^{\dagger}\right] U'^{\dagger}_{\overline{B}\,\overline{R}} \otimes |U_{\bar{\mathbf{b}}''}\rangle\,\langle U'_{\bar{\mathbf{b}}''}| \\
&= \int dU_{\bar{\mathbf{b}}'}\, W^{\bar{\mathbf{b}}''}_{\overline{B}\,\overline{R}}(U_{\bar{\mathbf{b}}'})\, \mathrm{tr}_{BR}\left[\left[V\,|\Psi\rangle\,\langle\Psi|\,V^{\dagger}\right] W^{\bar{\mathbf{b}}''}_{\overline{B}\,\overline{R}}(U_{\bar{\mathbf{b}}'})^{\dagger}\right].
\end{aligned} \tag{132}$$

Here we have defined the isometry

$$W^{\bar{\mathbf{b}}''}_{\overline{B}\,\overline{R}}(U_{\bar{\mathbf{b}}'}) = \int dU_{\bar{\mathbf{b}}''}\, U_{\overline{B}\,\overline{R}} \otimes |U_{\bar{\mathbf{b}}''}\rangle, \tag{133}$$

which depends implicitly on $U_{\bar{\mathbf{b}}'}$ through $U_{\overline{B}\,\overline{R}}$. By the concavity of the von Neumann entropy, and its invariance under isometries, we therefore have

$$S(\overline{B}\,\overline{R}\,\bar{\mathbf{a}}''\bar{\mathbf{r}}'')_{|\mathrm{CJ}\rangle|\psi\rangle} \geq S(\overline{B}\,\overline{R})_{V|\psi\rangle} = A_B(\mathbf{b}) + S(\mathbf{b})_{|\Psi\rangle} = S(\overline{B}\,\overline{R}\,\bar{\mathbf{a}}\bar{\mathbf{r}})_{|\mathrm{CJ}\rangle|\Psi\rangle}, \tag{134}$$

as desired.



# 5 Approximate and non-isometric codes

## 5.1 Error correction for non-isometric codes

In traditional applications of quantum error correcting codes, the map $V$ is always an isometry $V^\dagger V = \mathbb{1}$. This is for obvious physical reasons: time evolution in quantum mechanics is unitary, and so any physical process that encodes a quantum state into a quantum code must be isometric.[24]

This does not seem to be the case, however, in holographic codes, where the map $V$ has a very different physical interpretation as the "bulk-to-boundary map," relating the semiclassical description of the bulk state to the corresponding holographic boundary state. Because the bulk has (at least) one more dimension than the boundary, a naive counting suggests that there are in fact many more bulk states than there are boundary states!

Of course, if the bulk geometry is perturbatively close to vacuum AdS, we cannot excite too many bulk degrees of freedom without causing gravitational backreaction that creates a black hole.[25] However, there is no such limit within the interior of a black hole. Instead, the number of semiclassical degrees of freedom in a long wormhole behind a black hole horizon can be arbitrarily large compared to the Bekenstein-Hawking entropy – the classic example of this is an evaporating black hole after the Page time.

Traditionally, this fact has been regarded as unacceptable, and possibly as evidence for new physics that avoids the excess semiclassical degrees of freedom [39–41]. However, recent progress in understanding the black hole information paradox [9,10] has made it clear that the semiclassical description of an black hole needs to be taken seriously, even after the Page time. Instead, the apparent paradox is resolved by nonperturbative corrections to the semiclassical inner product: exactly orthogonal simple states in the bulk quantum field theory have exponentially small, but nonzero, overlap in quantum gravity [42]. Another way of saying this is that while the bulk-to-boundary map $V$ is still a linear map – the boundary dual of a superposition of semiclassical bulk states is just a superposition of the dual boundary states – it does not preserve the inner product and so is not an isometry. As a result, the arbitrarily large Hilbert space of semiclassical bulk states can be mapped by $V$ into a much smaller Hilbert space of holographic boundary states, the dimension of which is controlled by the Bekenstein-Hawking entropy.

It may seem counterintuitive, or even impossible, for $V$ to approximately preserve the inner products of all pairs of simple states – for concreteness we can take these to be product states on a large number of qubits – without $V$ being (approximately) an isometry. However, as the following theorem shows, not only in this possible, it is in fact generically true for random maps $V$. The probability that any inner product is not preserved is actually doubly exponentially suppressed.

**Theorem 5.1.** *Let $V : [\mathbb{C}^2]^{\otimes n} \to [\mathbb{C}^2]^{\otimes m}$ with $n > m > 1$ be defined by $V = 2^{(n-m)/2} \langle 0|^{\otimes(n-m)} U$ where $U$ is a Haar random unitary, and let $\varepsilon < 1$ be an arbitrary positive number. Then, with probability*

$$p \geq 1 - 2^{6n(n-m)+30n+4} n^{6n} \exp[-\frac{2^{m-5}\varepsilon^2}{81\pi^3}], \tag{135}$$

---

[24]The most natural physical interpretation of a non-isometric map $V$ in this setting is a QEC code that relies on postselection onto a particular measurement outcome to succeed. Since the outcome of a measurement cannot be predicted in advance, such codes would not be useful in practice.

[25]It is also worth noting that the negative curvature of AdS-space mean that the volume and surface area of a sphere grow at the same rate in the large radius limit.

*for all product states $|\psi\rangle = |\psi_1\rangle |\psi_2\rangle \dots |\psi_n\rangle$ and $|\phi\rangle = |\phi_1\rangle |\phi_2\rangle \dots |\phi_n\rangle$, we have*

$$\left| \langle \phi | V^\dagger V |\psi\rangle - \langle \phi |\psi\rangle \right| \le \varepsilon. \tag{136}$$

*Proof.* See Appendix A.4. □

However, this story is impossible to understand in terms of state-independent quantum error correction. You cannot use $m$ qubits to encode $n > m$ qubits in a state-independent way; there simply aren't enough degrees of freedom. Fortunately, state-specific reconstruction offers a resolution. The bulk Hilbert space can be larger than the boundary Hilbert space and nonetheless be encoded into it as a state-specific code, with all of the desirable features of holography.

A simple example of such a state-specific code is given by Theorem 5.1: given a product state $|\psi\rangle$ and a map $V : \mathcal{H}_{\text{code}} \to \mathcal{H}_B$ defined as in Theorem 5.1 – $\mathcal{H}_{\overline{B}}$ is trivial in this case – we can implement any product unitary acting on $|\psi\rangle$, in a state-specific way, by acting with a unitary on $\mathcal{H}_B$, because all product states have approximately the same norm.

**Remark 5.2.** While in Theorem 5.1 the state-specific reconstruction is only approximate, it is easy to construct other examples where exact state-specific reconstruction is possible for highly non-isometric codes. A simple example is a code space consisting of two qubits, with $|\psi\rangle$ maximally entangled, and the map $V$ projecting the first qubit into the state $|0\rangle$ (and rescaling by $\sqrt{2}$).

An alternative approach to this problem is to only consider a small subspace of bulk states, so that the restriction of the bulk-to-boundary map $V$ to that subspace is indeed (approximately) an isometry and traditional definitions of quantum error correction are applicable. This is of course completely valid as far as it goes, but it means that one can only treat a small number of bulk degrees of freedom as "real" in any given calculation, even though all the other bulk degrees of freedom are obviously necessary, for example in order to correctly apply the QES prescription. Using the framework of state-specific reconstruction, one can correctly understand how the entire bulk Hilbert space is encoded, and also how state-independent reconstruction emerges upon restriction to sufficiently small subspaces of states.

The obvious question is whether complementary state-specific codes with non-isometric maps $V$ obey a version of Theorem 4.2. The answer to that question will be the subject of Section 5.2, but the simple answer is that they do; in fact the basic proof strategy from Section 4.2 goes through essentially unchanged.

## 5.2 Theorem Statement

In this section, we extend Theorem 4.2 in two important and complementary ways. Firstly, we allow arbitrary linear maps $V$ that are not necessarily isometries, so long as $V$ does not increase the norm of any state $U |\psi\rangle$ related to $|\Psi\rangle$ by a product unitary $U$. (In practice $\|V U |\psi\rangle\|$ will need to be approximately independent of the product unitary $U$.) Secondly, we allow the existence of small corrections to each of the conditions in the theorem. As emphasized in Section 3, this is both necessary to correctly capture the physics of holography – nonperturbative corrections in quantum gravity will always prevent perfect reconstruction – and also appears to be important if we want to actually find interesting and nontrivial examples of state-specific codes.

The essence of the proof of Theorem 5.5 is the same as the proof of Theorem 4.2. However, the necessity of keeping track of the various epsilons makes it noticeably more technical. As such, we postpone the proof to Appendix C, and content ourselves for the moment with stating the theorem in full and then making a few brief remarks.

Before stating the theorem, however, we need to define the smooth min-entropy $H^{\varepsilon}_{\min}(A|B)_{|\psi\rangle}$. To do so, we first need to generalize the fidelity to subnormalized states:

**Definition 5.3** (Generalized Fidelity). For positive semidefinite operators $\rho, \sigma$ on Hilbert space $\mathcal{H}$ satisfying $\operatorname{tr}\rho, \operatorname{tr}\sigma \leq 1$, the *generalized fidelity* $\bar{F}(\rho, \sigma)$ is given by

$$\bar{F}(\rho, \sigma) := \sup_{\mathcal{H}'} \sup_{\bar{\rho},\bar{\sigma}} F(\bar{\rho}, \bar{\sigma}), \tag{137}$$

where the supremum is taken over all isometries $V' : \mathcal{H} \to \mathcal{H}'$ of $\mathcal{H}$ into a larger Hilbert space $\mathcal{H}'$ and over (normalized) density matrices $\bar{\rho}, \bar{\sigma}$ such that $\rho = V'^{\dagger}\bar{\rho}V'$ and $\sigma = V'^{\dagger}\bar{\sigma}V'$.

**Definition 5.4** (Smooth min-entropy). The smooth min-entropy $H^{\varepsilon}_{\min}(A|B)_{|\psi\rangle}$ is defined as

$$H^{\varepsilon}_{\min}(A|B)_{|\psi\rangle} = \sup_{\tilde{\rho}_{AB}} H^{\varepsilon}_{\min}(A|B)_{\tilde{\rho}_{AB}}, \tag{138}$$

where the supremum is over subnormalized density matrices $\tilde{\rho}_{AB}$ such that the generalized fidelity $\bar{F}(\tilde{\rho}_{AB}, \psi_{AB}) \geq \sqrt{1-\varepsilon^2}$.

**Theorem 5.5.** *Let $V : \mathcal{H}_{\mathrm{code}} \cong \otimes_i \mathcal{H}_{b_i} \to \mathcal{H}_B \otimes \mathcal{H}_{\bar{B}}$ be a linear map, with $\mathbf{b} = \{b_{i_1}, b_{i_2}...\}$ a subset of input legs, and $\bar{\mathbf{b}}$ its complement. Let $U_{\mathbf{b}}, \hat{U}_{\mathbf{b}}$ (respectively $U'_{\bar{\mathbf{b}}}, \hat{U}'_{\bar{\mathbf{b}}}$) be a product of local unitaries on $\mathbf{b}$ (respectively on $\bar{\mathbf{b}}$) chosen at random according to the Haar measure. Finally, let $|\Psi\rangle \in \mathcal{H}_{\mathrm{code}} \otimes \mathcal{H}_R \otimes \mathcal{H}_{\bar{R}}$ be a fixed, arbitrary state with $\mathcal{H}_R, \mathcal{H}_{\bar{R}}$ reference systems of arbitrary dimension such that for all product unitaries $U_{\mathbf{b}}, U'_{\bar{\mathbf{b}}}$, we have $\|VU_{\mathbf{b}}U'_{\bar{\mathbf{b}}}|\Psi\rangle\| \leq 1$.*

*Then the following two conditions are equivalent:*

1. *(Complementary Recovery) With probability $p \geq 1-\kappa_1$, there exist unitary operators $U_{BR}$, depending only on $U_{\mathbf{b}}, \hat{U}_{\mathbf{b}}$, and $U'_{\bar{B}\bar{R}}$, depending only on $U'_{\bar{\mathbf{b}}}, \hat{U}'_{\bar{\mathbf{b}}}$, such that*

$$\left\| U_{BR}U'_{\bar{B}\bar{R}}V\hat{U}_{\mathbf{b}}\hat{U}'_{\bar{\mathbf{b}}}|\Psi\rangle - VU_{\mathbf{b}}U'_{\bar{\mathbf{b}}}|\Psi\rangle \right\| \leq \varepsilon_1, \tag{139}$$

2. *(Holographic Entropy Prescription) With probability $p \geq 1-\kappa_2$,*

$$\left| S(BR)_{V\hat{U}_{\mathbf{b}}\hat{U}'_{\bar{\mathbf{b}}}|\Psi\rangle} - \left[ A_B(\mathbf{b}) + S(\mathbf{b}R)_{|\Psi\rangle} \right] \right| \leq \varepsilon_2. \tag{140}$$

*Moreover, both statements imply:*

3. *(One-shot Minimality) For all $\bar{\mathbf{b}}' \subseteq \bar{\mathbf{b}}$ and $\mathbf{b}' \subseteq \mathbf{b}$,*

$$H^{\varepsilon_3}_{\min}(\bar{\mathbf{b}}'|\mathbf{b}R)_{|\Psi\rangle} \geq A_B(\mathbf{b}) - A_B(\bar{\mathbf{b}}'\mathbf{b}), \tag{141}$$

$$H^{\varepsilon_3}_{\min}(\mathbf{b}'|\bar{\mathbf{b}}\bar{R})_{|\Psi\rangle} \geq A_{\bar{B}}(\bar{\mathbf{b}}) - A_{\bar{B}}(\mathbf{b}'\bar{\mathbf{b}}). \tag{142}$$

*This in turn implies:*

4. *(Minimality) For all $\mathbf{b}' \subseteq \mathbf{b} \cup \bar{\mathbf{b}}$,*

$$A_B(\mathbf{b}') + S(\mathbf{b}'R)_{|\Psi\rangle} \geq A_B(\mathbf{b}) + S(\mathbf{b}R)_{|\Psi\rangle} - \varepsilon_4. \tag{143}$$

*Specifically, for sufficiently small $\varepsilon_1, \kappa_1$, Condition 1 implies Condition 2 for any $\kappa_2 \gg \kappa_1$ with $\varepsilon_2 \leq \sqrt{\varepsilon_1^2 + \kappa_1/\kappa_2} \log[d_B^2 d_R^2/4(\varepsilon_1^2 + \kappa_1/\kappa_2)]$. Condition 2 implies Condition 1 for any $\kappa_1 > 0$ with $\varepsilon_1 \leq (16/\kappa_1)[8\kappa_2 \log(d_B d_R) + 8\varepsilon_2]^{1/4}$. Condition 1 implies Condition 3 with $\varepsilon_3 \leq \sqrt{2\varepsilon_1 + 2\kappa_1}$. Finally, for any small $\varepsilon_3 \geq 0$, Condition 1 implies Condition 4 with $\varepsilon_4 \leq 4\varepsilon_3 \log[d_{\mathrm{code}}^2 d_R d_{\bar{R}}/(4\varepsilon_3^2)]$.*

**Remark 5.6.** Some of the implications in Theorem 5.5 involve unavoidable factors of $\log d$ for various Hilbert space dimensions $d$. This is in contrast to the proofs in Section 3, where all bounds were universal, with no such factors. At first glance, this may appear somewhat problematic, since in Section 3 we emphasized that zero-bit codes and state-independent QEC codes are inequivalent because the corresponding errors can differ by a factor of $d_{\text{code}}$. The difference, however, is that $d \gg \log d$. In holographic codes, we have $\log d = O(1/G)$, whereas the reconstruction errors $\varepsilon$ can be made nonpeturbatively small in $G$. Hence, polylogarithmic factors of Hilbert space dimensions will still leave the errors nonperturbatively small. Polynomial factors on the other hand can lead to large errors.

**Remark 5.7.** Unlike in Theorem 4.2, in Theorem 5.5 we cannot guarantee that bulk reconstruction is possible for any specific single state $|\Psi\rangle$. Instead, we can only show that, with high probability, bulk reconstruction will be possible for the state $\hat{U}_{\mathbf{b}} \hat{U}'_{\bar{\mathbf{b}}} |\Psi\rangle$, where $\hat{U}_{\mathbf{b}}, \hat{U}'_{\bar{\mathbf{b}}}$ are chosen at random according to the Haar measure. This is because the area $A_B(\mathbf{b})$ is almost unaffected by the action of $V$ on a single state $|\Psi\rangle$. As a result, once $\varepsilon_2$ is nonzero, we can make bulk reconstruction impossible for the specific state $|\Psi\rangle$, without affecting Condition 2.

# 6 Discussion

## 6.1 State-specific definition of the entanglement wedge

In this paper, we have offered a new definition of the entanglement wedge as the region reconstructible in a particular *state-specific* way. The entanglement wedge of boundary region $B$ is precisely the bulk region whose state can be changed to any other state with the same spatial entanglement structure by acting unitarily on $B$ – if the state on the complementary bulk region can changed in the same way by acting on the complementary boundary region $\bar{B}$ (or $\bar{B}\bar{R}$ for the purification of a mixed state).

In Theorem 4.2, we showed this definition was equivalent to the more traditional definition – namely the bulk region $\mathbf{b}$ satisfying $S(B)_{V|\Psi\rangle} = A_B(\mathbf{b}) + S(\mathbf{b})_{|\Psi\rangle}$, where $V$ is the bulk-to-boundary map. That theorem also shows that, as a consequence, the entanglement wedge minimizes the generalized entropy and is equal to both the so-called min- and max-EWs that were defined in [11].

Theorem 4.2 is robust to small corrections, as we showed in Theorem 5.5. This is important for two reasons. Firstly, in holography bulk reconstruction is always only approximate, with inevitable errors of at least $e^{-\mathcal{O}(1/G)}$ [4, 17, 18]. Secondly, the known nontrivial examples of state-specific codes, such as random tensor networks, are generally approximate in nature. Indeed, for zero-bit codes, which are the special case of state-specific product unitary codes where there is only a single bulk subsystem, exact state-specific error correction always implies exact state-independent error correction. The difference between the two only appears when errors are introduced.

Finally, in Theorem 5.5, we also allowed the bulk-to-boundary map $V$ to be non-isometric. While fairly heretical from a traditional quantum information perspective, this generalization appears to be crucial to understand the encoding of the black hole interior into the boundary Hilbert space; we discuss this more in Section 6.5.

## 6.2 Geometry from entanglement

We have introduced a definition of "area" for surfaces bounding arbitrary bulk regions of arbitrary quantum codes. Moreover, we showed that it is the functionally unique definition of area that appears in the QMS prescription whenever such a prescription exists. We see this as

a significant step towards a general understanding of how the geometry of the bulk emerges from the entanglement structure of the dual CFT.

There is still some ambiguity here: one can redefine the area of surfaces that are never quantum minimal without affecting the QMS prescription, so long as we never make them smaller that the definition of area in Definition 2.1. Moreover, there is no reason to expect that Definition 2.1 will give areas that are consistent with a smooth metric at scales that are large compared to the cut-off scale.

Indeed, it is easy to check explicitly that our definition of area does not agree with geometric area for all surfaces in quantum gravity. It seems possible that there exist preferred properties that naturally pick out a "better" definition of area than Definition 2.1, thereby resolving this ambiguity. We leave that question to future work.

Regardless, it is clear that the emergent geometry depends not only on the entanglement structure of an individual boundary state, but also on the *quantum code V* relating the bulk to the boundary. From this perspective, geometry is still related to entanglement. But it is about more than that – it is about the specific relationship between the entanglement in the boundary and the encoding of the bulk into the boundary. This basic idea is not new; it's the same perspective given by Harlow in [1]. In this work, we have generalized that lesson to a much larger set of surfaces and areas.

This lesson is important, so we emphasize it: if we are to understand how the bulk geometry emerges from the dual CFT state, we need to understand the code that embeds bulk operators into the boundary.

While this might seem to make the emergence of geometry even more daunting – we have to understand not just boundary entanglement but also the bulk-to-boundary code – it actually presents a somewhat surprising simplification. Naively, one might expect that understanding the emergence of the bulk geometry requires understanding Planck scale physics. Instead, many features of the geometry depend only on the encoding of low-energy, effective-field-theory operators. This fact is one important reason to pursue an understanding of holographic codes in the quest to understand the emergence of spacetime. More speculatively, one might wonder how this relationship between bulk fields and geometry could connect to Einstein's equations, which also constrain the geometry based on data about the bulk fields; for previous work along somewhat similar lines, see [43–48].

## 6.3 Algebras with centers

Curiously, as discussed in Appendix B.4, it seems hard to make a theorem like 4.2 if the local bulk algebras contain nontrivial centers. No such theorem can be simultaneously consistent with both a) the definition of area from [1] for algebras with state-independent complementary reconstruction and b) Definition 2.1 for the area of algebras with trivial center.

Specifically, in Appendix B.4 we construct an example of a code with state-independent complementary reconstruction of algebras with nontrivial center, as in [1]. Then we show that the generalized entropy, defined using Definition 2.1, of the trivial algebra generated by only the identity is smaller than the generalized entropy of the entanglement wedge, defined as in [1]. That is, the entanglement wedge is not quantum minimal in this valid algebraic code.

This situation is fairly unsatisfactory. Local algebras with nontrivial centers play an important role in bulk gauge theories, which often appear in AdS/CFT. Perhaps more importantly, in code spaces where the bulk geometry is not fixed – and hence the area of a surface is a quantum operator rather than simply a number – the area operator itself should lie in the center of the reconstructible algebra.

There are a couple of potential resolutions to this issue. The first is related to the argument from [49] that bulk gauge fields in quantum gravity are always emergent at low energies, and

hence that the microscopic algebra does not contain a nontrivial center. This suggests that maybe we should really always be working with an "extended code space" that does factorize, and hence for which Theorem 4.2 applies, even when bulk gauge fields exist.

The downside of this proposal is that all states must have the same area for any bulk surface; any difference in geometry between different states has to be reinterpreted as a difference in bulk entanglement entropy. There is nothing wrong with such a reinterpretation – the equivalence of entanglement and area is the basic idea of ER=EPR [50] – but it doesn't achieve the goal of describing an emergent bulk geometry with nontrivial area operators.

An alternative possibility is that Definition 2.1 should be corrected for certain surfaces that are never quantum minimal, so that the counterexample found in Appendix B.4 no longer exists. In other words, the existence of algebras with centers in the quantum code changes Definition 2.1, even when applied to algebras that themselves have trivial center.

One alluring option is to combine these two possibilities, using extended Hilbert space considerations to learn how to redefine the area of non-minimal surfaces. For instance, in the example in Appendix B.4, the problem is that the trivial region has an area much smaller than the boundary entropy, so we could not simultaneously satisfy a holographic entropy prescription and minimality. Now note, the reason $A_B(\varnothing)$ is so small is that Definition 2.1 depends on the entropy of bulk states; increase the possible entropy of bulk states, and areas will in general increase as well. If we calculated $A_B(\varnothing)$ using a larger, factorizing "extended Hilbert space" of bulk states, we would find a much larger area and thereby avoid the issue!

## 6.4 Extremality

A long term goal of this program is to see the emergence of not just spatial geometry, but also to understand dynamics in holography. A first step would be to understand the quantum *extremal* surface formula. Can our theorems be generalized to go beyond quantum minimal surfaces and incorporate extremality in the time direction?

An immediate observation is that, as in the case of algebras with centers, the definition of area given in Definition 2.1 would need alteration. To see this, consider a Cauchy slice of a time-dependent spacetime in AdS/CFT that contains the minimal quantum extremal surface for some boundary region $B$, but which is not maximin (i.e. the Cauchy slice also contains a more minimal surface). Because the minimal QES divides the Cauchy slice into a part in the entanglement wedge of $B$ and a part in the entanglement wedge of $\overline{B}$, the fields on this Cauchy slice satisfy Condition 1 of Theorem 4.2, and hence obey a QMS prescription using the area from Definition 2.1. However, by assumption, the minimal QES is *not* quantum minimal on this Cauchy slice when we use the actual geometric area of surfaces.

To understand how this is consistent, we need to recall that Theorem 2.8, showing that the area from Definition 2.1 lower bounds any alternative definition of area $A'_B$, assumed explicitly that $A'_B$ itself obeyed a quantum minimal surface prescription. In other words, Theorem 2.8 only applies to surfaces that lie in a maximin slice (including e.g. any static slice). For surfaces in other slices, there is no guaranteed relationship between geometric area and Definition 2.1.

To obtain a full quantum maximin or QES prescription, one would presumably need to start with a set of bulk algebras associated to every spacetime region, with appropriate nesting properties, rather than simply a spatial tensor product structure. One issue here is that it is hard or impossible to construct finite-dimensional algebraic structures with exact relativistic lightcones.

## 6.5 State-specific reconstruction and the Page curve

One of the most exciting features of state-specific reconstruction is that, unlike state-independent QEC, it is compatible with codes where the linear map $V$ is not an isometry.

As such it can provide a "Hilbert space" justification of the use of the QES prescription to derive the Page curve of an evaporating black hole [9, 10] using the semiclassical Hawking state. In particular, it counters the objection of [?] that the "wrong" state is somehow being used to calculate the real state's entropy. While the physical state of the radiation is indeed not the Hawking state, it is a *state-specific encoding* $V |\Psi\rangle$ of the Hawking state $|\Psi\rangle$, via a non-isometric map $V$. Hence we can compute entropies from the Hawking state using the QES prescription. A detailed discussion of these issues will appear in upcoming work [52].

## 6.6 From Condition 3 to Condition 1?

An unfortunate feature of Theorem 4.2 is that it provides generally no *purely bulk* method to know that Conditions (1) and (2) are satisfied for some region **b**, even if you have been told the bulk geometry in advance. One can find the minimal generalized entropy bulk region, which will always be the region that satisfies Conditions 1 and 2 if any does, but you cannot know whether the code $V$ actually satisfies those conditions for the state $|\Psi\rangle$.

In contrast, in holography and in special classes of codes such as random tensor networks, Condition 3 appears to be both a necessary and sufficient condition for Conditions 1 and 2 to hold for the state $|\Psi\rangle$ [11]. Just by looking at bulk conditional min-entropies and areas, one can determine whether there is a well-defined entanglement wedge.

We shouldn't be surprised that the same is not true for general quantum codes; in general Condition 3 simply doesn't know enough about $V$ to be able to prove Conditions 1 and 2. However one might hope to find a simple additional constraint on the code $V$ that makes Condition 3 equivalent to Conditions 1 and 2. We leave the task of finding such a constraint to future work.

# Acknowledgements

It's a pleasure to thank Elba Alonso-Monsalve, Raphael Bousso, Netta Engelhardt, Patrick Hayden, Daniel Harlow, Tom Faulkner, Åsmund Folkstad, Adam Levine, Pratik Rath, and Arvin Shahbazi-Moghaddam for discussions. CA is supported by the Simons foundation as a member of the It from Qubit collaboration and the Air Force Office of Scientific Research under the award number FA9550-19-1-0360. GP is supported by the UC Berkeley physics department, the Simons Foundation through the "It from Qubit" program, the Department of Energy via the GeoFlow consortium (QuantISED Award DE-SC0019380) and also acknowledges support from a J. Robert Oppenheimer Visiting Professorship at the Institute for Advanced Study.

# A  Proofs of auxiliary theorems

Here we collect the proofs of the minor theorems that were not included in the main text. For the reader's convenience, we begin with proofs of some well-known theorems that will be used in both the proofs later in this appendix and in the proof of Theorem 5.5.

## A.1 Preliminaries

**Lemma A.1** (Hölder's inequality for the trace and operator norms)**.** *For any operators A, B we have*

$$\text{tr}[AB] \leq \|A\|_1 \|B\|_\infty. \tag{144}$$

*Proof.* Let $A = \sum_i \lambda_i |\psi_i\rangle \langle \phi_i|$ be a singular value decomposition. Then

$$\text{tr}[AB] = \sum_i \lambda_i \langle \phi_i|B|\psi_i\rangle \leq \sum_i \lambda_i \|B\|_\infty = \|A\|_1 \|B\|_\infty. \tag{145}$$

$\square$

**Lemma A.2** (Triangle inequality). *For any operators $A, B$ we have*

$$\|A + B\|_1 \leq \|A\|_1 + \|B\|_1. \tag{146}$$

*Proof.* Let $A + B = UDV^\dagger$ be a singular value decomposition. Then

$$\|A + B\|_1 = \text{tr}[(A+B)VU^\dagger] \leq \|A\|_1 \|VU^\dagger\|_\infty + \|B\|_1 \|VU^\dagger\|_\infty \leq \|A\|_1 + \|B\|_1, \tag{147}$$

where the first inequality uses Hölder's inequality. $\square$

**Lemma A.3** (Monotonicity). *Let $X : \mathcal{H}_A \otimes \mathcal{H}_B \to \mathcal{H}_A \otimes \mathcal{H}_B$ be an arbitrary operator. Then*

$$\|\text{tr}_B X\|_1 \leq \|X\|_1. \tag{148}$$

*Proof.* By Hölder's inequality,

$$\|\text{tr}_B X\|_1 = \max_{U_A} \text{tr}[XU_A] \leq \max_{U_{AB}} \text{tr}[XU_{AB}] = \|X\|_1. \tag{149}$$

Here we used the singular value decomposition to see that a unitary saturating Hölder's inequality exists. $\square$

**Lemma A.4** (Lemma 3.21 of [53]). *Let $X, Y : \mathcal{H}_A \to \mathcal{H}_B$ be arbitrary operators. It holds that*

$$F(XX^\dagger, YY^\dagger) = \|X^\dagger Y\|_1. \tag{150}$$

*Proof.* Let $X = U_1 D_1 V_1^\dagger$ and $Y = U_2 D_2 V_2^\dagger$ be singular value decompositions. We have

$$F(XX^\dagger, YY^\dagger) = \left\|\sqrt{XX^\dagger}\sqrt{YY^\dagger}\right\|_1 = \left\|U_1 D_1 U_1^\dagger U_2 D_2 U_2^\dagger\right\|_1 = \left\|V_1 D_1 U_1^\dagger U_2 D_2 V_2^\dagger\right\|_1 = \|X^\dagger Y\|_1. \tag{151}$$

$\square$

**Lemma A.5** (Uhlmann's theorem). *Consider Hilbert spaces $\mathcal{H}_A$ and $\mathcal{H}_B$. Let $\rho, \sigma$ be positive semidefinite operators on $\mathcal{H}_A$ with rank at most $\dim \mathcal{H}_B$, and let $|\psi\rangle \in \mathcal{H}_A \otimes \mathcal{H}_B$ satisfy $\text{tr}_B[|\psi\rangle \langle \psi|] = \rho$. It holds that*

$$F(\rho, \sigma) = \max\{|\langle \psi|\phi\rangle| : |\phi\rangle \in \mathcal{H}_A \otimes \mathcal{H}_B, \text{tr}_B[|\phi\rangle \langle \phi|] = \sigma\}. \tag{152}$$

*Proof.* See e.g. Theorem 3.22 of [53]. Let $X : \mathcal{H}_B \to \mathcal{H}_A$ be a linear operator satisfying $|\psi\rangle = \text{vec}(X)$, where

$$\text{vec}\left(\sum_{ij} c_{ij} |i\rangle_A \langle j|_B\right) := \sum_{ij} c_{ij} |i\rangle_A |j\rangle_B.$$

Similarly, let $Y : \mathcal{H}_B \to \mathcal{H}_A$ be a linear operator satisfying $|\phi\rangle = \text{vec}(Y)$ for some $|\phi\rangle$ satisfying $\text{tr}_B[|\phi\rangle \langle \phi|] = \sigma$. It follows that by Lemma A.4 that

$$F(\rho, \sigma) = F(XX^\dagger, YY^\dagger) = \|X^\dagger Y\|_1 = \max_{U_B} \text{tr}[X^\dagger Y U] = \max_{U_B} \langle \psi|U_B|\phi\rangle. \tag{153}$$

The result then follows from the equivalence of purification up to a unitary on the purifying system. $\square$

**Lemma A.6** (Lemma 3.34 of [53]). *Let $P_0$ and $P_1$ be positive semidefinite operators on Hilbert space $\mathcal{H}$. It holds that*

$$\|P_0 - P_1\|_1 \geq \|\sqrt{P_0} - \sqrt{P_1}\|_2^2. \tag{154}$$

*Proof.* Let $\Pi_0, \Pi_1 = \mathbb{1} - \Pi_0$ be projectors onto the positive and negative eigenspaces of $\sqrt{P_0} - \sqrt{P_1}$ and let $Q_j = (-1)^j \Pi_j(\sqrt{P_0} - \sqrt{P_1})\Pi_j$. By Hölder's inequality,

$$\operatorname{tr}\left[(\Pi_0 - \Pi_1)(P_0 - P_1)\right] \leq \|P_0 - P_1\|_1 \|\Pi_0 - \Pi_1\|_\infty = \|P_0 - P_1\|_1. \tag{155}$$

Now

$$\begin{aligned}
\operatorname{tr}\left[(\Pi_0 - \Pi_1)(P_0 - P_1)\right] &= \frac{1}{2}\operatorname{tr}\left[(\Pi_0 - \Pi_1)(\sqrt{P_0} - \sqrt{P_1})(\sqrt{P_0} + \sqrt{P_1})\right] \\
&\quad + \frac{1}{2}\operatorname{tr}\left[(\Pi_0 - \Pi_1)(\sqrt{P_0} + \sqrt{P_1})(\sqrt{P_0} - \sqrt{P_1})\right] \\
&= \operatorname{tr}\left[(Q_0 + Q_1)(\sqrt{P_0} + \sqrt{P_1})\right].
\end{aligned} \tag{156}$$

Finally, because $Q_0, Q_1, \sqrt{P_0}, \sqrt{P_1}$ are all positive semidefinite, we have

$$\begin{aligned}
\operatorname{tr}\left[(Q_0 + Q_1)(\sqrt{P_0} + \sqrt{P_1})\right] &\geq \operatorname{tr}\left[(Q_0 - Q_1)(\sqrt{P_0} - \sqrt{P_1})\right] \\
&= \|\sqrt{P_0} - \sqrt{P_1}\|_2^2.
\end{aligned} \tag{157}$$

$\square$

**Lemma A.7** (Fuchs-van de Graaf inequalities). *Let $\rho, \sigma$ be density matrices on Hilbert space $\mathcal{H}_A$. It holds that*

$$1 - \frac{1}{2}\|\rho - \sigma\|_1 \leq F(\rho, \sigma) \leq \sqrt{1 - \frac{1}{4}\|\rho - \sigma\|_1^2}, \tag{158}$$

*or equivalently*

$$2 - 2F(\rho, \sigma) \leq \|\rho - \sigma\|_1 \leq 2\sqrt{1 - F(\rho, \sigma)^2}. \tag{159}$$

*Proof.* See e.g. Theorem 3.33 of [53]. We will prove the two inequalities in (159), starting with the first. Using Lemma A.6, one obtains

$$\|\rho - \sigma\|_1 \geq \|\sqrt{\rho} - \sqrt{\sigma}\|_2^2 = \operatorname{tr}\left[(\sqrt{\rho} - \sqrt{\sigma})^2\right] = 2 - 2\operatorname{tr}\left[\sqrt{\rho}\sqrt{\sigma}\right] = 2 - 2F(\rho, \sigma). \tag{160}$$

Now for the second inequality in (159). Let $\mathcal{H}_B$ be a Hilbert space with $\dim \mathcal{H}_B = \dim \mathcal{H}_A$. It follows by Uhlmann's theorem that there exist states $|\psi\rangle, |\phi\rangle \in \mathcal{H}_A \otimes \mathcal{H}_B$ satisfying

$$\operatorname{tr}_B[|\psi\rangle\langle\psi|] = \rho, \quad \operatorname{tr}_B[|\phi\rangle\langle\phi|] = \sigma, \quad |\langle\psi|\phi\rangle| = F(\rho, \sigma). \tag{161}$$

Note it holds that

$$\||\psi\rangle\langle\psi| - |\phi\rangle\langle\phi|\|_1 = 2\sqrt{1 - |\langle\psi|\phi\rangle|^2} = 2\sqrt{1 - F(\rho, \sigma)^2}, \tag{162}$$

which can be proven by explicitly diagonalizing the rank-2 matrix $|\psi\rangle\langle\psi| - |\phi\rangle\langle\phi|$. By monotonicity, we have

$$\|\rho - \sigma\|_1 \leq \||\psi\rangle\langle\psi| - |\phi\rangle\langle\phi|\|_1, \tag{163}$$

completing the proof. $\square$

**Lemma A.8.** *Let $\eta(x) = -x \log x$. Then, for $0 \leq r, s \leq 1$ with $|r - s| \leq 1/2$, we have*

$$|\eta(r) - \eta(s)| \leq \eta(|r - s|). \tag{164}$$

*Proof.* Without loss of generality, we can assume $r > s$. We first show that $\eta(r) - \eta(s) \leq \eta(r-s)$. We have

$$\eta(r) - \eta(s) = \int_s^r dx\ \eta'(x) \leq \int_0^{r-s} dx\ \eta'(x) = \eta(r-s), \tag{165}$$

where the inequality follows from the monotonicity of $\eta'(x) = -\log x - 1$.

To show $\eta(s) - \eta(r) \leq \eta(r-s)$ for $0 \leq s \leq r \leq 1$ and $r - s \leq 1/2$ is slightly more tedious. We simply search systematically for the minimal value of $f(r,s) = \eta(r-s) + \eta(r) - \eta(s)$ within the allowed region and show that it is zero. Since $\nabla f = 0$ requires $r = s$, there are no minima within the interior of the region. Considering each boundary piece in turn: for $r = s$ we have $f = 0$. For $s = 0$ we have $f = 2\eta(r) \geq 0$. For $r = 1$, we have $f(1, 1/2) = f(1,1) = 0$ with a single maximum in between. Finally, for $r - s = 1/2$, $f(r,s)$ decreases monotonically with increasing $r, s$ reaching its minimal value at $f(1, 1/2) = 0$. This completes the proof. $\qquad\square$

**Lemma A.9** (Fannes' inequality). *Let $\rho, \sigma$ be (subnormalized) density matrices on the Hilbert space $\mathcal{H}_A$ such that $\|\rho - \sigma\|_1 \leq \varepsilon \leq 1/e$ and let $S(\rho) := -\mathrm{tr}[\rho \log \rho]$. Then*

$$|S(\rho) - S(\sigma)| \leq \varepsilon \log \frac{d_A}{\varepsilon}. \tag{166}$$

*Proof.* See e.g. Theorem 11.6 of [54]. If we write $\rho - \sigma = P - Q$, with $P$ and $Q$ positive semidefinite and orthogonal, then

$$\|\rho - \sigma\|_1 = \mathrm{tr}[P + Q] = \mathrm{tr}[2\tau - \rho - \sigma], \tag{167}$$

where $\tau = \rho + Q = \sigma + P$. Writing $r_i$, $s_i$, and $t_i$ respectively for the eigenvalues in descending order of $\rho$, $\sigma$, and $\tau$, we have $t_i \geq \max\{r_i, s_i\} = 1/2[r_i + s_i + |r_i - s_i|]$ and hence

$$\|\rho - \sigma\|_1 = \sum_i (2t_i - r_i - s_i) \geq \sum_i |r_i - s_i|. \tag{168}$$

Since for all $i$ we have $|r_i - s_i| \leq 1/e$, by Lemma A.8

$$|S(\rho) - S(\sigma)| = \left| \sum_i (\eta(r_i) - \eta(s_i)) \right| \leq \sum_i \eta(|r_i - s_i|). \tag{169}$$

If $\Delta = \sum_i |r_i - s_i|$, then

$$\sum_i \eta(|r_i - s_i|) = \eta(\Delta) \sum_i \delta\eta\left( \frac{|r_i - s_i|}{\Delta} \right) \leq \Delta \log \frac{d_A}{\Delta}, \tag{170}$$

where the inequality follows from the bound $S(\rho) \leq \log d_A$. The result then follows from the monotonicity of $\eta(x)$ in the range $0 \leq x \leq 1/e$. $\qquad\square$

## A.2 Areas in general quantum codes

**Lemma A.10.** *Let $|\mathrm{MAX}\rangle \in \otimes_i[\mathcal{H}_{b_i} \otimes \mathcal{H}_{r_i}]$, with $\mathcal{H}_{r_i} \cong \mathcal{H}_{b_i}^*$, be the canonical maximally entangled state and let $\mathbf{b}, \mathbf{b}' \subseteq \{b_1 \ldots b_n\}$ be arbitrary subsets of $\{b_1 \ldots b_n\}$, with $\mathbf{r} := \{r_i : b_i \in \mathbf{b}\}$ and $\mathbf{r}' := \{r_i : b_i \in \mathbf{b}'\}$ the corresponding subsets of $\{r_1 \ldots r_n\}$. Then*

$$S(\mathbf{b}'\mathbf{r})_{|\mathrm{MAX}\rangle} \geq S(\mathbf{b}'R)_{|\Psi\rangle} - S(\mathbf{b}R)_{|\Psi\rangle}, \tag{171}$$

*for any state $|\Psi\rangle \in \otimes_i \mathcal{H}_{b_i} \otimes \mathcal{H}_R \otimes \mathcal{H}_{\bar{R}}$.*

*Proof.* The state $|\text{MAX}\rangle$ is maximally entangled on $\mathcal{H}_{\mathbf{b}\cap\mathbf{b}'}\otimes\mathcal{H}_{\mathbf{r}\cap\mathbf{r}'}$, while the reduced state of $|\text{MAX}\rangle$ on $\mathcal{H}_{\mathbf{b}'}\otimes\mathcal{H}_{\mathbf{r}}$ is maximally mixed on $\mathcal{H}_{\mathbf{b}'\backslash\mathbf{b}}\otimes\mathcal{H}_{\mathbf{r}\backslash\mathbf{r}'}$. We therefore find

$$S(\mathbf{b}'\mathbf{r})_{|\text{MAX}\rangle} = S(\mathbf{b}'\backslash\mathbf{b})_{|\text{MAX}\rangle} + S(\mathbf{r}\backslash\mathbf{r}')_{|\text{MAX}\rangle} \tag{172}$$

$$= S(\mathbf{b}'\backslash\mathbf{b})_{|\text{MAX}\rangle} + S(\mathbf{b}\backslash\mathbf{b}')_{|\text{MAX}\rangle} \tag{173}$$

$$\geq S(\mathbf{b}'\backslash\mathbf{b})_{|\Psi\rangle} + S(\mathbf{b}\backslash\mathbf{b}')_{|\Psi\rangle}, \tag{174}$$

for any state $|\Psi\rangle \in \otimes_i \mathcal{H}_{b_i}\otimes\mathcal{H}_R\otimes\mathcal{H}_{\overline{R}}$.

It remains to show that

$$S(\mathbf{b}'\backslash\mathbf{b})_{|\Psi\rangle} + S(\mathbf{b}\backslash\mathbf{b}')_{|\Psi\rangle} \geq S(\mathbf{b}'R)_{|\Psi\rangle} - S(\mathbf{b}R)_{|\Psi\rangle}. \tag{175}$$

By subadditivity, we have

$$S(\mathbf{b}'R)_{|\Psi\rangle} \leq S(\mathbf{b}'\backslash\mathbf{b})_{|\Psi\rangle} + S(\mathbf{b}\cap\mathbf{b}'R)_{|\Psi\rangle}, \tag{176}$$

while, by the Araki-Lieb inequality, we have

$$S(\mathbf{b}R)_{|\Psi\rangle} \geq S(\mathbf{b}\cap\mathbf{b}'R)_{|\Psi\rangle} - S(\mathbf{b}\backslash\mathbf{b}')_{|\Psi\rangle}. \tag{177}$$

Combining these completes the proof. $\qquad\square$

**Proof of Theorem 2.8**

*Proof.* We first prove that the area function $A_B$ always leads to a QMS prescription for the state $|\text{MAX}\rangle \in \otimes_i[\mathcal{H}_{b_i}\otimes\mathcal{H}_{r_i}]$ and region $B\mathbf{r}$ with entanglement wedge $\mathbf{b}$. This is because

$$S(B\mathbf{r})_{V|\text{MAX}\rangle} = A_B(\mathbf{b}) = A_B(\mathbf{b}) + S(\mathbf{b}\mathbf{r})_{|\text{MAX}\rangle} \tag{178}$$

by definition. Also, any product unitary $U = U_{b_1}U_{b_2}\dots$ acting on $|\text{MAX}\rangle$ can be mirrored onto a product unitary $U_{r_1}U_{r_2}\dots$ acting on $\otimes_i\mathcal{H}_{r_i}$, which manifestly leaves $S(B\mathbf{r})$ unchanged. Hence we always have

$$S(B\mathbf{r})_{VU|\text{MAX}\rangle} = A_B(\mathbf{b}) = A_B(\mathbf{b}) + S(\mathbf{b}\mathbf{r})_{U|\text{MAX}\rangle}. \tag{179}$$

Finally, for any other set of subsystems $\mathbf{b}'$, we have

$$A_B(\mathbf{b}') + S(\mathbf{b}'\mathbf{r})_{|\text{MAX}\rangle} = S(B\mathbf{r}')_{|\text{CJ}\rangle} + S(\mathbf{r}'\backslash\mathbf{r})_{|\text{MAX}\rangle} + S(\mathbf{r}\backslash\mathbf{r}')_{|\text{MAX}\rangle} \geq S(B\mathbf{r}), \tag{180}$$

where the inequality follows from the Araki-Lieb inequality together with subadditivity as in the proof of Lemma A.10. This completes the proof that $A_B$ leads to a QMS prescription with entanglement wedge $\mathbf{b}$ for the state $|\text{MAX}\rangle$ and the region $B\mathbf{r}$.

Hence, according to Definition 2.7, the area function $A'_B$ must also lead to a QMS prescription for the state $|\text{MAX}\rangle$ and the region $B\mathbf{r}$. Of course, we do not yet know what the entanglement wedge is according to that prescription, but it must be some $\mathbf{b}' \subseteq \{b_1\dots b_n\}$. But then

$$A_B(\mathbf{b}) = S(B\mathbf{r})_{|\text{CJ}\rangle} = A'_B(\mathbf{b}') + S(\mathbf{b}'\mathbf{r})_{|\text{MAX}\rangle} \leq A'_B(\mathbf{b}) + S(\mathbf{b}\mathbf{r})_{|\text{MAX}\rangle} = A'_B(\mathbf{b}), \tag{181}$$

where in the second equality we used the $A'_B$ QMS prescription to evaluate $S(B\mathbf{r})_{|\text{CJ}\rangle}$ and in the inequality we used the requirement of minimality. $\qquad\square$

**Proof of Theorem 2.9**

*Proof.* By way of contradiction, we suppose that there does exist a state $|\Psi\rangle$ and subset $\mathbf{b}$ such that $\mathbf{b} = \mathrm{EW}'_{BR}(|\Psi\rangle)$, but $A'_B(\mathbf{b}) \neq A_B(\mathbf{b})$. Recall from the proof of Theorem 2.8 that the area function $A_B$ leads to a QMS prescription for the state $|\mathrm{MAX}\rangle$ and the region $B\mathbf{r}$, and hence by assumption the $A'_B$ area function must also do so. If we use the $A'_B$ QMS prescription to evaluate $A_B(\mathbf{b})$, we find

$$A_B(\mathbf{b}) = S(B\mathbf{r})_{|\mathrm{CJ}\rangle} = A'_B(\mathbf{b}') + S(\mathbf{b}'\mathbf{r})_{|\mathrm{MAX}\rangle}, \tag{182}$$

for some $\mathbf{b}' \subseteq \{b_1 \ldots b_n\}$. We can then use Lemma A.10 and Theorem 2.8 to obtain

$$A'_B(\mathbf{b}) + S(\mathbf{b}R)_{|\Psi\rangle} > A_B(\mathbf{b}) + S(\mathbf{b}R)_{|\Psi\rangle} \tag{183}$$

$$> A'_B(\mathbf{b}') + S(\mathbf{b}'\mathbf{r})_{|\mathrm{MAX}\rangle} + S(\mathbf{b}R)_{|\Psi\rangle} \tag{184}$$

$$> A'_B(\mathbf{b}') + S(\mathbf{b}'R)_{|\Psi\rangle}. \tag{185}$$

In (183), we used Theorem 2.8 together with the assumption that $A'_B(\mathbf{b}) \neq A_B(\mathbf{b})$. In (184), we used the $A'_B$ QMS prescription for $A_B(\mathbf{b})$ from (182). Finally, in (185), we used Lemma A.10. It can immediately be seen that (185) contradicts our assumption that $\mathbf{b} = \mathrm{EW}'_{BR}(|\Psi\rangle)$, since $\mathbf{b}$ does not have minimal generalized entropy for the state $|\Psi\rangle$. □

## A.3 State-specific reconstruction

**Proof of Theorem 3.1**

*Proof.* We prove four implications.

**1 → 2:** From (25) we have that for any $|\psi\rangle \in \mathcal{H}_{\mathrm{code}}$,

$$\langle\psi|O_{\mathrm{code}}|\psi\rangle = \mathrm{tr}[|\psi\rangle\langle\psi|O_{\mathrm{code}}] \tag{186}$$

$$= \mathrm{tr}\left[\mathrm{tr}_{\overline{B}E}[W_B V|\psi\rangle\langle\psi|V^\dagger W_B^\dagger]O_{\mathrm{code}}\right] \tag{187}$$

$$= \langle\psi|V^\dagger W_B^\dagger O_{\mathrm{code}}W_B^\dagger V|\psi\rangle \tag{188}$$

$$= \langle\psi|V^\dagger O_B V|\psi\rangle. \tag{189}$$

**2 → 3:** Let $\widetilde{O}_{\mathrm{code}} := O_{\mathrm{code}}^2$. Condition 2, applied to both $O_{\mathrm{code}}$ and $\widetilde{O}_{\mathrm{code}}$, implies

$$V^\dagger \widetilde{O}_B V = \widetilde{O}_{\mathrm{code}} = O_{\mathrm{code}}^2 = V^\dagger O_B V V^\dagger O_B V. \tag{190}$$

But because $VV^\dagger$ and $W_B W_B^\dagger$ are projectors,

$$V^\dagger O_B V V^\dagger O_B V \leq V^\dagger O_B O_B V \tag{191}$$

$$\leq V^\dagger W_B^\dagger O_{\mathrm{code}}W_B W_B^\dagger O_{\mathrm{code}}W_B V \tag{192}$$

$$\leq V^\dagger W_B^\dagger O_{\mathrm{code}}O_{\mathrm{code}}W_B V \tag{193}$$

$$\leq V^\dagger \widetilde{O}_B V, \tag{194}$$

with the first inequality (191) saturated if and only if

$$O_B V = VV^\dagger O_B V = V O_{\mathrm{code}}, \tag{195}$$

which is equivalent to (27).

**3 → 4:** Let $U_{\text{code}} = \exp(iO_{\text{code}})$. Statement 3 tells us that for all states $|\psi\rangle \in \mathcal{H}_{\text{code}}$

$$V U_{\text{code}} |\psi\rangle = V \exp(iO_{\text{code}}) |\psi\rangle = \exp(iO_B) V |\psi\rangle = U_B V |\psi\rangle \,, \tag{196}$$

where $U_B = \exp(iO_B)$.

**4 → 1:** This last step is easiest to prove using the technology that we develop in Section 4.2. In Lemma 4.4, it is shown that there exists an isometry $W : \mathcal{H}_A \to \mathcal{H}_A \otimes \mathcal{H}_{\mathbf{U}_A}$, where $\mathcal{H}_A$ is a finite-dimensional Hilbert space and $\mathcal{H}_{\mathbf{U}_A}$ is the space of square-integrable functions on the unitary group $\mathbf{U}_A$ with "position basis" $\{|U_A\rangle\}$, such that

$$W |\psi\rangle_A = \int dU_A |U_A\rangle_{\mathbf{U}_A} \otimes U_A |\psi\rangle_A = |\psi\rangle_{\mu_0} |\text{MAX}\rangle_{A\mu_0^*} \,. \tag{197}$$

Here $\mathcal{H}_{\mu_0} \otimes \mathcal{H}_{\mu_0}^* \subseteq \mathcal{H}_{\mathbf{U}_A}$ is a finite dimensional subspace of $\mathcal{H}_{\mathbf{U}_A}$ with $\mathcal{H}_{\mu_0} \cong \mathcal{H}_A$, and $dU_A$ is the Haar measure on $\mathbf{U}_A$. We can therefore define an isometry $W_B : \mathcal{H}_B \to \mathcal{H}_B \otimes \mathcal{H}_{\mathbf{U}_{\text{code}}}$ such that

$$W_B V |\psi\rangle = \int dU_{\text{code}} |U_{\text{code}}\rangle_{\mathbf{U}_{\text{code}}} \otimes U_B V |\psi\rangle \tag{198}$$

$$= \int dU_{\text{code}} |U_{\text{code}}\rangle V U_{\text{code}} |\psi\rangle \tag{199}$$

$$= |\psi\rangle V |\text{MAX}\rangle \,, \tag{200}$$

as desired for Condition 1.[26] $\qquad\square$

**Proof of Theorem 3.2**

*Proof.* This proof is almost identical to that of Theorem 3.1.

**1 → 2:** From (29) we have that for any $|\psi\rangle \in \mathcal{H}_{\text{code}}$,

$$\langle\psi|O_b|\psi\rangle = \text{tr}[\text{tr}_{\bar{b}}[|\psi\rangle\langle\psi|]O_b] \tag{201}$$

$$= \text{tr}\left[\text{tr}_{\overline{BE}}[W_B V |\psi\rangle\langle\psi| V^\dagger W_B^\dagger]O_b\right] \tag{202}$$

$$= \langle\psi|V^\dagger O_B V|\psi\rangle \,. \tag{203}$$

**2 → 3:** Let $\widetilde{O}_b := O_b^2$. Condition 2 implies

$$V^\dagger \widetilde{O}_B V = \widetilde{O}_b = O_b^2 = V^\dagger O_B V V^\dagger O_B V \,. \tag{204}$$

But because $W_B V V^\dagger W_B^\dagger \leq V V^\dagger \leq \mathbb{1}$, we have

$$V^\dagger O_B V V^\dagger O_B V \leq V^\dagger \widetilde{O}_B V \,, \tag{205}$$

with equality only possible if $O_B V = V V^\dagger O_B V = V O_b$.

**3 → 4:** Condition 4 follows immediately from Condition 3 if we write $U_b = \exp(iO_b)$ and $U_B = \exp(iO_B)$.

---

[26] One might worry here about whether $W_B$ is truly an isometry mapping $\mathcal{H}_B \to \mathcal{H}_{\text{code}} \otimes \mathcal{H}_E$ for some $\mathcal{H}_E$, as required by Condition 1. Since $\mathcal{H}_{\mathbf{U}_{\text{code}}}$ is infinite dimensional, we can always write $\mathcal{H}_{\mathbf{U}_{\text{code}}} \cong \mathcal{H}_{\text{code}} \otimes \mathcal{H}_{E'}$ such that $\mathcal{H}_{\text{code}}$ is identified with $\mathcal{H}_{\mu_0}$ on the subspace $\mathcal{H}_{\mu_0} \otimes \mathcal{H}_{\mu_0}^*$. Moreover since $\mathcal{H}_B$ and $\mathcal{H}_{\text{code}}$ are finite dimensional, the image of $W_B$ then lies in $\mathcal{H}_{\text{code}} \otimes \mathcal{H}_{E''}$ where $\mathcal{H}_{E''} \subseteq \mathcal{H}_{E'}$ is finite dimensional. We can therefore simply identify $\mathcal{H}_E \cong \mathcal{H}_B \otimes \mathcal{H}_{E''}$ to reach the exact form specified in Condition 1.

**4 → 1:**  As in the proof of Theorem 3.1, we define

$$W_B := \int dU_b \, |U_b\rangle \otimes U_B \, . \tag{206}$$

Then

$$W_B V \, |\psi\rangle_{b\bar{b}} = \int dU_b \, |U_b\rangle \, V U_b \, |\psi\rangle_{b\bar{b}} = V \, |\mathrm{MAX}\rangle_{b\mu_0^*} \, |\psi\rangle_{\mu_0 \bar{b}} \, , \tag{207}$$

where on the right hand side $V$ still acts on $\mathcal{H}_{\mathrm{code}} \cong \mathcal{H}_b \otimes \mathcal{H}_{\bar{b}}$. The reduced state $W_B V \, |\psi\rangle$ on $\mathcal{H}_{\mu_0}$ is therefore equal to the reduced state of $|\psi\rangle$ on $\mathcal{H}_b$, as required by Condition 1. $\qquad\square$

**Proof of Theorem 3.5**

*Proof.* We start with Condition 1 and prove the cycle of implications.

**1 → 2:**  Consider the state $|\Phi\rangle \in \mathcal{H}_{\mathrm{code}} \otimes \mathcal{H}_R$, where $\mathcal{H}_R$ is a two-dimensional reference system, given by

$$|\Phi\rangle = \frac{1}{\sqrt{2}}[|\psi\rangle \, |0\rangle + |\phi\rangle \, |1\rangle] \, , \tag{208}$$

where $\{|\psi\rangle, |\phi\rangle\}$ is an orthonormal basis for $\widetilde{\mathcal{H}}_{\mathrm{code}}$, and let $|\chi^\pm\rangle = 1/\sqrt{2}[|\psi\rangle \pm |\phi\rangle]$ and $|\chi^{\pm i}\rangle = 1/\sqrt{2}[|\psi\rangle \pm i\,|\phi\rangle]$. We have

$$\begin{aligned}
\Phi_{\bar{B}R} =& \frac{1}{4}(\chi_{\bar{B}}^+ - \chi_{\bar{B}}^-) \otimes X_R + \frac{1}{4}(\chi_{\bar{B}}^{-i} - \chi_{\bar{B}}^{+i}) \otimes Y_R \\
& + \frac{1}{2}\psi_{\bar{B}} \otimes |0\rangle\langle 0|_R + \frac{1}{2}\phi_{\bar{B}} \otimes |1\rangle\langle 1|_R \\
\approx& \frac{1}{2}\omega_{\bar{B}} \otimes \mathbb{1}_R \, .
\end{aligned} \tag{209}$$

Here $X_R$ and $Y_R$ are respectively the Pauli X and Y operators on $\mathcal{H}_R$. In the approximate equality we used (38), applied to all of $|\psi\rangle$, $|\phi\rangle$, $|\chi^\pm\rangle$, and $|\chi^{\pm i}\rangle$. More precisely, we can use the triangle inequality for Schatten 1-norms to see that

$$\begin{aligned}
\left\| \Phi_{\bar{B}R} - \frac{1}{2}\omega_{\bar{B}} \otimes \mathbb{1}_R \right\|_1 \le& \frac{1}{2}\left\| \chi_{\bar{B}}^+ - \omega_{\bar{B}} \right\|_1 + \frac{1}{2}\left\| \chi_{\bar{B}}^- - \omega_{\bar{B}} \right\|_1 + \frac{1}{2}\left\| \chi_{\bar{B}}^{+i} - \omega_{\bar{B}} \right\|_1 \\
& + \frac{1}{2}\left\| \chi_{\bar{B}}^{-i} - \omega_{\bar{B}} \right\|_1 + \frac{1}{2}\left\| \psi_{\bar{B}} - \omega_{\bar{B}} \right\|_1 + \frac{1}{2}\left\| \phi_{\bar{B}} - \omega_{\bar{B}} \right\|_1 \\
\le& \, 3\,\varepsilon_1 \, .
\end{aligned} \tag{210}$$

In (210) we implicitly used the trace norms $\|X_R\|_1 = \|Y_R\|_1 = 2$ and $\||0\rangle\langle 0|_R\|_1 = \||1\rangle\langle 1|_R\|_1 = 1$. By the first Fuchs-van de Graaf inequality, we therefore have

$$F(\Phi_{\bar{B}R}, \frac{1}{2}\omega_{\bar{B}} \otimes \mathbb{1}_R) \ge 1 - \frac{3}{2}\varepsilon_1 \, . \tag{211}$$

Let $|\omega\rangle \in \mathcal{H}_{\bar{B}} \otimes \mathcal{H}_E$ be a purification of $\omega_{\bar{B}}$ where the dimension $d_E \ge d_B$. By Uhlmann's theorem, there exists an isometry $\widetilde{W}_B : \mathcal{H}_B \to \mathcal{H}_{\mathrm{code}} \otimes \mathcal{H}_E$ such that

$$\left| \langle \Phi| \, \langle \omega| \, \widetilde{W}_B V \, |\Phi\rangle \right|^2 \ge 1 - \frac{3}{2}\varepsilon_1 \, . \tag{212}$$

But we can now use the second Fuchs-van de Graaf inequality to bound the Schatten 1-norm

$$\left\|\mathrm{tr}_{\overline{B}E}[\widetilde{W}_B V |\Phi\rangle\langle\Phi| V^\dagger \widetilde{W}_B^\dagger] - |\Phi\rangle\langle\Phi|\right\|_1 \leq 2\sqrt{3\varepsilon_1}. \tag{213}$$

For any state $\tilde{\rho}_{\mathrm{code}}$ there exists a positive operator $\tilde{\rho}_R^{1/2}$ with operator norm $\|\tilde{\rho}_R^{1/2}\| \leq \sqrt{2}$ such that $\mathrm{tr}_R[\tilde{\rho}_R^{1/2} |\Phi\rangle\langle\Phi| \tilde{\rho}_R^{1/2}] = \tilde{\rho}_{\mathrm{code}}$. Hence, by monotonicity under partial traces and the Hölder inequality, we have

$$\left\|\mathrm{tr}_{\overline{B}E}[\widetilde{W}_B V \tilde{\rho}_{\mathrm{code}} V^\dagger \widetilde{W}_B^\dagger] - \tilde{\rho}_{\mathrm{code}}\right\|_1 \leq 4\sqrt{3\varepsilon_1}. \tag{214}$$

Hence $\varepsilon_2 \leq 4\sqrt{3\varepsilon_1}$.

**2 → 3:**   Let $\widetilde{\mathcal{H}}_{\mathrm{code}} = \mathrm{span}\{|\psi\rangle, |\phi\rangle\}$. Now define the positive operator

$$P_B = \widetilde{W}_B^\dagger |\psi\rangle\langle\psi| \widetilde{W}_B.$$

Note that $P_B \leq \mathbb{1}$, but $P_B$ is not necessarily a projector. By (39), we have

$$1 - \langle\psi| V^\dagger P_B V |\psi\rangle = \mathrm{tr}\left[|\psi\rangle\langle\psi| \left(|\psi\rangle\langle\psi| - \mathrm{tr}_{\overline{B}E} \widetilde{W}_B V |\psi\rangle\langle\psi| V^\dagger \widetilde{W}_B^\dagger\right)\right] \leq \varepsilon_2, \tag{215}$$

and

$$\langle\phi| V^\dagger P_B V |\phi\rangle = \mathrm{tr}\left[|\psi\rangle\langle\psi| \left(\mathrm{tr}_{\overline{B}E} \widetilde{W}_B V |\phi\rangle\langle\phi| V^\dagger \widetilde{W}_B^\dagger - |\phi\rangle\langle\phi|\right)\right] \leq \varepsilon_2. \tag{216}$$

Hence

$$\mathrm{tr}[P_B(V |\psi\rangle\langle\psi| V^\dagger - V |\phi\rangle\langle\phi| V^\dagger)] \geq 1 - 2\varepsilon_2. \tag{217}$$

Suppose we now try to maximize the left hand side of (217) over all operators $0 \leq P_B \leq \mathbb{1}_B$. We are maximizing a linear function over a convex space, so the maximum always lies on the boundary of the space, i.e. at a projector $\Pi_B$. It follows that there exists a projector $\Pi_B$ satisfying (40) with $\varepsilon_3 \leq 2\varepsilon_2$.

**3 → 4:**   Let $\langle\psi_0|U_{\mathrm{code}}|\psi_0\rangle = e^{i\phi}\cos\theta$. We can then define the orthogonal, unnormalized states

$$|\psi_+\rangle = |\psi_0\rangle + e^{-i\phi} U_{\mathrm{code}} |\psi_0\rangle,$$

and

$$|\psi_-\rangle = |\psi_0\rangle - e^{-i\phi} U_{\mathrm{code}} |\psi_0\rangle.$$

Let $\Pi_B$ be a projector satisfying the inequalities $\langle\psi_+| V^\dagger \Pi_B V |\psi_+\rangle / \langle\psi_+|\psi_+\rangle \geq 1 - \varepsilon_3$ and $\langle\psi_-| V^\dagger \Pi_B V |\psi_-\rangle / \langle\psi_-|\psi_-\rangle \leq \varepsilon_3$. We then define the unitary operator $U_B = e^{i\phi}(2\Pi_B - \mathbb{1}_B)$. We have

$$\langle\psi_0|U_{\mathrm{code}}^\dagger V^\dagger U_B V |\psi_0\rangle = \frac{1}{4}(\langle\psi_+| - \langle\psi_-|) V^\dagger (2\Pi_B - \mathbb{1}_B) V (|\psi_+\rangle + |\psi_-\rangle) \tag{218}$$

$$= 1 - \delta, \tag{219}$$

where

$$\mathrm{Re}(\delta) = 1 + \frac{1}{4}\langle\psi_+|\psi_+\rangle - \frac{1}{4}\langle\psi_-|\psi_-\rangle - \frac{1}{2}\langle\psi_+|V^\dagger \Pi_B V|\psi_+\rangle + \frac{1}{2}\langle\psi_-|V^\dagger \Pi_B V|\psi_-\rangle$$

$$\leq 2\varepsilon_3. \tag{220}$$

Hence

$$\left\|V U_{\mathrm{code}} |\psi_0\rangle - U_B V |\psi_0\rangle\right\| = \sqrt{2 - 2\mathrm{Re}\left(\langle\psi_0|U_{\mathrm{code}}^\dagger V^\dagger U_B V|\psi_0\rangle\right)} \leq 2\sqrt{\varepsilon_3}. \tag{221}$$

**4 → 1:** Let $|\psi\rangle = U_{\text{code}}|\psi_0\rangle$. Then by the second Fuchs-van de Graaf inequality and monotonicity under partial traces,

$$\left\|\text{tr}_B[V|\psi\rangle\langle\psi|V^\dagger] - \text{tr}_B[U_B V|\psi_0\rangle\langle\psi_0|V^\dagger U_B^\dagger]\right\|_1 \leq 2\varepsilon_4. \tag{222}$$

However, if we define $\omega_{\overline{B}} = \text{tr}_B[V|\psi_0\rangle\langle\psi_0|V^\dagger]$ then this is exactly (38), with $\varepsilon_1 = 2\varepsilon_4$. $\qquad\square$

**Proof of Theorem 3.7**

*Proof.* By Uhlmann's theorem, Condition 1 is equivalent to the fidelity lower bound

$$F(\text{tr}_{BR}[V U_{\mathbf{b}}|\Psi\rangle\langle\Psi|U_{\mathbf{b}}^\dagger V^\dagger], \text{tr}_{BR}[V|\Psi\rangle\langle\Psi|V^\dagger]) \geq \sqrt{1-\varepsilon_1^2}. \tag{223}$$

Theorem 3.7 then follows immediately by applying the Fuchs-van de Graaf inequalities relating the Schatten 1-norm and the fidelity. $\qquad\square$

**Proof of Theorem 3.11**

*Proof.* We first prove each equivalence in turn.

**1 → 2:** For any $U_{\mathbf{b}}$, we have

$$\max_{U_B}\min_{|\Psi\rangle}\text{Re}\left[\langle\Psi|V^\dagger U_B^\dagger V U_{\mathbf{b}}|\Psi\rangle\right] \leq \min_{|\Psi\rangle}\max_{U_B}\text{Re}\left[\langle\Psi|V^\dagger U_B^\dagger V U_{\mathbf{b}}|\Psi\rangle\right], \tag{224}$$

where on the left hand side we are minimizing with respect to $|\Psi\rangle$ (in a way that can depend on $U_B$) before maximizing with respect to $U_B$, and on the right we do the opposite. Hence the left hand side is related to the error in Condition 2, where we consider the one $U_B$ with the best error for the worst-case state $|\Psi\rangle$, and the right hand side is related to Condition 1, where we allow each $U_B$ to depend on the state $|\Psi\rangle$. Thus Condition 2 manifestly implies Condition 1 with $\varepsilon_1 \leq \varepsilon_2$. To show the reverse implication, we need (224) to be approximately saturated.

The first step is to note that since

$$\langle\Psi|V^\dagger U_B^\dagger V U_{\mathbf{b}}|\Psi\rangle = \text{tr}[\rho_{\text{code}}V^\dagger U_B^\dagger V U_{\mathbf{b}}], \tag{225}$$

where $\rho_{\text{code}} = \text{tr}_{\overline{R}}|\Psi\rangle\langle\Psi|$, we can replace the minimization over $|\Psi\rangle$ by a minimization over density matrices $\rho_{\text{code}}$. By Hölder's inequality, we also have

$$\max_{U_B}\text{Re}(\text{tr}[\rho_{\text{code}}V^\dagger U_B^\dagger V U_{\mathbf{b}}]) = \left\|\text{tr}_{\overline{B}}[V U_{\mathbf{b}}\rho_{\text{code}}V^\dagger]\right\|_1 = \max_{\|O_B\|\leq 1}\text{Re}(\text{tr}[\rho_{\text{code}}V^\dagger O_B^\dagger V U_{\mathbf{b}}]), \tag{226}$$

where the maximization is now over operators $O_B$ with operator norm $\|O_B\| \leq 1$.

Now by von Neumann's minimax theorem,

$$\max_{\|O_B\|\leq 1}\min_{\rho_{\text{code}}}\text{Re}(\text{tr}[\rho_{\text{code}}V^\dagger O_B^\dagger V U_{\mathbf{b}}]) = \min_{\rho_{\text{code}}}\max_{\|O_B\|\leq 1}\text{Re}(\text{tr}[\rho_{\text{code}}V^\dagger O_B^\dagger V U_{\mathbf{b}}]), \tag{227}$$

since $\text{Re}(\text{tr}[\rho_{\text{code}}V^\dagger O_B^\dagger V U_{\mathbf{b}}])$ is a bilinear function and we are minimizing and maximizing over convex spaces.

For any $U_{\mathbf{b}}$, we therefore have a state-independent operator $O_B$ with $\|O_B\| \leq 1$ such that for all states $|\Psi\rangle$

$$\text{Re}(\langle\Psi|V^\dagger O_B^\dagger V U_{\mathbf{b}}|\Psi\rangle) \geq 1 - \frac{1}{2}\varepsilon_1^2. \tag{228}$$

However, we don't yet know that $O_B$ is unitary. Let $O_B$ have a singular value decomposition $O_B = \sum_i \lambda_i |i_R\rangle \langle i_L|$ with $0 \leq \lambda_i \leq 1$, and let $|\Psi\rangle$ have the Schmidt decomposition $|\Psi\rangle = \sum_j \sqrt{p_j} |\psi_j\rangle_{\text{code}} |j\rangle_{\overline{R}}$. Then

$$\text{Re}\left(\langle\Psi| V^\dagger O_B^\dagger V U_{\mathbf{b}} |\Psi\rangle\right) = \text{Re}\left(\sum_{i,j} \lambda_i p_j \langle\psi_j|V^\dagger|i_R\rangle \langle i_L|V U_{\mathbf{b}}|\psi_j\rangle\right) \tag{229}$$

$$\leq \sqrt{\left[\sum_{i,j} \lambda_i p_j \left|\langle\psi_j|V^\dagger|i_R\rangle\right|^2\right]\left[\sum_{i,j} \lambda_i p_j \left|\langle i_L|V U_{\mathbf{b}}|\psi_j\rangle\right|^2\right]}, \tag{230}$$

where we have used the Cauchy-Schwarz inequality. Since

$$\sum_{i,j} \lambda_i p_j \left|\langle\psi_j|V^\dagger|i_R\rangle\right|^2 \leq \sum_{i,j} p_j \left|\langle\psi_j|V^\dagger|i_R\rangle\right|^2 = 1, \tag{231}$$

and

$$\sum_{i,j} \lambda_i p_j |\langle i_L|V U_{\mathbf{b}}|i_L\rangle|^2 \leq \sum_{i,j} p_j |\langle i_L|V U_{\mathbf{b}}|i_L\rangle|^2 = 1, \tag{232}$$

we have

$$\sum_{i,j}(1-\lambda_i)p_j \left|\langle\psi_j|V^\dagger|i_R\rangle\right|^2 \leq \varepsilon_1^2, \tag{233}$$

and

$$\sum_{i,j}(1-\lambda_i)p_j |\langle i_L|V U_{\mathbf{b}}|i_L\rangle|^2 \leq \varepsilon_1^2. \tag{234}$$

Now let $U_B = \sum_i |i_R\rangle \langle i_L|$. If we again apply the Cauchy-Schwarz inequality, we find

$$\text{Re}\left(\langle\Psi| V^\dagger [O_B^\dagger - U_B^\dagger]V U_{\mathbf{b}} |\Psi\rangle\right)$$

$$\leq \sqrt{\left[\sum_{i,j}(1-\lambda_i)p_j |\langle i_L|V U_{\mathbf{b}}|i_L\rangle|^2\right]\left[\sum_{i,j}(1-\lambda_i)p_j \left|\langle\psi_j|V^\dagger|i_R\rangle\right|^2\right]} \tag{235}$$

$$\leq \varepsilon_1^2. \tag{236}$$

Thus

$$\text{Re}\left(\langle\Psi|V^\dagger U_B^\dagger V U_{\mathbf{b}}|\Psi\rangle\right) \geq 1 - \frac{3}{2}\varepsilon_1^2, \tag{237}$$

and $\varepsilon_2 \leq \sqrt{3}\varepsilon_1$.

**2 → 3:**  As in Section 4.2, we define the isometry $W_B : \mathcal{H}_B \to \mathcal{H}_B \otimes \mathcal{H}_{U_{\mathbf{b}}}$ such that

$$W_B V |\Psi\rangle = \int dU_{\mathbf{b}} |U_{\mathbf{b}}\rangle U_B V |\Psi\rangle. \tag{238}$$

We also identify $\mathcal{H}_\mathbf{a} \otimes \mathcal{H}_\mathbf{r} \subseteq \mathcal{H}_{\mathbf{U_b}}$ with the subspace of $\mathcal{H}_{\mathbf{U_b}}$ associated to the fundamental representation.[27] From (71), we have

$$VW_\mathbf{b}|\Psi\rangle_{\mathbf{b\bar{b}\bar{R}}} = \int dU_\mathbf{b}|U_\mathbf{b}\rangle VU_\mathbf{b}|\Psi\rangle_{\mathbf{b\bar{b}\bar{R}}} = V|\Psi\rangle_{\mathbf{a\bar{b}\bar{R}}}|\mathrm{MAX}\rangle_{\mathbf{br}}. \tag{239}$$

By Condition 2, we therefore have

$$\mathrm{Re}\left(\langle\Psi|_{\mathbf{a\bar{b}\bar{R}}}\langle\mathrm{MAX}|_{\mathbf{br}}V^\dagger W_B V|\Psi\rangle_{\mathbf{b\bar{b}\bar{R}}}\right) = \mathrm{Re}\left(\langle\Psi|_{\mathbf{b\bar{b}\bar{R}}}W_\mathbf{b}^\dagger V^\dagger W_B V|\Psi\rangle_{\mathbf{b\bar{b}\bar{R}}}\right) \tag{240}$$

$$= \int dU_\mathbf{b}\mathrm{Re}\left(\langle\Psi|U_\mathbf{b}^\dagger V^\dagger U_B V|\Psi\rangle\right) \tag{241}$$

$$\geq 1 - \frac{1}{2}\varepsilon_2^2. \tag{242}$$

Hence, by the second Fuchs-van de Graaf inequality, we have

$$\left\|\mathrm{tr}_{B\bar{B}\mathbf{r}}[W_B V\rho_\mathrm{code}V^\dagger W_B^\dagger] - \rho_\mathbf{a}\right\|_1 \leq 2\varepsilon_2, \tag{243}$$

where $\rho_\mathbf{a} = \mathrm{tr}_{\bar{\mathbf{b}}}\rho_\mathrm{code}$. This is exactly Condition 3.

$3 \to 1$: Let $O_B = W_B^\dagger U_\mathbf{b}W_B$. Note that $\|O_B\| \leq 1$ but $O_B$ is not in general unitary. From Condition 3 and Hölder's inequality, we know that, for all $\rho_\mathrm{code}$

$$\left|\mathrm{tr}[(V^\dagger O_B V - U_\mathbf{b})\rho_\mathrm{code}]\right| \leq \varepsilon_3. \tag{244}$$

Hence, the operator norm of both the Hermitian and anti-Hermitian parts of $(V^\dagger O_B V - U_\mathbf{b})$ is bounded by $\varepsilon_3$ and, by the triangle inequality,

$$\|V^\dagger O_B V - U_\mathbf{b}\| \leq 2\varepsilon_3. \tag{245}$$

Thus

$$\mathrm{Re}\left(\langle\Psi|(U_\mathbf{b}^\dagger - V^\dagger O_B^\dagger V)U_\mathbf{b}|\Psi\rangle\right) \leq 2\varepsilon_3, \tag{246}$$

and

$$\mathrm{Re}\left(\langle\Psi|V^\dagger O_B^\dagger V U_\mathbf{b}|\Psi\rangle\right) \geq 1 - 2\varepsilon_3. \tag{247}$$

But, as discussed in the proof that Condition 1 implies Condition 2, the maximum of (247) over all operators $\|O_B\| \leq 1$ is always achieved by a unitary $U_B$. Hence $\varepsilon_1 \leq 2\sqrt{\varepsilon_3}$. $\qquad\square$

## A.4 Approximate and non-isometric codes

**Preliminary lemmas**

**Lemma A.11** (Levy's lemma (see, e.g. [56])). *Given a function $f : \mathbb{S}^d \to \mathbb{R}$ with Lipschitz constant $K$, and a random point $\phi$ on the $d$-dimensional sphere $\mathbb{S}^d$, then for any $\varepsilon > 0$ we have $|f(\phi) - \langle f\rangle| \leq \varepsilon$ with probability*

$$p \geq 1 - 2\exp\left[\frac{-(d+1)\varepsilon^2}{9\pi^3 K^2}\right]. \tag{248}$$

---

[27]Recall from the discussion in Footnote 26 that we can trivially replace $W_B$ by an isometry $\mathcal{H}_B \to \mathcal{H}_\mathbf{b} \otimes \mathcal{H}_E$ between finite-dimensional Hilbert spaces as in the statement of Condition 3. For notational clarity, however, we will maintain the distinction between $\mathcal{H}_\mathbf{a}$ and $\mathcal{H}_\mathbf{b}$.

**Lemma A.12** (Product state $\eta$-net). *There exists a discrete set $S$ of product states $|\psi^i\rangle = |\psi_1^i\rangle \ldots |\psi_n^i\rangle \in [\mathbb{C}^2]^{\otimes n}$ with size*

$$|S| \leq \frac{2^{4n}n^{3n}(4+\eta^2/n^2)^n}{\eta^{3n}}, \tag{249}$$

*such that any (normalized) product state $|\psi\rangle = |\psi_1\rangle \ldots |\psi_n\rangle$ satisfies $\||\psi\rangle - |\psi^i\rangle\| \leq \eta$ for some $|\psi^i\rangle \in S$.*

*Proof.* We first construct an $[\eta/n]$-net for the single-qubit Hilbert space $\mathbb{C}^2$.[28] To do so, we consider a maximal set of states where all pairs of states are separated by a distance at least $\eta/n$. Clearly, this is an $[\eta/n]$-net. However, we can upper bound the number of states in the set using a volume counting argument: if each state lies at the center of a sphere of radius $\eta/2n$, then none of the spheres can intersect. Therefore the total volume of all the spheres, $|S|V_4(\eta/2n)^4$ with $V_4$ the volume of the unit 4-sphere, is upperbounded by the difference in volume of two spheres centered on the origin, one with radius $1 + \eta/2n$ and the other with radius $1 - \eta/2n$. This bounds the number of points by

$$|S| \leq \frac{(1+\eta/2n)^4 - (1-\eta/2n)^4}{(\eta/2n)^4} = \frac{8 + 2\eta^2/n^2}{(\eta/2n)^3}. \tag{250}$$

To construct the set $S$ we simply take all possible products of states in this $[\eta/n]$-net. By the triangle inequality, for any product states $|\psi\rangle = |\psi_1\rangle \ldots |\psi_n\rangle$ and $|\phi\rangle = |\phi_1\rangle \ldots |\phi_n\rangle$, we have

$$\||\psi\rangle - |\phi\rangle\| \leq \sum_i \||\psi_i\rangle - |\phi_i\rangle\|. \tag{251}$$

It follows immediately that $S$ is an $\eta$-net as desired. $\qquad\square$

**Lemma A.13** (Norm bound). *Let $V$ be defined as in Theorem 5.1 and let $|\psi\rangle$ be any (normalized) state. Then, for any $\delta > 0$, we have*

$$1 - \delta - 2^{-m-1} \leq \|V|\psi\rangle\| \leq 1 + \delta, \tag{252}$$

*with probability*

$$p \geq 1 - 2\exp[-\frac{2^{m+1}\delta^2}{9\pi^3}]. \tag{253}$$

*Proof.* We first estimate $\langle\|V|\psi\rangle\|\rangle$. Since the map $x \to x^2$ is convex, we have

$$\langle\|V|\psi\rangle\|\rangle^2 \leq \langle\|V|\psi\rangle\|^2\rangle \tag{254}$$

$$\leq 2^{n-m}\int dU \langle\psi|U^\dagger|0\rangle\langle0|^{\otimes(n-m)}U|\psi\rangle \tag{255}$$

$$\leq 2^{-m}\langle\psi|\psi\rangle\operatorname{tr}[|0\rangle\langle0|^{\otimes(n-m)}\otimes\mathbb{1}_{2^m}] = 1. \tag{256}$$

To obtain a lower bound on the average norm, we use the fact that $x \geq 3x^2/2 - x^4/2$ for all $x \geq 0$. Hence

$$\langle\|V|\psi\rangle\|\rangle \geq \frac{3}{2}\langle\|V|\psi\rangle\|^2\rangle - \frac{1}{2}\langle\|V|\psi\rangle\|^4\rangle \tag{257}$$

$$\geq \frac{3}{2} - 2^{2n-2m-1}\int dU \langle\psi|U^\dagger|0\rangle\langle0|^{\otimes(n-m)}U|\psi\rangle^2 \tag{258}$$

$$\geq \frac{3}{2} - \frac{2^{n-2m-1}}{2^n+1}\left(\operatorname{tr}[|0\rangle\langle0|^{\otimes(n-m)}\otimes\mathbb{1}_{2^m}]^2 + \operatorname{tr}[|0\rangle\langle0|^{\otimes(n-m)}\otimes\mathbb{1}_{2^m}]\right) \tag{259}$$

$$\geq 1 - 2^{-m-1}. \tag{260}$$

---

[28]One can mildly improve the scaling here by only constructing a net for qubit states up to a global phase. We don't do so primarily out of laziness, but hopefully also to make the proof slightly more readable.

In (259), we used the standard result (see e.g. [56])

$$\int dU U \, |\psi\rangle \langle\psi| \, U^\dagger \otimes U \, |\psi\rangle \langle\psi| \, U^\dagger = \frac{\mathbb{1} + F_{\text{SWAP}}}{2^{2n} + 2^n}, \tag{261}$$

where the operator $F_{\text{SWAP}}$ permutes the two copies of $[\mathbb{C}^2]^{\otimes n}$. We now have exponentially close upper and lower bounds on $\langle \|V \, |\psi\rangle\| \rangle$.

To bound the fluctuations, we note that, by the triangle inequality,

$$\|2^{(n-m)/2} \langle 0|^{\otimes(n-m)} U_1 \, |\psi\rangle\| - \|2^{(n-m)/2} \langle 0|^{\otimes(n-m)} U_2 \, |\psi\rangle\| \leq 2^{(n-m)/2} \|(U_1 - U_2) \, |\psi\rangle\|. \tag{262}$$

Hence $\|V \, |\psi\rangle\|$ is a Lipschitz continuous function of $U \, |\psi\rangle$ with Lipschitz constant $2^{(n-m)/2}$. We can then use Lemma A.11 to obtain

$$\text{Prob}(|\|V \, |\psi\rangle\| - \langle\|V \, |\psi\rangle\|\rangle| \geq \delta) \leq 2 \exp[-\frac{2^{m+1}\delta^2}{9\pi^3}]. \tag{263}$$

Combining (263) with (256) and (260) completes the proof. $\qquad\square$

**Lemma A.14** (Norm-squared bound). *Let $V$ be defined as in Theorem 5.1 and let $|\psi\rangle$ be any (normalized) state. Then, for any positive $\delta < 1$, we have*

$$\left| \|V \, |\psi\rangle\|^2 - 1 \right| \leq 3\delta, \tag{264}$$

*with probability*

$$p \geq 1 - 2 \exp[-\frac{2^{m+1}\delta^2}{9\pi^3}]. \tag{265}$$

*Proof.* The result follows almost directly from Lemma A.13. We first note that (265) is trivial unless $\delta > 2^{-m}$. Hence

$$1 - 3\delta < (1 - \delta - 2^{-m-1})^2 \leq \|V \, |\psi\rangle\|^2 \leq (1 + \delta)^2 < 1 + 3\delta. \tag{266}$$

$\qquad\square$

**Proof of Theorem 5.1**

*Proof.* For any two states $|\psi\rangle, |\phi\rangle$, we have

$$\begin{aligned} 4\langle\phi|V^\dagger V|\psi\rangle &= \|V(|\psi\rangle + |\phi\rangle)\|^2 - \|V(|\psi\rangle - |\phi\rangle)\|^2 + i\|V(|\psi\rangle + i|\phi\rangle)\|^2 - i\|V(|\psi\rangle - i|\phi\rangle)\|^2 \\ &\approx \||\psi\rangle + |\phi\rangle\|^2 - \||\psi\rangle - |\phi\rangle\|^2 + i\||\psi\rangle + i|\phi\rangle\|^2 - i\||\psi\rangle - i|\phi\rangle\|^2 \\ &\approx 4\langle\phi|\psi\rangle, \end{aligned} \tag{267}$$

where the approximation is valid with high probability thanks to Lemma A.14. To be precise, we can use the union bound to see that with probability

$$p \geq 1 - 8 \exp[-\frac{2^{m+1}\delta^2}{9\pi^3}], \tag{268}$$

we have

$$\begin{aligned} \left| \langle\phi|V^\dagger V|\psi\rangle - \langle\phi|\psi\rangle \right| &\leq \frac{3\delta}{4} \left[ \||\psi\rangle + |\phi\rangle\|^2 + \||\psi\rangle - |\phi\rangle\|^2 + \||\psi\rangle + i|\phi\rangle\|^2 + \||\psi\rangle - i|\phi\rangle\|^2 \right] \\ &\leq 12\delta. \end{aligned} \tag{269}$$

To extend this argument to all product states we use Lemma A.12. One can show $\langle\phi|V^{\dagger}V|\psi\rangle$ is a Lipschitz continuous function of $|\psi\rangle$ for fixed $|\phi\rangle$ (and vice versa) with Lipschitz constant $2^{n-m}$ as follows.

$$
\begin{aligned}
|\langle\phi|V^{\dagger}V|\psi\rangle - \langle\phi|V^{\dagger}V|\psi'\rangle| &\leq \|V^{\dagger}V|\phi\rangle\| \cdot \||\psi\rangle - |\psi'\rangle\| \\
&\leq 2^{n-m}\||\psi\rangle - |\psi'\rangle\|,
\end{aligned}
\tag{270}
$$

where the first inequality follows from Cauchy-Schwarz, and the second follows directly from the definition of $V$. It is therefore sufficient to show that

$$
\left|\langle\psi_i|V^{\dagger}V|\psi_j\rangle - \langle\psi_i|\psi_j\rangle\right| \leq \frac{\varepsilon}{2},
\tag{271}
$$

for all pairs of states $|\psi_i\rangle, |\psi_j\rangle$ in an $\eta$-net for product states with $\eta \leq 2^{m-n-3}\varepsilon$. To see this, note that, for any pair of states $|\psi\rangle, |\phi\rangle$, there exist states $|\psi_i\rangle, |\psi_j\rangle$ in the $\eta$-net such that by the triangle inequality

$$
\begin{aligned}
|\langle\phi|V^{\dagger}V|\psi\rangle - \langle\phi|\psi\rangle| \leq\ & |\langle\phi|V^{\dagger}V|\psi\rangle - \langle\psi_i|V^{\dagger}V|\psi\rangle| + |\langle\psi_i|V^{\dagger}V|\psi\rangle - \langle\psi_i|V^{\dagger}V|\psi_j\rangle| \\
& + \left|\langle\psi_i|V^{\dagger}V|\psi_j\rangle - \langle\psi_i|\psi_j\rangle\right| + |\langle\psi_i|\psi_j\rangle - \langle\phi|\psi_i\rangle| + |\langle\phi|\psi_j\rangle - \langle\phi|\psi\rangle| \\
\leq\ & 2(2^{n-m}\eta) + \frac{\varepsilon}{2} + 2\eta \leq \varepsilon.
\end{aligned}
\tag{272}
$$

From Lemma A.12, we know that the number of states needed for such a net is at most

$$
|S| \leq \frac{2^{4n}n^{3n}(4 + \eta^2/n^2)^n}{\eta^{3n}} \leq 2^{15n+3n(n-m)+1}n^{3n}.
\tag{273}
$$

We can therefore apply the union bound to (268) for all pairs of $|\psi_i\rangle$, using that the number of pairs is bounded by $|S|^2$. If we substitute $\varepsilon = 24\delta$ in (269), this leads to the desired result. $\quad\square$

# B  Non-theorems and their counterexamples

While we have emphasized throughout this paper that Theorem 4.2 is a natural abstraction of the core features of holographic codes, it is reasonable to ask whether there exist other interesting theorems with the same basic interpretation, but different technical details. For example:

1. In Condition (2), we insisted that the holographic entropy prescription (1) should apply not just for the state $|\Psi\rangle$ but also for states related to $|\Psi\rangle$ by a product of local unitaries. While this should always be true in quantum gravity whenever $|\Psi\rangle$ itself satisfies (1),[29] it is clearly a stronger condition for general codes. Is it perhaps unnecessarily strong?

2. Furthermore, did we even need to talk about arbitrary product unitaries in Condition (1)? Couldn't we just consider *local* unitaries acting on a single bulk site?

3. In Condition (4), we minimized the generalized entropy over all possible subsets of $\{b_1 \ldots b_n\}$. But bulk entropies can be defined much more generally for any subsystem in any tensor product decomposition of the bulk Hilbert space, as can areas using Definition 2.1. Can we prove minimality over all bulk subsystems, including ones that don't commute with the subsystems $\mathcal{H}_{b_i}$?

---

[29]This is because gravitational replica trick calculations depend only on the entanglement structure of $|\Psi\rangle$, and not on the specific bulk quantum state.

4. Finally, 4.2 assumes that the bulk factorizes into a tensor product of local subsystems. Is there a theorem like it for more general bulk von Neumann algebras, as in Theorem 5.1 of [1]?

The answer to all of these questions is no. At least in their most obvious versions, none of these other potentials theorems actually exist. In the following subsections, we will present a counterexample to each one in turn.

## B.1  Counterexample with a holographic entropy formula for only the state itself

Suppose we have an isometry $V$, subsystem $\mathcal{H}_b$ and a state $|\Psi\rangle$, such that

$$S(BR)_{V|\Psi\rangle} = A_B(\mathbf{b}) + S(\mathbf{b}R)_{|\Psi\rangle}. \tag{274}$$

However, unlike in Condition 2 of Theorem 4.2, we make no assumptions about the entropy $S(BR)_{VU_{\mathbf{b}}U_{\bar{\mathbf{b}}}|\Psi\rangle}$ for product unitaries $U_{\mathbf{b}}, U_{\bar{\mathbf{b}}}$. As we shall see, on its own this is insufficient to derive Condition 1, or anything similar to it.

As a counterexample, consider the isometry $V : \mathcal{H}_{b_1} \to \mathcal{H}_B \otimes \mathcal{H}_{\overline{B}}$ (for simplicity we consider only a single bulk subsystem), mapping the qudit $\mathcal{H}_{b_1}$ of dimension $d_{b_1}$ to two qudits, $\mathcal{H}_B$ and $\mathcal{H}_{\overline{B}}$ with much larger dimension, such that

$$V|j\rangle_{b_1} = |\phi_j\rangle_{B\overline{B}} := \begin{cases} \frac{1}{\sqrt{D}}\sum_{i=1}^{D} |i\rangle_B |i\rangle_{\overline{B}}, & j = 0, \\ \sum_{i=D+1}^{D+d} c_i |i\rangle_B |i+jd\rangle_{\overline{B}}, & j > 0, \end{cases} \tag{275}$$

where $c_i$ are set of arbitrary normalized coefficients. All $|\phi_j\rangle$ are orthogonal, so $V$ is indeed an isometry.

By a judicious choice of parameters $D, d, c_i, d_{b_1}$, we can choose

$$S(B)_{|\phi_0\rangle} = A_B(b_1) + S(b_1)_0, \tag{276}$$
$$S(B)_{|\phi_{j>0}\rangle} \neq A_B(b_1) + S(b_1)_j. \tag{277}$$

Indeed, let $S := S(B)_{|\phi_j\rangle}$, determined by the parameters $c_i$. Then

$$A_B(b_1) := S(Br_1)_{|CJ\rangle} = \log d_{b_1} + \frac{1}{d_{b_1}}\left(\log D + (d_{b_1} - 1)S\right). \tag{278}$$

If

$$S = \log\left(Dd_{b_1}^{-\frac{d_{b_1}}{d_{b_1}-1}}\right), \tag{279}$$

which is achievable by many choices of $D, d, c_i$, then

$$S(B)_{V|\psi_0\rangle} = \log D = A_B(b_1) = A_B(b_1) + S(b_1)_{|\psi_0\rangle}, \tag{280}$$

ensuring (276).

However, since

$$S(B)_{V|\psi_{j\neq 0}\rangle} = S \neq S(B)_{V|\psi_0\rangle}, \tag{281}$$

then any unitary $U_{b_1}$ such that $U_{b_1}|\psi_0\rangle = |\psi_{j\neq 0}\rangle$ will change the entropy on $B$, and hence cannot be reconstructible on $B$. Since there is only a single bulk subsystem, the unitary $U_{b_1}$ is automatically "local", thus ruling out any version of Condition 1.

### B.2 Counterexample with reconstruction possible only for local unitaries

Another naive possibility is that *local* unitaries $U_{b_i}$ are sufficient in Condition 1, without the need for more general product unitaries. For example, one might wonder whether the following is true:

**Untrue theorem** (local unitaries). *The following two statements are equivalent.*

1. (Complementary Recovery) *For all $U_{b_i}$, if $b_i \in \mathbf{b}$ (respectively if $b_i \in \bar{\mathbf{b}}$) then there exists a unitary operator $U_{BR}$ (respectively $U_{\bar{B}\bar{R}}$) such that,*

$$U_{BR} V |\Psi\rangle = V U_{b_i} |\Psi\rangle \, , \, (\text{respectively } U_{\bar{B}\bar{R}} V |\Psi\rangle = V U_{b_i} |\Psi\rangle) \, . \tag{282}$$

2. (Holographic Entropy Formula) *For all $U_{b_i}$,*

$$S(BR)_{V U_{b_i} |\Psi\rangle} = A_B(b) + S(bR)_{U_{b_i} |\Psi\rangle} \, . \tag{283}$$

Let us first construct an example that satisfies Condition 1, but not Condition 2, and then we will construct an example that satisfies Condition 2 but not Condition 1. Let $\mathcal{H}_{b_1}$ and $\mathcal{H}_{b_2}$ be qubits, with $\mathcal{H}_B$ and $\mathcal{H}_{\bar{B}}$ qudits for large $d$. Let

$$V |0\rangle_{b_1} |0\rangle_{b_2} = |0\rangle_B |0\rangle_{\bar{B}} \, , \tag{284}$$

$$V |1\rangle_{b_1} |0\rangle_{b_2} = |1\rangle_B |0\rangle_{\bar{B}} \, , \tag{285}$$

$$V |0\rangle_{b_1} |1\rangle_{b_2} = |2\rangle_B |0\rangle_{\bar{B}} \, , \tag{286}$$

$$V |1\rangle_{b_1} |1\rangle_{b_2} = \sum_{i=1}^{d-1} |i\rangle_B |i\rangle_{\bar{B}} \, . \tag{287}$$

Finally let $|\psi\rangle = |0\rangle |0\rangle$. For any $U_{b_1}$ we have $U_{b_1} |\psi\rangle \in \text{span}\{|00\rangle, |10\rangle\}$. Similarly, for any $U_{b_2}$ we have $U_{b_2} |\psi\rangle \in \text{span}\{|00\rangle, |01\rangle\}$. Within both these subspaces, $\text{tr}_B[V |\phi\rangle \langle\phi| V^\dagger] = |0\rangle \langle 0|_{\bar{B}}$ is independent of $|\phi\rangle$; hence all local operators are reconstructible on $\mathcal{H}_B$ and Condition 1 is satisfied for $\mathbf{b} = \{b_1, b_2\}$. However $S(B)_{|\psi\rangle} = 0$, while $A_B(\mathbf{b}) = S(Br_1 r_2)_{|CJ\rangle} = S(\bar{B})_{|CJ\rangle} = O(\log d)$. So Condition 2 is not satisfied.

Note that the product unitary $X_{b_1} X_{b_2}$ cannot be reconstructed on $\mathcal{H}_B$, since

$$V X_{b_1} X_{b_2} |\psi\rangle = V |11\rangle \, , \tag{288}$$

which has a different reduced state on $\mathcal{H}_{\bar{B}}$. Hence neither Condition 1 nor Condition 2 of the true Theorem 4.2 are satisfied.

To see the reverse (an example that satisfies Condition 2 but not Condition 1 of the hypothetical local unitary theorem), let

$$V |0\rangle_{b_1} |0\rangle_{b_2} = \frac{1}{\sqrt{d}} \sum_{i=0}^{d-1} |i\rangle_B |i\rangle_{\bar{B}} \, , \tag{289}$$

$$V |1\rangle_{b_1} |0\rangle_{b_2} = \frac{1}{\sqrt{d}} \sum_{i=0}^{d-1} |i\rangle_B |i+d\rangle_{\bar{B}} \, , \tag{290}$$

$$V |0\rangle_{b_1} |1\rangle_{b_2} = \frac{1}{\sqrt{d}} \sum_{i=0}^{d-1} |i\rangle_B |i+2d\rangle_{\bar{B}} \, , \tag{291}$$

$$V |1\rangle_{b_1} |1\rangle_{b_2} = \sum_{i=0}^{d-1} c_i |i+d\rangle_B |i+3d\rangle_{\bar{B}} \, , \tag{292}$$

for some normalized coefficients $c_i$ such that $V\,|11\rangle$ has entanglement entropy $S$. Again, we can choose $d, S$ such that with $\mathbf{b} = \{b_1, b_2\}$ and for all $U_{b_1}, U_{b_2}$

$$S(B)_{VU_{b_1}|00\rangle} = S(B)_{VU_{b_2}|00\rangle} = S(B)_{V|00\rangle} = \log d\,, \tag{293}$$

and

$$A_B(\mathbf{b}) + S(\mathbf{b})_{U_{b_1}|00\rangle} = A_B(\mathbf{b}) + S(\mathbf{b})_{U_{b_2}|00\rangle} = A_B(\mathbf{b}) = S(Br_1r_2)_{|\text{CJ}\rangle} \tag{294}$$

$$= \log 4 + \frac{1}{4}(3\log d + S) \tag{295}$$

are equal to one another. However, it is immediately obvious that in general the reduced density matrix of $VU_{b_1}\,|00\rangle$ or $VU_{b_2}\,|00\rangle$ on $\mathcal{H}_{\overline{B}}$ is different from that of $V\,|00\rangle$, so Condition 1 of the hypothetical local unitary theorem is violated.

### B.3 Counterexample to minimality over all subsystems

While we have proved that $\text{EW}_{BR}(V\,|\Psi\rangle)$ has minimal generalized entropy $A_B(\mathbf{b}) + S(\mathbf{b}R)_{|\Psi\rangle}$ over subsystems associated to subsets of $\{b_1, \dots b_n\}$, we have *not* shown that it is minimal over *all* subsystems of the bulk Hilbert space. Indeed, to prove that the subsystem $\mathcal{H}_{\mathbf{b}}$ was more minimal than some other subsystem $\mathcal{H}_{\mathbf{b}'}$, we had to use strong subadditivity,

$$S(\mathbf{b}R) + S(\mathbf{b}'R) \geq S([\mathbf{b}\cup\mathbf{b}']R) + S([\mathbf{b}\cap\mathbf{b}']R)\,. \tag{296}$$

In other words, we made explicit use of the fact that there exists a tensor product decomposition

$$\mathcal{H}_{\text{code}} \cong \mathcal{H}_{\mathbf{b}\cap\mathbf{b}'} \otimes \mathcal{H}_{\mathbf{b}\cap\bar{\mathbf{b}}'} \otimes \mathcal{H}_{\bar{\mathbf{b}}\cap\mathbf{b}'} \otimes \mathcal{H}_{\bar{\mathbf{b}}\cap\bar{\mathbf{b}}'}\,. \tag{297}$$

Given two arbitrary subsystems of $\mathcal{H}_{\text{code}}$ (i.e. Hilbert spaces $\mathcal{H}_{\mathbf{b}}$ and $\mathcal{H}_{\mathbf{b}'}$ such that $\mathcal{H}_{\text{code}} \cong \mathcal{H}_{\mathbf{b}} \otimes \mathcal{H}_{\bar{\mathbf{b}}} \cong \mathcal{H}_{\mathbf{b}'} \otimes \mathcal{H}_{\bar{\mathbf{b}}'}$ for some $\mathcal{H}_{\bar{\mathbf{b}}}$ and $\mathcal{H}_{\bar{\mathbf{b}}'}$), no such decomposition will exist, and so our proof of minimality will not work. In general, a decomposition analogous to (297) exists if and only if the superoperators that project operators into the subsystem algebras on $\mathcal{H}_{\mathbf{b}}$, $\mathcal{H}_{\mathbf{b}'}$, $\mathcal{H}_{\bar{\mathbf{b}}}$ and $\mathcal{H}_{\bar{\mathbf{b}}'}$ all commute with one another.[30]

This is not just a failure of our proof method – it is simply not true in general that $\text{EW}_{BR}$ is minimal over all subalgebras. Here is an example. Let $\mathcal{H}_{b_1}$ and $\mathcal{H}_{b_2}$ be qudits with dimension $d$, and let $V$ map them trivially to $B$ and $\overline{B}$ respectively:

$$V\,|i\rangle_{b_1}\,|j\rangle_{b_2} = |i\rangle_B\,|j\rangle_{\overline{B}}\,. \tag{298}$$

It is straightforward to compute $A_B(b_1) = 0$. Now consider a maximally-entangled input state $|\psi\rangle := \sum_i |i\rangle_{b_1}\,|i\rangle_{b_2}/\sqrt{d}$. Clearly, Condition 1 of Theorem 4.2 is satisfied with $\mathbf{b} = \{b_1\}$. The QMS prescription tells us that

$$S(B)_{V|\psi\rangle} = A_B(b_1) + S(b_1)_{|\psi\rangle} = d\log d\,, \tag{299}$$

and moreover that this is minimal relative to the $A_B + S_{|\psi\rangle}$ associated to inputs $\varnothing, \{b_2\}$, and $\{b_1, b_2\}$.

---

[30]Given a decomposition of the form (297), the commutativity of the projectors (e.g. $\mathcal{P}_{\mathbf{b}}(\mathcal{O}) = \text{tr}_{\bar{\mathbf{b}}}[\mathcal{O}] \otimes \mathbb{1}_{\bar{\mathbf{b}}}/d_{\bar{\mathbf{b}}}$) follows trivially from the commutativity of partial traces onto independent subsystems. Conversely, given commuting projectors we can construct a decomposition of the form (297) by defining, e.g., $\mathcal{P}_{\mathbf{b}\cap\mathbf{b}'} = \mathcal{P}_{\mathbf{b}}\mathcal{P}_{\mathbf{b}'} = \mathcal{P}_{\mathbf{b}'}\mathcal{P}_{\mathbf{b}}$.

Now define another subsystem via the following procedure. Let unitary $U$ act as

$$U |j\rangle_{b_1} |j\rangle_{b_2} := \frac{1}{\sqrt{d}} \sum_{k=1}^{d} e^{2\pi i jk/d} |k\rangle_{b_1} |k\rangle_{b_2} , \tag{300}$$

$$U |j\rangle_{b_1} |j'\rangle_{b_2} := |j\rangle_{b_1} |j'\rangle_{b_2} , \quad j \neq j' .$$

Consider all operators of the form

$$U(O_{b_1} \otimes \mathbb{1}_{\bar{b}}) U^\dagger . \tag{301}$$

These form a von Neumann algebra $\mathcal{A}_{b_1'}$ on $\mathcal{H}_{\text{code}}$ with projector

$$\mathcal{P}_{b_1'}(O) = U \operatorname{tr}_{b_2}[U^\dagger O U] \otimes \mathbb{1}_{b_2} U^\dagger . \tag{302}$$

Since the algebra $\mathcal{A}_{b_1'}$ is isomorphic to the algebra $\mathcal{A}_{b_1}$ of operators acting on $\mathcal{H}_{b_1}$, $\mathcal{A}_{b_1'}$ is also the algebra of operators acting on a subsystem $\mathcal{H}_{b_1'}$, which is related to $\mathcal{H}_{b_1}$ by conjugation with $U$. However $\mathcal{P}_{b_1'}$ does not commute with $\mathcal{P}_{b_1}$, so our minimality proof does not apply.

Indeed, as we shall see,

$$A_B(b_1') + S(b_1')_{|\psi\rangle} < S(B)_{V|\psi\rangle} = A_B(b_1) + S(b_1)_{|\psi\rangle} , \tag{303}$$

giving a counterexample to minimality over all possible subsystems. To compute $A_B(b_1')$, we need the reduced density matrix of $\mathcal{H}_B \otimes \mathcal{H}_{r_1'}$ in the Choi-Jamiolkwoski state, which equals the reduced density matrix of $\mathcal{H}_B \otimes \mathcal{H}_{r_1}$ in the state $VU |\text{MAX}\rangle$.

$$
\begin{aligned}
U |\text{MAX}\rangle_{b_1 b_2 r_1 r_2} &= \frac{1}{d} U \sum_{jk} |j\rangle_{b_1} |k\rangle_{b_2} |j\rangle_{r_1} |k\rangle_{r_2} \\
&= \frac{1}{d^{3/2}} \sum_{j\ell} e^{2\pi i j\ell/d} |\ell\rangle_{b_1} |\ell\rangle_{b_2} |j\rangle_{r_1} |j\rangle_{r_2} + \frac{1}{d} \sum_{j\neq k} |j\rangle_{b_1} |k\rangle_{b_2} |j\rangle_{r_1} |k\rangle_{r_2} \\
&= \frac{1}{d} \sum_{jk} |j\rangle_{b_1} |k\rangle_{b_2} |j\rangle_{r_1} |k\rangle_{r_2} \\
&\quad + \frac{1}{d^{3/2}} \sum_{jk} e^{2\pi i jk/d} |k\rangle_{b_1} |k\rangle_{b_2} |j\rangle_{r_1} |j\rangle_{r_2} - \frac{1}{d} \sum_{j} |j\rangle_{b_1} |j\rangle_{b_2} |j\rangle_{r_1} |j\rangle_{r_2} \\
&= |\text{MAX}\rangle + O\left(\frac{1}{d^{1/2}}\right) .
\end{aligned}
\tag{304}
$$

Using Fannes inequality and the fact that trace distances are monotonically decreasing under partial traces, we have

$$A_B(b_1') := S(Br_1')_{|\text{CJ}\rangle} = S(Br_1)_{|\text{CJ}\rangle} + \mathcal{O}((1/\sqrt{d}) \log d) = \mathcal{O}((1/\sqrt{d}) \log d) . \tag{305}$$

All that remains is to compute $S(b_1')_{|\psi\rangle}$. Noting that

$$U |\psi\rangle = \frac{1}{\sqrt{d}} U \sum_{j=1}^{d} |j\rangle_{b_1} |j\rangle_{b_2} = \frac{1}{d} \sum_{j,k=1}^{d} e^{2\pi i jk/d} |k\rangle_{b_1} |k\rangle_{b_2} = |d\rangle_{b_1} |d\rangle_{b_2} , \tag{306}$$

we have

$$S(b_1')_{|\psi\rangle} = S(b_1)_{U|\psi\rangle} = 0 , \tag{307}$$

and therefore

$$A_B(b_1') + S(b_1')_{|\psi\rangle} = \mathcal{O}\left(\frac{\log d}{\sqrt{d}}\right) , \tag{308}$$

which is much less than $A_B(b_1) + S(b_1)_{|\psi\rangle} = \log d$ for large $d$.

### B.4 Counterexample to minimality for algebras with centers

Theorem 4.2 is about bulk Hilbert spaces that factorize into a tensor product of local subsystems. As such, it is comparable to (although much more general than) Theorem 4.1 of [1]. There is however a more general theorem (Theorem 5.1) in [1], where the local subsystems are replaced by local von Neumann algebras. The new feature is that these subalgebras can have a nontrivial center – operators that commute with every operator in the subalgebra. (Indeed a finite-dimensional subalgebra with trivial center – i.e. containing only operators proportional to the identity – is always just the algebra of all operators on some particular subsystem.) In this way, the 'area' of a given surface can become a non-trivial operator that lies in the center of the associated subalgebra, as we expect in AdS/CFT for code spaces where the bulk can have any of multiple distinct possible geometries.

It is natural to hope that a similar generalization of Theorem 4.2 exists where the local subsystems are replaced by local subalgebras. And indeed there exist fairly natural extensions of our definition of area and state-specific product operator reconstruction for which some aspects of Theorem 4.2 continue to hold.

However, there are also significant issues that show up. In particular there are serious problems with trying to prove a version of minimality (Condition 4) for general von Neumann algebras. For any such theorem, at least one of the following must *not* be true:

1. The definition of reconstruction includes state-independent algebraic reconstruction à la Theorem 5.1 of [1] as a special case,

2. The definition of area reduces to the definition in Theorem 5.1 of [1] for cases where that definition is valid,

3. The definition of area reduces to the value given by Definition 2.1 when evaluated on algebras with trivial center.

To see this, consider the isometry

$$
\begin{aligned}
V\,|0\rangle &= |0\rangle_B\,|0\rangle_{\overline{B}}\,, \\
V\,|1\rangle &= 2^{-n/2}\sum_{i=1}^{2^n}|i\rangle_B\,|i\rangle_{\overline{B}}\,.
\end{aligned}
\tag{309}
$$

This is an algebraic QEC code in which $B$ can state-independently reconstruct the algebra $\mathcal{A}_{\mathbf{b}} = \{\mathbb{1}, Z\}$ acting on $\mathcal{H}_{\mathrm{code}}$. This is a commuting subalgebra that is equal to its own commutant $\mathcal{A}_{\bar{\mathbf{b}}} = \mathcal{A}_{\mathbf{b}}$ and is also reconstructible from $\mathcal{H}_{\overline{B}}$. We therefore have a state-independent complementary algebraic QECC, exactly the conditions needed for Harlow's Theorem 5.1.

Harlow's area operator $\hat{A}(\mathbf{b}) \in \mathcal{A}_{\mathbf{b}}$ is

$$
\hat{A}_B(\mathbf{b}) = n\,|1\rangle\langle 1|\,.
\tag{310}
$$

The entropy of $B$ can be computed with the holographic entropy formula

$$
S(B)_{V|\psi\rangle} = \langle\psi|\hat{A}_B(\mathbf{b})|\psi\rangle + S(\mathbf{b})_{|\psi\rangle}\,,
\tag{311}
$$

for all $|\psi\rangle \in \mathcal{H}_{\mathrm{code}}$. Here $S(\mathbf{b})_{|\psi\rangle}$ is the algebraic entropy of the state $|\psi\rangle$ for the subalgebra $\mathcal{A}_{\mathbf{b}}$. Hence in particular

$$
S(B)_{V|1\rangle} = n\,,
\tag{312}
$$

as can be readily verified. However, this generalized entropy is not minimal, even when considering only algebras whose projectors commute with that of $\mathcal{A}_{\mathbf{b}}$. According to Definition 2.1, the area $A_B(\varnothing)$ of the trivial algebra is

$$
A_B(\varnothing) = \frac{n}{2} + \mathcal{O}(n^0)\,,
\tag{313}
$$

while the corresponding entropy vanishes $S(\varnothing)_{|\psi\rangle} = 0$. Hence at large $n$ the trivial algebra has generalized entropy much less than $\mathcal{M}_b$.

It is important to emphasize here that this counterexample does not depend on any specific proposal for the definition of area for general algebras. Instead, it simply used Harlow's definition in [1], applied to a code with complementary state independent recovery, together with Definition 2.1 for the areas of algebras with trivial center, applied to the trivial algebra of operators proportional to the identity. Changing either of these definitions seems a priori undesirable. We comment on possible resolutions in Section 6.3.

## C   Proof of Theorem 5.5

**Lemma C.1.** *Let* $|\psi\rangle, |\phi\rangle \in \mathcal{H}_A \otimes \mathcal{H}_B$ *satisfy* $\||\psi\rangle\|, \||\phi\rangle\| \leq 1$. *Then*

$$\big\| |\psi\rangle\langle\psi| - |\phi\rangle\langle\phi| \big\|_1 \leq 2 \big\| |\psi\rangle - |\phi\rangle \big\|. \tag{314}$$

*Proof.* By explicit diagonalization, we have

$$\big\| |\psi\rangle\langle\psi| - |\phi\rangle\langle\phi| \big\|_1^2 = (\langle\psi|\psi\rangle + \langle\phi|\phi\rangle)^2 - 4|\langle\psi|\phi\rangle|^2 \tag{315}$$

$$\leq (\langle\psi|\psi\rangle + \langle\phi|\phi\rangle)^2 - 4[\mathrm{Re}(\langle\psi|\phi\rangle)]^2 \tag{316}$$

$$\leq 4 \big\| |\psi\rangle - |\phi\rangle \big\|^2. \tag{317}$$

In the last inequality, we expanded $A^2 - B^2 = (A+B)(A-B)$ and then used

$$\langle\psi|\psi\rangle + \langle\phi|\phi\rangle + 2\mathrm{Re}(\langle\psi|\phi\rangle) \leq 4. \tag{318}$$

$\square$

**Lemma C.2.** *Let*

$$\Phi_{\overline{B}\overline{R}}(U_{\mathbf{b}}) := \mathrm{tr}_{BR}\big[ V U_{\mathbf{b}} |\Phi\rangle\langle\Phi| U_{\mathbf{b}}^\dagger V^\dagger \big]. \tag{319}$$

*where* $|\Phi\rangle \in \mathcal{H}_{\mathrm{code}} \otimes \mathcal{H}_R \otimes \mathcal{H}_{\overline{R}}$ *satisfies* $\|V U_{\mathbf{b}} |\Phi\rangle\| \leq 1$ *for all* $U_{\mathbf{b}}$ *and all other terms are defined as in Theorem 5.5. If*

$$S\left( \int dU_{\mathbf{b}} \Phi_{\overline{B}\overline{R}}(U_{\mathbf{b}}) \right) - \int dU_{\mathbf{b}} S(\Phi_{\overline{B}\overline{R}}(U_{\mathbf{b}})) \leq \delta, \tag{320}$$

*for sufficiently small* $\delta > 0$, *then with probability* $p \geq 1 - \kappa$, *there exists a* $U_{BR}$ *such that*

$$\big\| U_{BR} V \hat{U}_{\mathbf{b}} |\Phi\rangle - V U_{\mathbf{b}} |\Phi\rangle \big\|^2 \leq \frac{8\sqrt{2\delta}}{\kappa}. \tag{321}$$

*Proof.* Recall how one proves that

$$S(\rho) \geq \sum_i p_i S(\rho_i), \tag{322}$$

for a density matrix $\rho := \sum_i p_i \rho_i$ that is a mixture of density matrices $\rho_i$. Let $\rho$ be defined on system $A$, and introduce auxiliary system $B$, in joint state

$$\rho_{AB} = \sum_i p_i |i\rangle\langle i|_B \otimes \rho_i. \tag{323}$$

These systems each have entropy

$$S(\rho_A) = S(\rho) \,, \tag{324}$$

$$S(\rho_B) = S\left(\sum_i p_i \, |i\rangle \, \langle i|_B\right) = -\sum_i p_i \log p_i \,, \tag{325}$$

$$S(\rho_{AB}) = \sum_i p_i S(\rho_i) - \sum_i p_i \log p_i \,. \tag{326}$$

The inequality (322) is then simply subadditivity. Explicitly,

$$S(\rho) - \sum_i p_i S(\rho_i) = I(A:B)_{\rho_{AB}} = S_{\text{rel}}(\rho_{AB} || \rho_A \otimes \rho_B) \,. \tag{327}$$

We want to write the analogue of (323), but for the continuous variable $U_{\mathbf{b}}$, with probability measure $dU_{\mathbf{b}}$, rather than the discrete variable $i$. To do so, we need to use the language of operator algebras. We define the von Neumann algebra $\mathcal{A}$ as the direct integral

$$\int^{\oplus} \mathcal{A}_{\overline{B}\overline{R}} \, dU_{\mathbf{b}} \,, \tag{328}$$

where $\mathcal{A}_{\overline{B}\overline{R}}$ is the algebra of operators on $\mathcal{H}_{\overline{B}} \otimes \mathcal{H}_{\overline{R}}$. The analogue of the density matrices $\rho_{AB}$ and $\rho_A \otimes \rho_B$ are the linear functionals $\rho_{U_{\mathbf{b}}\overline{B}\overline{R}}$ and $\rho_{U_{\mathbf{b}}} \otimes \rho_{\overline{B}\overline{R}}$ defined by

$$\rho_{U_{\mathbf{b}}\overline{B}\overline{R}}\left(\int^{\oplus} O_{\overline{B}\overline{R}}(U_{\mathbf{b}}) dU_{\mathbf{b}}\right) = \int dU_{\mathbf{b}} \, \text{tr}\left[O_{\overline{B}\overline{R}}(U_{\mathbf{b}}) \Phi_{\overline{B}\overline{R}}(U_{\mathbf{b}})\right] \,, \tag{329}$$

$$\rho_{U_{\mathbf{b}}} \otimes \rho_{\overline{B}\overline{R}}\left(\int^{\oplus} O_{\overline{B}\overline{R}}(U_{\mathbf{b}}) dU_{\mathbf{b}}\right) = \int dU_{\mathbf{b}} dU_{\mathbf{b}}' \, \text{tr}\left[O_{\overline{B}\overline{R}}(U_{\mathbf{b}}) \Phi_{\overline{B}\overline{R}}(U_{\mathbf{b}}')\right] \,. \tag{330}$$

Note that $\rho_{U_{\mathbf{b}}\overline{B}\overline{R}}$ and $\rho_{U_{\mathbf{b}}} \otimes \rho_{\overline{B}\overline{R}}$ are both subnormalized

$$\rho_{U_{\mathbf{b}}\overline{B}\overline{R}}(\mathbb{1}) = \rho_{U_{\mathbf{b}}} \otimes \rho_{\overline{B}\overline{R}}(\mathbb{1}) = \int dU_{\mathbf{b}} \, \text{tr}\left[\Phi_{\overline{B}\overline{R}}(U_{\mathbf{b}})\right] \leq 1 \,. \tag{331}$$

However, we can extend them to normalized states by defining an isometry $V'_{\overline{B}} : \mathcal{H}_{\overline{B}} \to \mathcal{H}_{\overline{B}'}$ with $d_{\overline{B}'} = d_{\overline{B}} + 1$, as well as similar isometries for $\mathcal{H}_{\overline{R}}$ and $\mathcal{H}_{U_{\mathbf{b}}}$. We then define normalized states $\hat{\rho}_{U_{\mathbf{b}'}\overline{B}'\overline{R}'} = V' \rho_{U_{\mathbf{b}}\overline{B}\overline{R}} V'^{\dagger} + \rho_0$ and $\hat{\rho}_{U_{\mathbf{b}'}} \otimes \hat{\rho}_{\overline{B}'\overline{R}'} = V' \rho_{U_{\mathbf{b}}} \otimes \rho_{\overline{B}\overline{R}} V'^{\dagger} + \rho_0$ with $V' = V'_{\overline{B}} V'_{\overline{R}} V'_{U_{\mathbf{b}}}$ and

$$V'^{\dagger}_{\overline{B}} \rho_0 V'_{\overline{B}} = V'^{\dagger}_{\overline{R}} \rho_0 V'_{\overline{R}} = V'^{\dagger}_{U_{\mathbf{b}}} \rho_0 V'_{U_{\mathbf{b}}} = 0 \,. \tag{332}$$

We then have

$$
\begin{aligned}
S_{\text{rel}}(\hat{\rho}_{U_{\mathbf{b}'}\overline{B}'\overline{R}'} || \hat{\rho}_{U_{\mathbf{b}'}} \otimes \hat{\rho}_{\overline{B}'\overline{R}'}) &= S_{\text{rel}}(\rho_{U_{\mathbf{b}}\overline{B}\overline{R}} || \rho_{U_{\mathbf{b}}} \otimes \rho_{\overline{B}\overline{R}}) \\
&= \int dU_{\mathbf{b}} \, \text{tr}\left[\Phi_{\overline{B}\overline{R}}(U_{\mathbf{b}})\left[\log\left(\int dU_{\mathbf{b}}' \Phi_{\overline{B}\overline{R}}(U_{\mathbf{b}}')\right) - \log \Phi_{\overline{B}\overline{R}}(U_{\mathbf{b}})\right]\right] \\
&= S\left(\int dU_{\mathbf{b}} \Phi_{\overline{B}\overline{R}}(U_{\mathbf{b}})\right) - \int dU_{\mathbf{b}} S(\Phi_{\overline{B}\overline{R}}(U_{\mathbf{b}})) \leq \delta \,.
\end{aligned}
\tag{333}
$$

By Pinsker's inequality, it follows that

$$\left\|\rho_{U_{\mathbf{b}}\overline{B}\overline{R}} - \rho_{U_{\mathbf{b}}} \otimes \rho_{\overline{B}\overline{R}}\right\|_1 = \left\|\hat{\rho}_{U_{\mathbf{b}'}\overline{B}'\overline{R}'} - \hat{\rho}_{U_{\mathbf{b}'}} \otimes \hat{\rho}_{\overline{B}'\overline{R}'}\right\|_1 \leq \sqrt{2\delta} \,. \tag{334}$$

However, we can explicitly evaluate

$$\left\| \rho_{\mathbf{U_b}\overline{B}\overline{R}} - \rho_{\mathbf{U_b}} \otimes \rho_{\overline{B}\overline{R}} \right\|_1 = \int d U_\mathbf{b} \left\| \Phi_{\overline{B}\overline{R}}(U_\mathbf{b}) - \int d U'_\mathbf{b} \Phi_{\overline{B}\overline{R}}(U'_\mathbf{b}) \right\|_1 . \tag{335}$$

Finally, by Markov's inequality, we know that with probability $p \geq 1 - \kappa/2$

$$\left\| \Phi_{\overline{B}\overline{R}}(U_\mathbf{b}) - \int d U'_\mathbf{b} \Phi_{\overline{B}\overline{R}}(U'_\mathbf{b}) \right\|_1 \leq \frac{2\sqrt{2\delta}}{\kappa} , \tag{336}$$

where we can choose $\kappa$ such that $1 \gg \kappa \gg \delta$. Now we can use the triangle inequality to show that with probability $p \geq 1 - \kappa$

$$\left\| \Phi_{\overline{B}\overline{R}}(U_\mathbf{b}) - \Phi_{\overline{B}\overline{R}}(\hat{U}_\mathbf{b}) \right\|_1 \leq \frac{4\sqrt{2\delta}}{\kappa} . \tag{337}$$

If we extend $\Phi_{\overline{B}\overline{R}}(U_\mathbf{b}), \Phi_{\overline{B}\overline{R}}(\hat{U}_\mathbf{b})$ to normalized states $\hat{\Phi}_{\overline{B}'\overline{R}'}(U_\mathbf{b}), \hat{\Phi}_{\overline{B}'\overline{R}'}(\hat{U}_\mathbf{b})$ in the same way as before, we find

$$\left\| \hat{\Phi}_{\overline{B}'\overline{R}'}(U_\mathbf{b}) - \hat{\Phi}_{\overline{B}'\overline{R}'}(\hat{U}_\mathbf{b}) \right\|_1 = \left\| \Phi_{\overline{B}\overline{R}}(U_\mathbf{b}) - \Phi_{\overline{B}\overline{R}}(\hat{U}_\mathbf{b}) \right\|_1 + \left| \|\Phi_{\overline{B}\overline{R}}(U_\mathbf{b})\|_1 - \|\Phi_{\overline{B}\overline{R}}(\hat{U}_\mathbf{b})\|_1 \right| \tag{338}$$

$$\leq \frac{8\sqrt{2\delta}}{\kappa} , \tag{339}$$

where we used the triangle inequality. We then use the first Fuchs-van de Graaf inequality to bound the fidelity

$$F\left( \hat{\Phi}_{\overline{B}\overline{R}}(U_\mathbf{b}), \hat{\Phi}_{\overline{B}\overline{R}}(\hat{U}_\mathbf{b}) \right) \geq 1 - \frac{4\sqrt{2\delta}}{\kappa} . \tag{340}$$

It follows by Uhlmann's theorem (together with the fact that projecting the states back into the original Hilbert space using $V'^\dagger_{\overline{B}} V'^\dagger_{\overline{R}}$ can only decrease the norm) that there exists a unitary $U_{BR}$ such that

$$\left\| U_{BR} V \hat{U}_\mathbf{b} |\Phi\rangle - V U_\mathbf{b} |\Phi\rangle \right\|^2 \leq \frac{8\sqrt{2\delta}}{\kappa} . \tag{341}$$

$\square$

**Lemma C.3.** *For any state $|\psi\rangle \in \mathcal{H}_A \otimes \mathcal{H}_B \otimes \mathcal{H}_C$ and sufficiently small $\varepsilon > 0$, we have*

$$H^\varepsilon_{\min}(A|B)_{|\psi\rangle} \leq S(A|B)_{|\psi\rangle} - 4\varepsilon \log \frac{d_A d_B}{2\varepsilon} . \tag{342}$$

*Proof.* Let $|\tilde{\psi}\rangle$ maximize the min-entropy $H_{\min}(A|B)$ within an $\varepsilon$-ball of $|\psi\rangle$. The state $|\tilde{\psi}\rangle$ may in general be subnormalized; however, as in the proof of Lemma C.2, we can extend the Hilbert spaces $\mathcal{H}_{A/B/C}$ with isometries $V_{A/B/C} : \mathcal{H}_{A/B/C} \to \mathcal{H}'_{A/B/C}$ and define a normalized state $|\tilde{\psi}'\rangle \in \mathcal{H}'_A \otimes \mathcal{H}'_B \otimes \mathcal{H}'_C$ such that

$$V^\dagger_A |\tilde{\psi}'\rangle = V^\dagger_B |\tilde{\psi}'\rangle = V^\dagger_C |\tilde{\psi}'\rangle = |\tilde{\psi}\rangle . \tag{343}$$

We have

$$H^\varepsilon_{\min}(A|B)_{|\psi\rangle} = H_{\min}(A|B)_{|\tilde{\psi}\rangle} \leq S(A|B)_{|\tilde{\psi}\rangle} = S(A|B)_{|\tilde{\psi}'\rangle} . \tag{344}$$

But, by the definition of the generalized fidelity (Definition 5.3),[31] we have

$$|\langle \tilde{\psi}' | \psi \rangle| \geq \sqrt{1 - \varepsilon^2} , \tag{345}$$

---

[31]Because $|\psi\rangle$ is normalized, there is no need to supremize over isometries $V$.

and hence

$$\|\tilde{\psi}'_{AB} - \psi_{AB}\|_1 \le 2\varepsilon. \tag{346}$$

Finally, by Fannes' inequality, we have[32]

$$S(A|B)_{|\psi\rangle} \ge S(A|B)_{|\tilde{\psi}'\rangle} - 4\varepsilon \log \frac{d_A d_B}{2\varepsilon}. \tag{347}$$

$\square$

**Proof** $(1) \implies (2)$:

The proof proceeds analogously to the exact version in Section 4.2. We again have

$$S(BR\mathbf{ar})_{VW_{\mathbf{b}}W_{\bar{\mathbf{b}}}|\Psi\rangle} = A_B(\mathbf{b}) + S(\mathbf{b}R)_{|\Psi\rangle}. \tag{348}$$

For a given $\hat{U}_{\mathbf{b}}, \hat{U}'_{\bar{\mathbf{b}}}$, let

$$W_{BR}^{(\hat{U}_{\mathbf{b}})} = \int dU_{\mathbf{b}} |U_{\mathbf{b}}\rangle \otimes U_{BR}, \tag{349}$$

and

$$W_{\bar{B}\bar{R}}^{(\hat{U}'_{\bar{\mathbf{b}}})} = \int dU'_{\bar{\mathbf{b}}} |U'_{\bar{\mathbf{b}}}\rangle \otimes U'_{\bar{B}\bar{R}}, \tag{350}$$

with $U_{BR}, U'_{\bar{B}\bar{R}}$ satisfying (139) when possible and arbitrarily chosen otherwise.

Let $p(\text{RCVR})$ be the probability that $U_{BR}, U'_{\bar{B}\bar{R}}$ satisfying (139) exist for randomly chosen $U_{\mathbf{b}}, U'_{\bar{\mathbf{b}}}, \hat{U}_{\mathbf{b}}, \hat{U}'_{\bar{\mathbf{b}}}$. Meanwhile let $p(\text{RCVR}|\hat{U}_{\mathbf{b}}, \hat{U}'_{\bar{\mathbf{b}}})$ be the probability that $U_{BR}, U'_{\bar{B}\bar{R}}$ satisfying (139) exist for randomly chosen $U_{\mathbf{b}}, U'_{\bar{\mathbf{b}}}$, conditioned on some particular $\hat{U}_{\mathbf{b}}, \hat{U}'_{\bar{\mathbf{b}}}$. By assumption of Condition 1

$$1 - p(\text{RCVR}) = \int d\hat{U}_{\mathbf{b}} d\hat{U}'_{\bar{\mathbf{b}}} [1 - p(\text{RCVR}|\hat{U}_{\mathbf{b}}, \hat{U}'_{\bar{\mathbf{b}}})] \le \kappa_1. \tag{351}$$

Hence by Markov's inequality, for any $\kappa_2 \gg \kappa_1$, the inequality

$$1 - p(\text{RCVR}|\hat{U}_{\mathbf{b}}, \hat{U}'_{\bar{\mathbf{b}}}) \le \frac{\kappa_1}{\kappa_2}, \tag{352}$$

is satisfied with probability $p(\hat{U}_{\mathbf{b}}, \hat{U}'_{\bar{\mathbf{b}}}) \ge 1 - \kappa_2$ for Haar random unitaries $\hat{U}_{\mathbf{b}}, \hat{U}'_{\bar{\mathbf{b}}}$. From now on, we assume that $\hat{U}_{\mathbf{b}}, \hat{U}'_{\bar{\mathbf{b}}}$ satisfy (352).

By explicit evaluation

$$\left\| W_{BR} W_{\bar{B}\bar{R}} V \hat{U}_{\mathbf{b}} \hat{U}'_{\bar{\mathbf{b}}} |\psi\rangle - V W_{\mathbf{b}} W_{\bar{\mathbf{b}}} |\Psi\rangle \right\|^2 \le \int dU_{\mathbf{b}} dU'_{\bar{\mathbf{b}}} \left\| U_{BR} U'_{\bar{B}\bar{R}} V \hat{U}_{\mathbf{b}} \hat{U}'_{\bar{\mathbf{b}}} |\psi\rangle - V U_{\mathbf{b}} U'_{\bar{\mathbf{b}}} |\Psi\rangle \right\|^2$$

$$\le \varepsilon_1^2 + \frac{\kappa_1}{\kappa_2}. \tag{353}$$

Applying Lemma C.1, together with Fannes' inequality, then tells us that

$$\left| S(BR)_{V\hat{U}_{\mathbf{b}}\hat{U}'_{\bar{\mathbf{b}}}|\psi\rangle} - \left[ A_B(\mathbf{b}) + S(\mathbf{b}R)_{|\Psi\rangle} \right] \right| = \left| S(BR)_{V\hat{U}_{\mathbf{b}}\hat{U}'_{\bar{\mathbf{b}}}|\psi\rangle} - S(BR)_{VW_{\bar{\mathbf{b}}}W_{\mathbf{b}}|\Psi\rangle} \right| \tag{354}$$

$$\le \sqrt{\varepsilon_1^2 + \frac{\kappa_1}{\kappa_2}} \log \frac{d_B^2 d_R^2}{4(\varepsilon_1^2 + \kappa_1/\kappa_2)}. \tag{355}$$

---

[32]Here we have simply applied the standard Fannes' inequality to $S(AB)$ and $S(B)$ separately. You can improve this bound somewhat (and avoid any $d_B$ dependence) using the Fannes-Alicki inequality [57], but we have not done so in the interests of being self-contained.

**Proof** (2) $\implies$ (1):

Recall from the proof of the exact case in Section 4.2 that

$$S(\overline{B}\,\overline{R}\mathbf{\bar{a}\bar{r}})_{VW_{\mathbf{b}}W_{\bar{\mathbf{b}}}|\Psi\rangle} = S(BR\mathbf{ar})_{VW_{\mathbf{b}}W_{\bar{\mathbf{b}}}|\Psi\rangle} = A_B(\mathbf{b}) + S(\mathbf{b}R)_{|\Psi\rangle}\,, \tag{356}$$

and that

$$S(\overline{B}\,\overline{R}\,\mathbf{\bar{a}\bar{r}})_{VW_{\mathbf{b}}W_{\bar{\mathbf{b}}}|\Psi\rangle} \geq \int dU_{\mathbf{b}} S(BR)_{VU_{\mathbf{b}}W_{\bar{\mathbf{b}}}|\Psi\rangle} \geq \int dU_{\mathbf{b}} dU'_{\bar{\mathbf{b}}} S(BR)_{VU_{\mathbf{b}}U'_{\bar{\mathbf{b}}}|\Psi\rangle}\,. \tag{357}$$

But by Condition 2,

$$\int dU_{\mathbf{b}} dU'_{\bar{\mathbf{b}}} S(BR)_{VU_{\mathbf{b}}U'_{\bar{\mathbf{b}}}|\Psi\rangle} \geq (1-\kappa_2)(A_B(\mathbf{b}) + S(\mathbf{b}R)_{|\Psi\rangle}) - \varepsilon_2 \tag{358}$$

$$\geq A_B(\mathbf{b}) + S(\mathbf{b}R)_{|\Psi\rangle} - \kappa_2 \log(d_B d_R) - \varepsilon_2\,. \tag{359}$$

Hence, by Lemma C.2 (applied to $|\Phi\rangle = W_{\bar{\mathbf{b}}}|\Psi\rangle$), for any $\kappa > 0$ there exists $U_{BR}$ such that, with probability $p \geq 1 - \kappa$,

$$\left\| U_{BR} V \hat{U}_{\mathbf{b}} W_{\bar{\mathbf{b}}} |\Psi\rangle - V U_{\mathbf{b}} W_{\bar{\mathbf{b}}} |\Psi\rangle \right\|^2 \leq \frac{8\sqrt{2\kappa_2 \log(d_B d_R) + 2\varepsilon_2}}{\kappa}\,. \tag{360}$$

Since

$$\left\| U_{BR} V \hat{U}_{\mathbf{b}} W_{\bar{\mathbf{b}}} |\Psi\rangle - V U_{\mathbf{b}} W_{\bar{\mathbf{b}}} |\Psi\rangle \right\|^2 = \int dU'_{\bar{\mathbf{b}}} \left\| U_{BR} V \hat{U}_{\mathbf{b}} U'_{\bar{\mathbf{b}}} |\Psi\rangle - V U_{\mathbf{b}} U'_{\bar{\mathbf{b}}} |\Psi\rangle \right\|^2\,, \tag{361}$$

we can use Markov's inequality to show that

$$\left\| U_{BR} V \hat{U}_{\mathbf{b}} U'_{\bar{\mathbf{b}}} |\Psi\rangle - V U_{\mathbf{b}} U'_{\bar{\mathbf{b}}} |\Psi\rangle \right\|^2 \leq \frac{8\sqrt{2\kappa_2 \log(d_B d_R) + 2\varepsilon_2}}{\kappa^2}\,, \tag{362}$$

with probability $p \geq 1 - 2\kappa$. By an analogous argument, there also exists $U'_{\overline{B}\,\overline{R}}$ (depending only on $U'_{\bar{\mathbf{b}}}$) such that

$$\left\| U'_{\overline{B}\,\overline{R}} V \hat{U}_{\mathbf{b}} \hat{U}'_{\bar{\mathbf{b}}} |\Psi\rangle - V U_{\mathbf{b}} \hat{U}'_{\bar{\mathbf{b}}} |\Psi\rangle \right\|^2 \leq \frac{8\sqrt{2\kappa_2 \log(d_B d_R) + 2\varepsilon_2}}{\kappa^2}\,, \tag{363}$$

with probability $p \geq 1 - 2\kappa$. By the union bound, both (362) and (363) are true with probability $p \geq 1 - 4\kappa$. Using the triangle inequality, and choosing $\kappa_1 = 4\kappa$, we find

$$\left\| U_{BR} U'_{\overline{B}\,\overline{R}} V \hat{U}_{\mathbf{b}} \hat{U}'_{\bar{\mathbf{b}}} |\Psi\rangle - V U_{\mathbf{b}} U'_{\bar{\mathbf{b}}} |\Psi\rangle \right\| \leq \left\| U_{BR} U'_{\overline{B}\,\overline{R}} V \hat{U}_{\mathbf{b}} \hat{U}'_{\bar{\mathbf{b}}} |\Psi\rangle - U_{BR} V \hat{U}_{\mathbf{b}} U'_{\bar{\mathbf{b}}} |\Psi\rangle \right\|$$
$$+ \left\| U_{BR} V \hat{U}_{\mathbf{b}} U'_{\bar{\mathbf{b}}} |\Psi\rangle - V U_{\mathbf{b}} U'_{\bar{\mathbf{b}}} |\Psi\rangle \right\| \tag{364}$$

$$\leq \left\| U'_{\overline{B}\,\overline{R}} V \hat{U}_{\mathbf{b}} \hat{U}'_{\bar{\mathbf{b}}} |\Psi\rangle - V \hat{U}_{\mathbf{b}} U'_{\bar{\mathbf{b}}} |\Psi\rangle \right\|$$
$$+ \left\| U_{BR} V \hat{U}_{\mathbf{b}} U'_{\bar{\mathbf{b}}} |\Psi\rangle - V U_{\mathbf{b}} U'_{\bar{\mathbf{b}}} |\Psi\rangle \right\| \tag{365}$$

$$\leq \frac{16[8\kappa_2 \log(d_B d_R) + 8\varepsilon_2]^{1/4}}{\kappa_1}\,. \tag{366}$$

**Proof** $(1) \implies (3)$**:**

By Condition 1, there exists $\hat{U}_{\mathbf{b}}, \hat{U}'_{\bar{\mathbf{b}}}$ such that (139) is satisfied for a randomly chosen $U_{\mathbf{b}}, U'_{\bar{\mathbf{b}}}$ with probability $p \geq 1 - \kappa_1$ and thus

$$\left\| W_{BR} W_{\bar{B}\bar{R}} V \hat{U}_{\mathbf{b}} \hat{U}'_{\bar{\mathbf{b}}} |\Psi\rangle - V W_{\mathbf{b}} W_{\bar{\mathbf{b}}} |\Psi\rangle \right\| = \int dU_{\mathbf{b}} dU'_{\bar{\mathbf{b}}} \left\| U_{BR} U'_{\bar{B}\bar{R}} V \hat{U}_{\mathbf{b}} \hat{U}'_{\bar{\mathbf{b}}} |\psi\rangle - V U_{\mathbf{b}} U'_{\bar{\mathbf{b}}} |\Psi\rangle \right\| \quad (367)$$

$$\leq \varepsilon_1 + \kappa_1. \quad (368)$$

By Lemma C.1 and monotonicity, we have

$$\left\| \rho - \sigma \right\|_1 \leq 2\varepsilon_1 + 2\kappa_1, \quad (369)$$

where $\rho$ and $\sigma$ are the reduced states on $\mathcal{H}_B \otimes \mathcal{H}_R \otimes \mathcal{H}_{\mathbf{U}_{\bar{\mathbf{b}}'}} \otimes \mathcal{H}_{\mathbf{U}_{\mathbf{b}}}$ of $W_{BR} W_{\bar{B}\bar{R}} V \hat{U}_{\mathbf{b}} \hat{U}'_{\bar{\mathbf{b}}} |\Psi\rangle$ and $V W_{\mathbf{b}} W_{\bar{\mathbf{b}}} |\Psi\rangle$ respectively. As in the proof of Lemma C.2, these states can be extended to normalized states $\hat{\rho}, \hat{\sigma}$ on larger "primed" Hilbert spaces, while at most doubling the Hilbert space distance between them. Hence by the first Fuchs-van de Graaf inequality, the generalized fidelity

$$\bar{F}(\rho, \sigma) \geq F(\hat{\rho}, \hat{\sigma}) \geq 1 - (2\varepsilon_1 + 2\kappa_1). \quad (370)$$

It follows, if $\varepsilon_3 = \sqrt{2\varepsilon_1 + 2\kappa_1}$, then

$$H_{\min}^{\varepsilon_3}(\bar{\mathbf{a}}'\bar{\mathbf{r}}'|B R \, \mathbf{ar})_{V W_{\mathbf{b}} W_{\bar{\mathbf{b}}} |\Psi\rangle} \geq H_{\min}(\mathbf{U}_{\bar{\mathbf{b}}'}|B R \mathbf{U}_{\mathbf{b}})_{W_{BR} W_{\bar{B}\bar{R}} V \hat{U}_{\mathbf{b}} \hat{U}'_{\bar{\mathbf{b}}} |\Psi\rangle}. \quad (371)$$

But, as in (123), if $|\phi\rangle = W_{BR} W_{\bar{B}\bar{R}} V \hat{U}_{\mathbf{b}} \hat{U}'_{\bar{\mathbf{b}}} |\Psi\rangle$, then

$$\phi_{BR\mathbf{U}_{\mathbf{b}}\mathbf{U}_{\bar{\mathbf{b}}'}} \leq \phi_{BR\mathbf{U}_{\mathbf{b}}} \otimes \mathbb{1}_{\mathbf{U}_{\bar{\mathbf{b}}'}}, \quad (372)$$

and so

$$H_{\min}(\mathbf{U}_{\bar{\mathbf{b}}'}|B R \mathbf{U}_{\mathbf{b}})_{W_{BR} W_{\bar{B}\bar{R}} V \hat{U}_{\mathbf{b}} \hat{U}'_{\bar{\mathbf{b}}} |\Psi\rangle} \geq 0. \quad (373)$$

From this, Condition 3 follows by Lemma 4.7.

**Proof** $(3) \implies (4)$**:**

Recall from (127) in the proof of the exact version in Section 4.2, that by strong subadditivity

$$A_B(\mathbf{b}') \geq A_B(\mathbf{b} \cup \mathbf{b}') + A_B(\mathbf{b}' \cap \mathbf{b}') - A_B(\mathbf{b}). \quad (374)$$

Hence, by Condition 3, and the equality of complementary areas (Remark 2.2), we have

$$A_B(\mathbf{b}') \geq A_B(\mathbf{b}) - H_{\min}^{\varepsilon_3}(\mathbf{b}' \setminus \mathbf{b}|\mathbf{b}R)_{|\Psi\rangle} - H_{\min}^{\varepsilon_3}(\bar{\mathbf{b}}' \setminus \bar{\mathbf{b}}|\bar{\mathbf{b}}\bar{R})_{|\Psi\rangle}. \quad (375)$$

Finally, applying Lemma C.3 to the two smooth min-entropies gives

$$A_B(\mathbf{b}') \geq A_B(\mathbf{b}) + 2S(\mathbf{b}R)_{|\Psi\rangle} - S([\mathbf{b} \cup \mathbf{b}']R)_{|\Psi\rangle} - S([\mathbf{b} \cap \mathbf{b}']R)_{|\Psi\rangle} - 4\varepsilon_3 \log \frac{d_{\text{code}}^2 d_R d_{\bar{R}}}{4\varepsilon_3^2}, \quad (376)$$

from which Condition 4 follows immediately by strong sub-additivity.

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
