# Peer review of "Quantum minimal surfaces from quantum error correction"

_SciPost Physics, doi:SciPost Phys. 12, 157 (2022)_

## Round 2 · Referee Report · Anonymous · 2022-2-18

Report
This paper presents a significant generalization of previous work on the emergence of a quantum minimal surface prescription (emergent "geometry") from quantum error correcting codes with complementary recovery. In particular, the authors close an important gap in the subject by studying a situation, which often occurs in AdS/CFT: code spaces where different states have different entanglement wedges. In such cases the reconstruction map becomes state-dependent. The central result of the paper is a theorem, which states that complementary state-dependent reconstruction is equivalent to the existence/emergence of a "geometric" prescription for computing entropies (where "area" is defined as bounding arbitrary collections of logical subsystems, i.e., bulk regions). Furthermore, if such a prescription exists, then minimality of the subsystem in question follows (any other "bulk" subsystem has a larger generalized entropy). Robustness against small errors is demonstrated.
The paper is written very carefully and pedagogically (for high energy theorists). The main ideas, context, and results are all explained very clearly and intuitively before the main theorem is proven rigorously. The discussion section is inspiring and addresses some important questions. Appendix B is a nice bonus as it provides further intuition and checks for the formal results. All this offers a nice balance of intuitive explanations and precise arguments. I think that this paper is a significant contribution to the topic of quantum error correction in holography and also provides a comprehensive framework for understanding several previous ideas in much greater generality. I recommend publication.
Suggestions:
- The paper explains in detail how geometry emerges from quantum information structure on time-reflection symmetric slices. In the discussion the authors briefly comment that this is still a long way from seeing the emergence of gravitational dynamics (e.g. Einstein's equations). As this seems to be one of the big questions motivating the subject, can more be said/speculated about what will be needed in this language to see the emergence of not just spatial geometry, but a time direction?
- Typo in the penultimate sentence on p.6
Author: Christopher Akers on 2022-04-18 [id 2392]
(in reply to Report 1 on 2022-02-18)Thank you for this report! These are good suggestions and we have implemented them in the new version.
Author: Christopher Akers on 2022-04-18 [id 2393]
(in reply to Report 2 on 2022-03-02)We thank you for these suggestions and have implemented them in the new version.

---

## Round 2 · Referee Report · Anonymous · 2022-3-2

Report
In this manuscript, the authors show an important relation between quantum codes (with complementary state-specific product unitary) and geometry (areas and the quantum minimal surface prescription).
This is a very interesting paper which I recommend for publication in SciPost Physics. I have two minor suggestions below.
Requested changes
1. Remark 2.5 states that ``we reserve $A_B$ and EW to denote the use of area as defined in Definition 2.1." But Definition 2.1 defines only $A_B$, not EW. For EW, it seems clearer to refer to the definition around Eqs. (1.6) and (1.7).
2. In the third paragraph of Section 6.1, $e^{\mathcal{O}(1/G)}$ should probably be changed to $e^{-\mathcal{O}(1/G)}$.

---

## Round 3 · Author Response

We thank the reviewers, and we have modified the paper based on their suggestions.

---

## Round 3 · List of Changes

1. We fixed typos, including a minus sign in the third paragraph of section 6.1 and in page 6.
  2. We clarified Remark 2.5.
  3. We added a section 6.4 about extremality, in which we discuss a natural next step towards understanding holographic dynamics in these codes.

---

## Editorial Decision

published